# Sci2Pol: Evaluating and Fine-tuning LLMs on Scientific-to-Policy Brief Generation

**Weimin Wu**[†,1]    **Alexander C. Furnas**[‡]    **Eddie Yang**[‡]    **Gefei Liu**[♮]
**Akhil Pandey Akella**[‡]    **Xuefeng Song**[†]    **Dashun Wang**[‡*,2]    **Han Liu**[†§*,3]

[†] Center for Foundation Models and Generative AI, Northwestern University, IL 60208, USA
   Department of Computer Science, Northwestern University, IL 60208, USA

[‡] Center for Science of Science and Innovation, Northwestern University, IL 60208, USA
   Kellogg School of Management, Northwestern University, IL 60208, USA

[♮] Department of Computer Science, Brown University, RI 02912, USA

[§] Department of Statistics and Data Science, Northwestern University, IL 60208, USA

[1] wwm@u.northwestern.edu

[2] dashun.wang@kellogg.northwestern.edu

[3] hanliu@northwestern.edu

## Abstract

We propose **Sci2Pol-Bench** and **Sci2Pol-Corpus**, the first benchmark and training dataset for evaluating and fine-tuning large language models (LLMs) on policy brief generation from a scientific paper. We build Sci2Pol-Bench on a five-stage taxonomy to mirror the human writing process: (i) Autocompletion, (ii) Understanding, (iii) Summarization, (iv) Generation, and (v) Verification. It features 18 tasks in multiple-choice and open-ended formats. Specifically, for the Generation stage, we show that BERTScore and ROUGE scores fail to capture the quality of brief writing, and introduce a new LLM-based evaluation metric aligned with expert judgement. Using this benchmark, we evaluate 13 leading open-source and commercial LLMs to uncover key limitations. To improve LLM performance on brief writing, we curate the Sci2Pol-Corpus for fine-tuning. We start by linking each cited scientific paper to its corresponding policy document, drawn from 5.6 million policy records. This produces 140,000 candidate pairs. We then employ an LLM-as-a-judge to filter high-quality examples, followed by in-context polishing using three expert-written samples as references. This process yields a final set of 639 new pairs. Finally, we fine-tune three models on Sci2Pol-Corpus: LLaMA-3.1-8B, Gemma-12B, and Gemma-27B. Fine-tuning leads to consistent performance improvements across Sci2Pol-Bench. Notably, after fine-tuning, Gemma-27B surpasses the much larger GPT-4o and DeepSeek-V3 (671B). These demonstrate the effectiveness of our corpus in bridging the gap between science and policy.

**Project Page**: https://github.com/WeiminWu2000/Sci2Pol

**Keywords:** Benchmark, Dataset, Science, Policy, LLM

## 1 Introduction

We propose **Sci2Pol-Bench** and **Sci2Pol-Corpus**, the first benchmark and training dataset for evaluating and fine-tuning LLMs on scientific-to-policy brief generation. A policy brief is a concise article that distills the content of a technical scientific paper for a policymaker audience (Appendix C). Turning scientific evidence into policy remains critical and difficult for both policymakers and scientists. Today's major challenges (e.g., climate change, public health, and rapid technological shifts) require timely input from science (Wang and Barabási, 2021). Yet policymakers often struggle to convert dense, technical research into clear and usable guidance. This issue is also relevant to the scientific community, as it underscores the essential role of science in shaping societal outcomes. However, most scientists lack policy expertise. This gap limits how science informs real-world decisions (Straf et al., 2012). With the rise of powerful LLMs, we ask two key questions: (i) To what extent can LLMs assist in scientific-to-policy brief generation? (ii) How can their performance be further

---

[*]Co-corresponding authors.

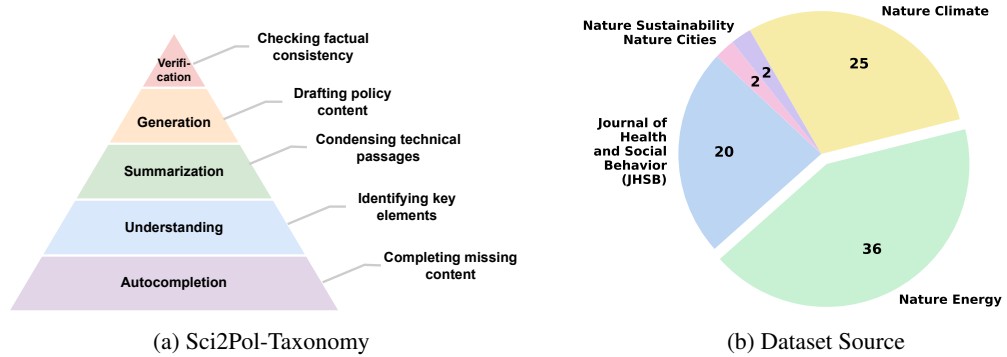

(a) Sci2Pol-Taxonomy

(b) Dataset Source

Figure 1: **Overview of Sci2Pol-Taxonomy and Dataset Source.** (a) Sci2Pol-Taxonomy defines a five-stage decomposition of the brief writing process. (b) Complete set of published 85 pairs.

improved? To address these, we introduce Sci2Pol-Bench and Sci2Pol-Corpus. Sci2Pol-Bench offers the first comprehensive benchmark for fine-grained evaluation of scientific-to-policy generation. Sci2Pol-Corpus provides the first targeted training dataset to enhance LLM performance on this task.

Although LLMs demonstrate strong general capabilities, prior work shows they hallucinate scientific content, misverify claims, and produce unstable or biased policy reasoning (Coz et al., 2025; Bai et al., 2025; Wadden et al., 2024). To further validate these concerns, we start with expert-reviewed examples and highlight four key limitations in policy brief generation (Appendix F): (i) Missing core content: LLMs often fail to capture a study's essential details, including its quantitative findings, methods, and broader context. They omit key facts or add irrelevant information (see Appendix F.1). (ii) Hallucinated claims: LLMs invent numbers or causal statements that do not appear in the source paper (see Appendix F.2). (iii) Inappropriate tone: Even when accurate, the language is often too technical or overly verbose for policy audiences (see Appendix F.3). (iv) Low actionability: Recommendations tend to be vague and weakly supported by evidence (see Appendix F.4).

Rigorous evaluation in this domain requires a clear decomposition of the writing process and a dataset that challenges models with authentic, domain-matched targets. To this end, we define the Scientific-to-Policy Taxonomy (**Sci2Pol-Taxonomy**, Figure 1a), a five-stage framework for the brief writing workflow: (i) Autocompletion: Completing missing content in scientific or policy texts; (ii) Understanding: Identifying and interpreting key claims, caveats, and methods; (iii) Summarization: Condensing technical passages into language accessible to policy audiences; (iv) Generation: Drafting coherent, persuasive prose that integrates evidence with policy context; (v) Verification: Checking factual consistency of claims or drafts against the source literature.

Grounded in this taxonomy, we build **Sci2Pol-Bench**, a suite of 18 diverse tasks for evaluating model performance in generating policy briefs from scientific research. We construct domain-specific targets from 85 expert-written paper-brief pairs, representing the complete set of published pairs. These pairs are drawn from high-impact venues, including *Nature Energy*, *Nature Climate Change*, *Nature Cities*, *Nature Sustainability*, and the *Journal of Health and Social Behavior*. Figure 1b presents summary statistics of these pairs. Each task takes the form of either a multiple-choice probe or an open-ended format. They enable fine-grained evaluation along both correctness and writing quality. For the Generation stage, we show that BERTScore and ROUGE scores fail to capture the quality of the brief writing, and we design a dedicated evaluation metric using an LLM-as-a-judge approach. We then conduct the first large-scale evaluation of scientific-to-policy brief generation across 13 LLMs, covering both open-source and commercial models. Our results show that even advanced LLMs such as Grok and DeepSeek-R1 leave substantial room for improvement. These findings highlight concrete research opportunities in controllable generation and domain adaptation for policy applications.

To further improve LLM performance on policy brief generation, we introduce **Sci2Pol-Corpus**, the first training dataset designed for this task. The construction process consists of two resource-intensive steps followed by a novel polishing step. (i) Retrieving candidate science-policy pairs: We begin with a large-scale collection of 5.6 million policies indexed by Overton.io and scraped as PDFs from the public websites that published them (e.g., Government Printing Office, Congress.gov,

The Brookings Institution, The World Bank) (Furnas et al., 2025; Szomszor and Adie, 2022). From these, we construct 140,000 citation-based candidate pairs by linking policy documents to their cited scientific papers. To ensure tighter relevance, we filter for policy files that cite no more than three papers and treat each citation as a potential paper-brief pair. (ii) Filtering high-quality pairs with an LLM-as-a-judge approach: We employ GPT-o3 to evaluate whether each policy document is centered on the cited scientific paper. This two-stage filtering process yields 639 high-quality pairs. (iii) Refining briefs through in-context revision: To further align the collected briefs with the style and structure of expert-written examples, we select three high-quality samples from the 85 published pairs. We then use GPT-o3 in an in-context learning setup to revise the 639 identified policy briefs. This step ensures their tone, structure, and clarity match the expert-written briefs. Finally, we use Sci2Pol-Corpus to fine-tune three open-source LLMs: LLaMA-3.1-8B-Instruct, Gemma-12B-Instruct, and Gemma-27B-Instruct. Fine-tuning on Sci2Pol-Corpus leads to consistent performance improvements across Sci2Pol-Bench. Notably, after supervised fine-tuning, Gemma-27B-Instruct surpasses the much larger GPT-4o and DeepSeek-V3 (671B). These demonstrate the effectiveness of our corpus in scientific-to-policy brief generation.

In summary, we have the following three contributions:

- We propose **Sci2Pol-Bench**, the first comprehensive benchmark for evaluating LLMs on policy brief generation from scientific papers. We build the benchmark on a five-stage workflow framework, **Sci2Pol-Taxonomy**, and include the complete set of 85 published expert-written paper-brief pairs. It features 18 tasks and provides the first large-scale evaluation across 13 LLMs.
- We curate **Sci2Pol-Corpus**, the first training dataset for fine-tuning LLMs on policy brief generation. It consists of two resource-intensive steps followed by a novel polishing step: (i) retrieving 140,000 candidate science-policy pairs by linking cited scientific papers across 5.6 million policy documents; (ii) selecting 639 high-quality pairs using an LLM-as-a-judge filtering method; and (iii) enhancing these pairs via in-context revision to improve clarity and alignment.
- We use **Sci2Pol-Corpus** to fine-tune three open-source LLMs: LLaMA-3.1-8B-Instruct, Gemma-12B-Instruct, and Gemma-27B-Instruct. Fine-tuning leads to consistent performance improvements across **Sci2Pol-Bench**. Notably, after supervised fine-tuning, Gemma-27B-Instruct surpasses the much larger GPT-4o and DeepSeek-V3 (671B). These demonstrate the effectiveness of Sci2Pol-Corpus in enhancing LLM capabilities for scientific-to-policy brief generation.

**Organization.** Section 2 introduces Sci2Pol-Bench. Section 3 details the Sci2Pol-Corpus. Section 4 shows the evaluation results across 13 LLMs, and supervised fine-tuning results.

## 2 SCI2POL-BENCH

We detail our Sci2Pol-Bench here. Section 2.1 outlines the core design principles. Section 2.2 describes the data collection and processing pipeline. Section 2.3 presents the task definitions and summarizes the dataset. Section 2.4 details the evaluation metrics, with a particular focus on an LLM-based evaluation metric for Generation-related tasks to better align with expert judgement.

### 2.1 DESIGN PRINCIPLE

The overall goal of Sci2Pol-Bench is to provide researchers and practitioners with a transparent, fine-grained assessment of how well LLMs can translate dense scientific research into actionable policy briefs. Inspired by the progressive, ability-oriented evaluation framework of Li et al. (2024a), we introduce a five-stage Sci2Pol-Taxonomy to mirror the workflow of brief writing.

As illustrated in Figure 1a, the taxonomy defines five levels of ability: (i) Autocompletion tasks require LLMs to predict continuations or recombine sentences. This tests their grasp of local cohesion and textual fluency. (ii) Understanding tasks involve classifying sentence intent and answering multiple-choice questions. This evaluates the model's factual comprehension and ability to process complex research narratives. (iii) Summarization tasks focus on distilling scientific text into concise summaries. These tasks assess the ability to extract salient points and deliver them. (iv) Generation tasks ask LLMs to compose new policy brief content from scratch. This requires synthesizing scientific evidence, contextual understanding, and persuasive framing into coherent, structured writing. (v) Verification tasks challenge models to fact-check scientific or policy-related claims against the source literature. This is critical for mitigating hallucinations. Following the five stages, Sci2Pol-Bench provides a robust framework to evaluate the LLMs in bridging science and policy.

Table 1: **Summary of Sci2Pol-Bench.** Sci2Pol-Bench comprises 18 tasks. Each task specifies an ID, task description, data source, sample size, and evaluation metric.

| Taxonomy | ID | Task Description | Source | Size | Metric |
|---|---|---|---|---|---|
| **Autocompletion** | 1 | Scientific Text Autocompletion | Our Dataset | 255 | Micro F1 |
| | 2 | Political Text Autocompletion | Our Dataset | 255 | Micro F1 |
| | 3 | Scientific Sentence Reordering | Our Dataset | 255 | Micro F1 |
| | 4 | Political Sentence Reordering | Our Dataset | 255 | Micro F1 |
| **Understanding** | 5 | Sentence Classification | Our Dataset | 1200 | Micro F1 |
| | 6 | Scientific Knowledge Understanding | MMLU-Pro | 1000 | Micro F1 |
| **Summarization** | 7 | Policy Problem Summarization | Our Dataset | 200 | Reference-free Score |
| | 8 | Research Findings Summarization | Our Dataset | 200 | Reference-free Score |
| | 9 | Study Methods Summarization | Our Dataset | 200 | Reference-free Score |
| | 10 | Policy Implications Summarization | Our Dataset | 200 | Reference-free Score |
| **Generation** | 11 | Policy Problem Generation | Our Dataset | 85 | Reference-based Score |
| | 12 | Research Findings Generation | Our Dataset | 85 | Reference-based Score |
| | 13 | Study Methods Generation | Our Dataset | 85 | Reference-based Score |
| | 14 | Policy Implications Generation | Our Dataset | 85 | Reference-based Score |
| | 15 | Policy Brief Generation | Our Dataset | 85 | Reference-based Score |
| **Verification** | 16 | Scientific Claims Verification | Our Dataset | 850 | Micro F1 |
| | 17 | Scientific Claims Verification 2 | SciRIFF | 1000 | Micro F1 |
| | 18 | Policy Implications Verification | Our Dataset | 700 | Micro F1 |

## 2.2 DATA COLLECTION AND PROCESSING

**Data Sources.** Sci2Pol-Bench draws from two sources: (i) **Existing benchmarks:** We incorporate tasks from established datasets focused on scientific understanding, including SciRIFF (Wadden et al., 2024) and MMLU-Pro (Wang et al., 2024). (ii) **Newly collected pairs:** We curate the expert-authored paper-brief pairs published across top journals, as summarized in Figure 1b. To ensure data quality, we apply a strict inclusion criterion: the policy brief must be written by the same authors as the original scientific paper. This guarantees that the policy content reflects authentic expert interpretation. The final collection consists of 85 high-quality pairs (see Appendix J.1 for the full list).

**Data Processing and Annotation.** We download each article's metadata and full text to maintain quality control. Most policy briefs share titles with their corresponding scientific articles, so we face little disambiguation. We apply OCR to extract text from PDFs and use a light preprocessing script to remove tabs, extra spaces, and stray characters. We store all data in structured JSON format.

## 2.3 TASK DEFINITION AND DATASET SUMMARY

Guided by the Sci2Pol-Taxonomy, we construct 18 evaluation tasks (Table 1). Two tasks come from existing datasets, and experts construct the remaining sixteen. We describe each task in detail below.

### 2.3.1 AUTOCOMPLETION (TASKS 1-4)

These tasks probe local coherence: given a short scientific or policy passage, the model selects the next sentence (multiple choice) or restores a shuffled sequence.

**Task 1: Scientific Text Autocompletion (Multiple Choice).** This task tests local discourse coherence in scientific writing. The dataset contains 255 items drawn from 85 articles, with 3 items selected per paper. We construct each example in two steps: (i) Extract a sequence of three consecutive sentences from the article. Use the first two sentences as the prompt and the third as the gold (correct) continuation. (ii) Construct five candidate continuations for each prompt: one gold sentence (the true continuation) and four distractor sentences sampled from the nearby context in the same paper. We ask the model to choose the correct answer from five options. See Table 45 for an example.

**Task 2: Political Text Autocompletion (Multiple Choice).** This task tests coherence in policy writing. The dataset contains 255 items from 85 professional policy briefs, with 3 items selected per brief. We construct each example in two steps: (i) Extract a sequence of three consecutive sentences from the brief. Use the first two sentences as the prompt and the third as the gold (correct) continuation. (ii) Construct five candidate continuations for each prompt: one gold sentence (the true

continuation) and four distractor sentences sampled from the nearby context in the same brief. We ask the model to choose the correct answer from five options. See Table 46 for an example.

**Task 3: Scientific Sentence Reordering (Recombination).** This task probes discourse-level coherence in scientific writing. We reuse the 255 three-sentence triplets from Task 1, shuffle each triplet, and ask the model to restore the original order. See Table 47 for an example.

**Task 4: Political Sentence Reordering (Recombination).** This task probes discourse coherence in policy briefs. We reuse the 255 three-sentence triplets from Task 2, shuffle each triplet, and ask the model to restore the original order. See Table 48 for an example.

### 2.3.2 UNDERSTANDING (TASKS 5-6)

Tasks 5-6 evaluate sentence-level comprehension and scientific knowledge. Task 5 classifies sentences from scientific papers into five policy brief-relevant categories. Task 6 evaluates scientific knowledge.

**Task 5: Sentence Classification (Multiple Choice).** This task assesses the classification of policy brief-relevant sentences by rhetorical role or content. The dataset includes 1,200 manually verified samples from paper-brief pairs. We curate in three steps: (i) For each pair, use a templated prompt (see Appendix L.7) to generate 15 labeled examples with GPT-o3; (ii) Political experts manually review labels and language quality; (iii) Select a random subset of 1,200 samples. Each instance consists of one sentence and a label from a fixed set (e.g., *Policy Problem*, *Scientific Research Findings*, *Scientific Research Study Methods*, *Policy Implications*, *None*). See Table 49 for an example.

**Task 6: Scientific Knowledge Understanding (Multiple Choice).** This task evaluates broad scientific knowledge with multiple-choice questions from MMLU-Pro (Wang et al., 2024). The full subset contains 3,511 questions across health, chemistry, economics, and biology. We treat these as one task and sample 1,000 questions randomly for evaluation. This baseline enables comparison between general-purpose and science-specialized models. See Table 50 for an example.

### 2.3.3 SUMMARIZATION (TASKS 7-10)

Tasks 7-10 evaluate policy-oriented summarization. We show the details as follows.

**Task 7: Policy Problem Summarization (Writing).** We evaluate the ability to identify and summarize the policy problem motivating a study. The dataset includes 200 examples. We curate the dataset following three steps: (i) For each pair, select up to three paragraphs relevant to the brief's policy problem; (ii) If fewer exist, include all reliably matched paragraphs; (iii) From the final 233 paragraph sets, sample 200 randomly for evaluation. See Table 51 for an example.

**Task 8: Research Findings Summarization (Writing).** We evaluate the summarization of core scientific findings. The dataset includes 200 examples. We curate the dataset following two steps: (i) For each pair, select three paragraphs aligned with the brief's findings; (ii) From 255 sets, sample 200 randomly for evaluation. See Table 52 for an example.

**Task 9: Study Methods Summarization (Writing).** We evaluate the summarization of methodology in policy brief-relevant terms. The dataset includes 200 examples. We curate it following two steps: (i) For each pair, select three paragraphs that describe the methodology reflected in the brief; (ii) From 255 sets, sample 200 randomly for evaluation. See Table 53 for an example.

**Task 10: Policy Implications Summarization (Writing).** We evaluate the articulation of policy implications grounded in the source paper. The dataset includes 200 examples. We curate the dataset following three steps: (i) For each pair, select up to three paragraphs that support the brief's implications; (ii) If fewer exist, include all reliably matched paragraphs; (iii) From 222 sets, sample 200 randomly for evaluation. See Table 54 for an example.

### 2.3.4 GENERATION (TASKS 11-15)

These tasks evaluate brief generation: policy problem, findings, methods, implications, and full brief. We provide a detailed justification for separating section-by-section generation (Tasks 11-14) from full-brief generation (Task 15) in Appendix H.7. In short, Tasks 11-14 complement Task 15 by disentangling factual precision from holistic coherence. Section-level generation emphasizes accurate grounding, while full-brief generation assesses overall readability. Evaluating both provides a more comprehensive view of LLM performance and uncovers trade-offs that Task 15 alone can not capture.

**Task 11: Policy Problem Generation (Writing).** We generate the *Policy Problem* section from the full scientific paper (85 examples). For each pair, we extract the brief's *Policy Problem* as a reference and provide the full paper as input. For 20 *Journal of Health and Social Behavior* pairs, they lack a clear *Policy Problem* section. We prompt GPT-o3 (see Appendix L.8) to construct the *Policy Problem* section with inputs: the full paper, the brief's *Research Problem & Data* section, and three in-context expert-written examples from the remaining 65 *Nature* journals. See Table 55 for an example.

**Task 12: Research Findings Generation (Writing).** We generate the *Research Findings* section from the full paper (85 examples). For each pair, we extract the brief's *Research Findings* as reference and provide the full paper as input. See Table 56 for an example.

**Task 13: Study Methods Generation (Writing).** We generate the *Research Study* section (85 examples). For each pair, we extract the brief's *Research Study* as reference and provide the full paper as input. For 20 *Journal of Health and Social Behavior* pairs, the briefs lack a clear *Research Study* section. We prompt GPT-o3 (see Appendix L.9) to construct the *Research Study* section using the full paper, the brief's *Research Problem & Data* section, and three in-context expert-written examples from the remaining 65 *Nature* journals. See Table 57 for an example.

**Task 14: Policy Implications Generation (Writing).** We generate the *Policy Implications* section (85 examples). For each pair, we extract the ground-truth policy implications as a reference and provide the full scientific paper as input. See Table 58 for an example.

**Task 15: Policy Brief Generation (Writing).** We generate an entire policy brief (85 examples). For each pair, we build the reference by concatenating *Title*, *Policy Problem*, *Research Findings*, *Research Study*, and *Policy Implications*. Input is the full paper. See Table 59 for an example.

### 2.3.5 VERIFICATION (TASKS 16-18)

Tasks 16-18 assess consistency between claims (findings or policy implications) and the source paper.

**Task 16: Scientific Claims Verification (Multiple Choice).** We evaluate whether a scientific claim is supported by the paper (850 samples). We construct in two steps: (i) For each paper, prompt GPT-o3 (see Appendix L.10) to generate 10 labeled samples (total 850); (ii) Our political experts manually review all samples and correct three issues. See Table 60 for an example.

**Task 17: Scientific Claims Verification 2 (Multiple Choice).** We evaluate claim-evidence entailment using SciRIFF subsets (Wadden et al., 2024): *covidfact entailment*, *healthver entailment*, and *scifact entailment* (1,220 samples). We sample 1,000 randomly for evaluation. Each instance presents a claim (e.g., *support*) and associated evidence. See Table 61 for an example.

**Task 18: Policy Implications Verification (Multiple Choice).** We evaluate whether a policy implication follows from the paper (700 samples). We construct in four steps: (i) Extract all implications from each brief and label them *support*; (ii) Prompt GPT-o3 to generate contradicted implications (see Appendix L.11) and label them *contradict* after manual review of our experts; (iii) Combine to yield 706 samples; (iv) Sample 700 randomly for evaluation. Each instance includes a paper, a policy implication, and a label (*support* or *contradict*). See Table 62 for an example.

### 2.4 EVALUATION METRICS

In this part, we report detailed evaluation metrics for each task, with a particular focus on Tasks 11-15.

**Micro F1 (Tasks 1-6, 16-18).** We compute per-item correctness and use Micro-F1 as the main score. We choose this metric for classification tasks with a firm correct answer. This group includes autocompletion (Tasks 1-4), understanding (Tasks 5-6), and verification (Tasks 16-18).

**Reference-free Score (Tasks 7-10).** We use Gemini-2.5-Pro as an LLM judge to score section summaries. The judge evaluates four dimensions of a summary: *clarity*, *accuracy*, *coverage*, and *overall quality*. We select this metric because these tasks involve free-form writing without a single referenced correct answer. See Appendix L.1 for the full prompt and calculation details.

For Tasks 11–15, we first demonstrate the limitations of BERTScore and ROUGE scores, and then introduce a task-specific reference-based score for more accurate evaluation.

**Limitations in BERTScore and ROUGE Scores for Tasks 11-15.** As shown in Appendix H.1, BERTScore remains high even when key sections are missing, as overlapping words inflate similarity.

ROUGE scores penalize paraphrasing and drop sharply with minor grammatical changes, despite preserved meaning. Neither metric captures reasoning, structure, or evidence linkage.

**Reference-based Score (Tasks 11-15).** We evaluate the generation tasks using content-aware LLM judging, guided by paper-grounded rubrics that rely on both the paper and policy brief sections.

**Task 11 (Policy Problem).** We score by content and structure because a policy problem contains linked and causal sentences. We describe five parts as the full space of content in a policy problem, but any subset may appear, and the order may vary. (i) *Background* sets the scene. (ii) *Existing problem* states the current obstacle. (iii) *Consequences* describe risks if the problem stays unsolved. (iv) *Attention problem* names the issue that calls for action. (v) *Supporting detail* adds facts, numbers, or sources that help this flow. For each part, we judge two things: its *importance* in the paper and its *quality* in the candidate. This checks what to say and how well it is said, balancing relevance and quality. See Appendix L.2 for the full prompt and calculation details.

**Task 12 (Research Findings).** We score by content only because findings are mostly independent. The judge rates five aspects. We check (i) *completeness*, (ii) *importance*, and (iii) *accuracy* of the candidate findings. (iv) *Summarizing findings* checks if the text highlights the key results rather than a long list. (v) *Specification to findings* checks scope, context, and limits. This rubric rewards correct, essential, and well-focused content. See Appendix L.3 for the full prompt and calculation details.

**Task 13 (Study Methods).** We score by content only because method points are independent. The LLM judge rates three aspects. (i) *Clarity and purpose* checks if the text explains what method is used and why, in a clear output. (ii) *Technicality appropriateness* checks if the level of detail fits a policy audience without jargon. (iii) *Explanation of terms* checks if models, data, and acronyms are explained in plain words. Note that *clarity and purpose* and *technicality appropriateness* carry more weight in evaluation, because *explanation of terms* only serves as an extra signal. This rubric rewards clear intent, appropriate detail, and good definitions. See Appendix L.4 for details.

**Task 14 (Policy Implications).** We score by content only because implications are written as separate points. The LLM judge rates four aspects. (i) *Accuracy* checks if the implications are supported by the paper without speculation or hallucination. (ii) *Coverage* checks if all major implications are included. (iii) *Conciseness and distinctness* checks if each implication is concise and non-redundant. (iv) *Alignment with paper intent* checks if the implications match the paper's main message, such as a recommendation, warning, or call to awareness. This rubric rewards grounded, complete, and actionable implications. See Appendix L.5 for the full prompt and calculation details.

**Task 15 (Full Policy Brief).** We score by content and style together. The LLM judge rates four aspects. (i) *Contextual depth* checks if the brief captures key findings, methods, and context without missing facts or adding fluff. (ii) *Hallucination risk* checks if every claim is traceable to the paper, with penalties for unsupported numbers or causal links. (iii) *Readability tone* checks if the text is concise, structured, active, and suitable for policymakers. (iv) *Actionability* checks if the implications are concrete, tied to evidence, and useful for policy. This rubric rewards briefs that are accurate, clear, credible, and practical. See Appendix L.6 for the full prompt and calculation details.

## 3    SCI2POL-CORPUS

In this section, we give the details of the Sci2Pol-Corpus. It comprises 639 high-quality paper-brief pairs curated from 5.6 million policy documents. It includes three steps: (i) Retrieving candidate science-policy pairs (Section 3.1); (ii) Filtering high-quality pairs with an LLM-as-a-judge approach (Section 3.2); (iii) Polishing briefs through in-context revision (Section 3.3).

### 3.1    CANDIDATE PAIR RETRIEVAL

We begin with a large-scale political dataset collection. This collection is derived from documents indexed by Overton, the world's largest database of policy literature(Furnas et al., 2025; Szomszor and Adie, 2022; Yin et al., 2022). From there, we retrive pdf and html policy documents from the public websites of the original publishing organizations and the United States Government Printing Office, IGOs like, the OECD, the UN, and the WHO and numerous think tanks like the Brookings Institution and the Heritage Foundation. These documents cover a broad range of domains, topics and geographic regions. They provide a rich foundation for identifying scientific publications cited in real-world policy contexts. Leveraging Overton's citation metadata, we identify the scientific papers cited by each policy document. Each citation forms a candidate paper-brief pair, and links a scientific

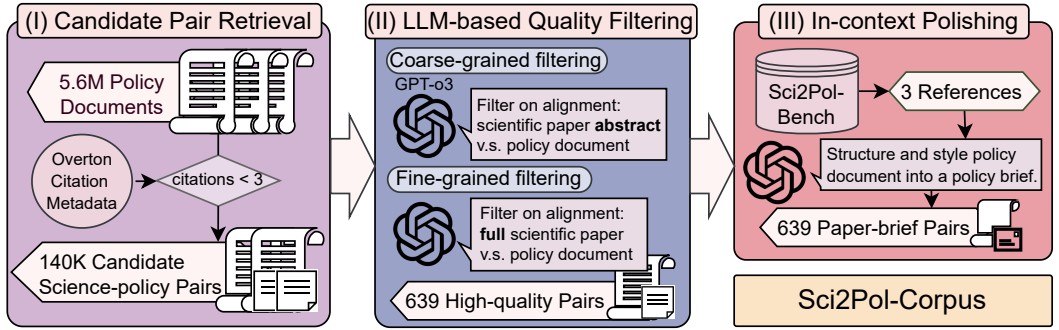

Figure 2: **Overview of the Sci2Pol-Corpus Curation Process.** It consists of 639 paper–brief pairs. Pair retrieval relies on the original policy documents and the scientific papers they cite as candidates.

publication to a policy document that references it. To prioritize relevance, we apply a heuristic: The fewer scientific papers a policy document cites, the more likely it is to focus on each one. This assumption increases the likelihood that the policy document reflects or interprets the cited scientific content. Based on this insight, we retain only policy documents that cite no more than three scientific papers. This yields a high-quality pool of 140,000 candidate pairs for further filtering.

## 3.2 LLM-BASED QUALITY FILTERING

In this stage, we employ GPT-o3 for automated quality filtering to assess whether the policy document centers on the scientific content. It extracts 639 high-quality pairs from 140,000 candidate pairs with two steps: (i) Coarse-grained filtering based on the alignment between the scientific paper abstract and the policy document, and (ii) Fine-grained filtering based on the alignment between the full scientific paper and the policy document. The rationale for this design is to reduce the filtering cost.

**Coarse-grained Filtering.** Scientific papers are often long, typically over 10 pages, and sometimes more than 30. Assuming 500 words per page, this corresponds to about 5,000 to 15,000 words per paper. Assuming each word maps to one token in the GPT-o3 embedding layer, the total token count for 140,000 papers exceeds 700 million. At a rate of $2 per million tokens, the cost of processing full texts becomes very high. To reduce cost while preserving essential information, we use only the paper abstracts in this filtering stage. We extract these abstracts from SciSciNet (Lin et al., 2023). If a paper is not found in SciSciNet, we discard the corresponding candidate pair. Given a scientific paper abstract and its associated policy document, we prompt GPT-o3 to assess whether the policy document centers on the scientific content described in the abstract. See Table 75 for the detailed prompt of GPT-o3. As a result, we obtain 1,407 potential high-quality pairs from this step.

After the GPT-o3-based filtering, we observe that some policy documents are too long and exceed 10 pages. For standard policy briefs, we prefer shorter documents, typically fewer than 10 pages. Among the 1,407 pairs, 777 pairs contain policy documents under 10 pages, while 630 pairs involve longer documents. To make use of the 630 longer policy documents, we manually extract their executive summaries when available and treat these as the pseudo-policy briefs. For each such case, we constructed two pairs: (i) the executive summary paired with the corresponding scientific paper, and (ii) the executive summary paired with the remaining portion of the policy document. In the second case, the remaining text serves as a pseudo-scientific paper, since it often delivers science-related technical details. If a long policy document did not contain a summary, we discarded it. After this process, we retain 234 usable pairs from the long policy documents. Combined with the 777 pairs involving policy documents under 10 pages, this yields a total of 1,011 curated pairs.

**Fine-grained Filtering.** We use GPT-o3 with the full scientific paper and the policy document for fine-grained filtering. The main goal is still to verify whether the policy document centers on the scientific content. However, some pairs originate from the same long policy document, where the executive summary serves as a policy brief and the remaining content acts as a proxy for the scientific paper. We need to avoid these pairs if the two texts are too similar. To handle this, we add a new criterion that measures similarity between the paper and the policy document. This step goes beyond the original metrics used in coarse-grained filtering. See Table 76 for the detailed GPT-o3 prompt.

Table 2: **Performance of LLMs on Sci2Pol-Bench.** We report average scores for 13 LLMs across the five categories of Sci2Pol-Taxonomy. Tasks 1-6 and 16-18 use Micro F1. Tasks 7-10 use a reference-free score, while Tasks 11-15 use a reference-based score. Both of these are judged on Gemini-2.5-Pro.

| Model | Sci2Pol-Taxonomy | | | | | Avg. | Rank |
|---|---|---|---|---|---|---|---|
| | Auto. (1-4) | Under. (5-6) | Sum. (7-10) | Gene. (11-15) | Ver. (16-18) | | |
| **Grok-3-beta** | 50.77±2.89 | 80.12±1.22 | **83.26**±0.05 | **86.70**±0.98 | 85.45±0.86 | **77.01**±1.20 | 1 |
| **DeepSeek-R1** | 44.76±3.11 | 86.61±1.04 | 80.83±0.04 | 84.75±1.26 | 83.84±1.05 | 75.05±1.34 | 2 |
| **Qwen3-235B** | 47.22±3.03 | **87.19**±0.94 | 77.02±0.15 | 84.80±1.30 | 83.76±0.99 | 74.81±1.34 | 3 |
| **DeepSeek-V3** | 39.54±3.06 | 79.35±1.28 | 78.97±0.05 | 86.23±1.26 | 85.48±0.85 | 73.35±1.33 | 4 |
| **GPT-4o** | 52.17±3.00 | 77.17±1.32 | 74.23±0.06 | 76.39±1.28 | 85.45±0.82 | 72.12±1.32 | 5 |
| **Gemma-3-27B** | 43.60±2.83 | 67.82±1.42 | 74.55±0.05 | 84.82±1.16 | 84.29±0.98 | 71.40±1.28 | 6 |
| **Claude-3.7-Sonnet** | 44.06±3.00 | 80.06±1.19 | 82.71±0.05 | 73.59±3.61 | 83.24±1.04 | 71.38±1.99 | 7 |
| **Mistral-Large** | 44.09±2.92 | 76.27±1.23 | 78.57±0.05 | 75.09±1.42 | 81.87±1.11 | 70.23±1.38 | 8 |
| **LLaMA-3.3-70B-IT** | **53.16**±2.72 | 74.14±1.38 | 71.22±0.06 | 69.89±1.62 | **85.71**±0.87 | 69.58±1.37 | 9 |
| **LLaMA-4-Maverick** | 38.74±2.90 | 83.81±1.01 | 72.47±0.06 | 74.95±1.38 | 84.16±0.95 | 68.87±1.31 | 10 |
| **Qwen3-8B** | 35.15±2.88 | 80.84±1.21 | 74.08±0.17 | 77.79±1.49 | 81.87±1.01 | 68.51±1.39 | 11 |
| **Gemma-3-12B** | 42.96±2.79 | 69.61±1.28 | 71.79±0.05 | 77.34±1.44 | 82.51±1.06 | 68.47±1.35 | 12 |
| **LLaMA-3.1-8B-IT** | 27.12±2.53 | 47.74±1.54 | 64.42±0.05 | 65.78±1.71 | 76.25±1.27 | 56.63±1.43 | 13 |

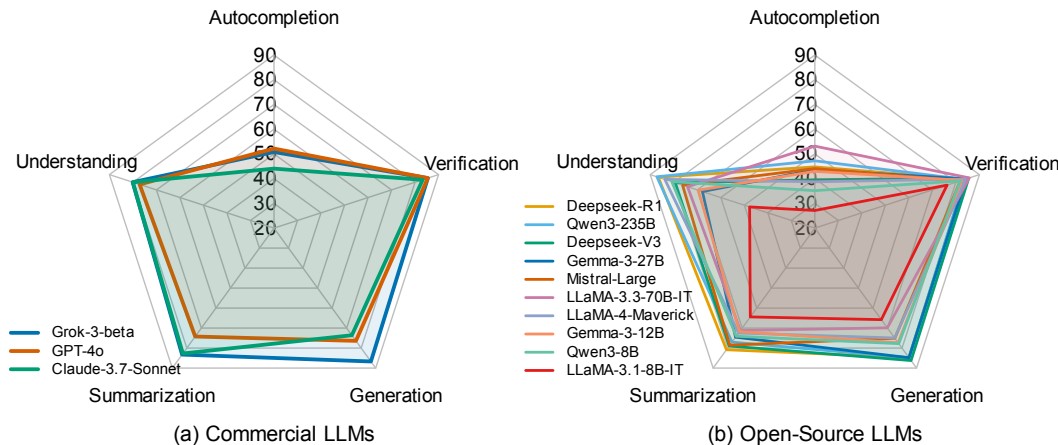

Figure 3: **Visualization of Performance of 13 LLMs on Sci2Pol Bench.** We show the average performance of commercial and open source LLMs across the five categories of the Sci2Pol-Taxonomy.

## 3.3 IN-CONTEXT POLISHING

The last step polishes the policy document. Our policy documents come from official organizations and focus on policy, not briefs. Their format and style do not match a standard policy brief. We propose in-context polishing to align them. We select three expert-written paper-brief pairs from the 85 pairs in Sci2Pol-Bench as references. We then provide the scientific paper and the policy document to GPT-o3 and ask it to revise the document into a standard policy brief while preserving facts and citations. See Table 77 for the detailed prompt. This step does not inject scientific or political content from the three expert-written paper-brief pairs. It only follows their writing style and format. To further validate that this step does not introduce information leakage from Sci2Pol-Bench, we conduct additional experiments, with details provided in Appendix H.9.

## 4 EXPERIMENTAL STUDIES

In this section, we present the performance of LLMs on Sci2Pol-Bench (Section 4.1) and demonstrate the performance gains achieved through supervised fine-tuning on Sci2Pol-Corpus (Section 4.2).

### 4.1 LLMs PERFORMANCE ON SCI2POL-BENCH

We evaluate 13 models, including both commercial and open-source models: (1) ChatGPT-4o, (2) Claude-3.7-Sonnet, (3) Gemma-3-27B, (4) Gemma-3-12b, (5) Grok-3-beta, (6) DeepSeek-R1, (7) DeepSeek-V3, (8) Qwen3-235B-A22B, (9) Qwen3-8B, (10) Mistral-Large (11) LLaMA-4-Maverick-

Table 3: **Performance of LLMs after Supervised Fine-tuning (SFT) on Sci2Pol-Corpus.** We fine-tune three models: LLaMA-3.1-8B-Instruct, Gemma-3-12B, and Gemma-3-27B on Sci2Pol-Corpus, and report their average performance across the Sci2Pol-Taxonomy before and after fine-tuning.

| Model | Sci2Pol-Taxonomy | | | | | Avg. | Gain |
|---|---|---|---|---|---|---|---|
| | Auto. (1-4) | Under. (5-6) | Sum. (7-10) | Gene. (11-15) | Ver. (16-18) | | |
| **LLaMA-3.1-8B-IT** | 27.12±2.53 | 47.74±1.54 | 64.42±0.05 | 65.78±1.71 | 76.25±1.27 | 56.63±1.43 | - |
| **LLaMA-3.1-8B-SFT** | 31.27±2.90 | 44.34±1.48 | 78.28±1.25 | 77.62±1.55 | 80.59±1.08 | 64.27±1.70 | **+7.64** |
| **Gemma-3-12B** | 42.96±2.79 | 69.61±1.28 | 71.79±0.05 | 77.34±1.44 | 82.51±1.06 | 68.47±1.35 | - |
| **Gemma-3-12B-SFT** | 43.14±2.86 | 69.53±1.25 | 84.19±1.19 | 78.57±1.53 | 82.48±1.05 | 71.59±1.64 | **+3.12** |
| **Gemma-3-27B** | 43.60±2.83 | 67.82±1.42 | 74.55±0.05 | 84.82±1.16 | 84.29±0.98 | 71.40±1.28 | - |
| **Gemma-3-27B-SFT** | 45.39±2.90 | 67.44±1.36 | 86.36±1.06 | 81.53±1.60 | 84.06±0.96 | **73.43±1.64** | **+2.03** |
| **DeepSeek-V3** | 39.54±3.06 | 79.35±1.28 | 78.97±0.05 | 86.23±1.19 | 85.48±0.85 | 73.35±1.33 | - |
| **GPT-4o** | 52.17±3.00 | 77.17±1.32 | 74.23±0.06 | 76.39±1.28 | 85.45±0.82 | 72.12±1.32 | - |

17B-128E, (12) LLaMA-3.3-70B-Instruct, and (13) LLaMA-3.1-8B-Instruct. For each setting, we conduct 1,000-iteration bootstrap significance tests (seed = 42) and report the mean and standard deviation. We summarize average performance scores across the Sci2Pol-Taxonomy in Table 2 and visualize the performance of open-source and commercial models in Figure 3. Appendix G.2 provides full results for all 18 tasks, and see Appendix G.1 for the detailed experimental settings.

We include further analysis in Appendix H, covering: (1) human baseline (Appendix H.2); (2) common failure modes (Appendix H.3); (3) what SFT learns (Appendix H.4); (4) the impact of prompt length (Appendix H.5); (5) the reliability of the Gemini-based judge (Appendix H.6); (6) over-endorse analysis in Tasks 16 and 18 (Appendix H.8); (7) information leakage testing of in-context polishing (Appendix H.9); (8) potential circularity in benchmark construction (Appendix H.10); (9) saturation analysis (Appendix H.11); (10) trade-offs in generating briefs from abstracts, introductions, or full papers (Appendix H.12).

## 4.2 SUPERVISED FINE-TUNING ON SCI2POL-CORPUS

We validate the effectiveness of Sci2Pol-Corpus by fine-tuning three models: (1) LLaMA-3.1-8B-Instruct, (2) Gemma-3-12B, and (3) Gemma-3-27B. Table 3 shows the results. All fine-tuned models show consistent improvements across Sci2Pol-Bench. Notably, fine-tuned Gemma-3-27B outperforms both GPT-4o and DeepSeek-V3 (671B), despite their significantly larger scale or broader capabilities. These findings highlight the value of domain-specific supervision in Sci2Pol-Corpus. It indeed captures policy-relevant reasoning. See Appendix G.1 for the experimental details. The cross-domain generality of our SFT methodology is further shown in Appendix I.

## 5 CONCLUSION

We present Sci2Pol-Bench and Sci2Pol-Corpus, the first benchmark and training dataset designed specifically for evaluating and improving large language models on scientific-to-policy brief generation. Grounded in a five-stage taxonomy that mirrors the human writing process (Autocompletion, Understanding, Summarization, Generation, and Verification), Sci2Pol-Bench offers a fine-grained evaluation framework spanning 18 tasks. Our results show that even state-of-the-art LLMs struggle with factual grounding, actionability, and policy-appropriate reasoning, and that commonly used metrics such as ROUGE and BERTScore fail to capture brief-writing quality.

To address these limitations, we curate Sci2Pol-Corpus from 5.6 million policy documents, producing 639 high-quality paper–brief pairs through citation-based retrieval, LLM-assisted filtering, and expert-guided in-context polishing. Fine-tuning on this corpus yields consistent gains across Sci2Pol-Bench. Notably, a fine-tuned Gemma-27B model surpasses much larger commercial systems, demonstrating that targeted, domain-specific supervision can outweigh scale for scientific-to-policy translation.

Together, these resources establish foundational infrastructure for studying and improving how language models translate scientific evidence into actionable policy guidance. We hope Sci2Pol-Bench and Sci2Pol-Corpus will catalyze future research toward models that not only understand science but can responsibly and effectively inform policy in high-stakes societal domains. We also provide detailed LLM usage in Appendix A, dataset links in Appendix B, limitations, future work, and broader impacts in Appendix D, and related works in Appendix E.

## REPRODUCIBILITY STATEMENT

We have made efforts to ensure the reproducibility of our work. The full details of our benchmark construction, dataset curation, and model evaluation are provided in the main text and appendices. For all 18 tasks in Sci2Pol-Bench, we describe the data sources, annotation protocols, and evaluation metrics in detail (Section 2). The construction steps of Sci2Pol-Corpus, including citation-based retrieval, LLM-based filtering, and in-context polishing, are documented in Section 3. We release all prompt templates and scoring rubrics in Appendix L. To facilitate replication, we include the code in the supplementary materials. For each supervised fine-tuning experiment, we specify training configurations and hardware setup in Appendix G.1. Collectively, these resources ensure that our results can be verified and extended by the community.

## ETHIC STATEMENT

**Human-in-the-loop Annotation Process.** All reviewers involved in task development and verification are postdoctoral researchers or research faculty in political science. They are co-authors of this paper and serve as domain experts throughout the benchmark and dataset construction. No monetary compensation is provided; their participation is motivated by scholarly collaboration and a shared commitment to advancing evidence-based policymaking.

**Details of Human Involvement in Label Generation.** Three tasks (5, 16, and 18) use GPT-o3 to propose initial labeled items, and all of them undergo full human verification. Each task is reviewed independently by two reviewers, with 100% reviewer agreement across all accepted items. For all 1,200 samples in Task 5, no corrections are required. For Task 16, reviewers examine all 850 samples and correct three mislabeled cases (correction rate $\approx 0.35\%$). For Task 18, reviewers validate all 706 generated contradictory implications, and no corrections are needed.

**Copyright and Use of Scientific Papers.** The benchmark and dataset are built from publicly accessible content such as article metadata. We release only article DOIs, metadata, access dates, policy brief metadata, and our annotations. The benchmark and dataset are released under CC-BY-NC 4.0, with a two-tier protocol (open metadata + controlled transformed excerpts), and no publisher-owned text is redistributed. Derived tasks (e.g., summarization in Sci2Pol-Bench) rely solely on reformulated excerpts for non-commercial academic purposes.

We also document potential impacts and implement explicit safeguards: all tasks require evidence-based summarization (not prescriptive advice), prompts prohibit unsupported claims, and evaluation judges penalize fabricated, speculative, or over-confident policy recommendations. To reduce evaluation variance and bias, we employ rubric-anchored scoring, multi-model judging, and expert-labeled calibration samples. These measures establish clear licensing boundaries, release protocols, bias-mitigation procedures, and risk-assessment practices appropriate for policy-oriented datasets.

For transparency, we provide a detailed list of all used paper–brief pairs in Appendix J.1, including DOIs, sources, and access dates.

## ACKNOWLEDGMENTS

WW would like to thank Robin Luo, Yichi Zhang, Qineng Wang, and Kongdi Zheng for enlightening discussions on related topics.

We acknowledge grant support from the National Science Foundation (award No. **2404035**).

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

# Supplementary Material

## A  LLM USAGE ACKNOWLEDGEMENT

Large Language Models (LLMs) were used in this project as general-purpose assistive tools across several stages of data curation, benchmark design, evaluation, and paper writing. We detail their precise contributions below.

- **Manuscript Editing:** We used LLMs (e.g., **GPT-4o**) to improve the clarity, grammar, and fluency of English writing throughout the paper.

- **Sci2Pol-Bench:** (i) Data Augmentation and Annotation: For Understanding, Generation, and Verification tasks, **GPT-o3** was used to generate training and evaluation samples, including sentence classification data, missing brief sections (e.g., Policy Problem and Study Methods for JHSB samples), and supported/contradicted claims for claim verification tasks. (ii) Prompt-Based Evaluation: **Gemini-2.5-Pro** was used as an LLM-as-a-judge to evaluate model outputs for Tasks 7–15 based on clarity, factual accuracy, completeness, and policy relevance. (iii) LLM Evaluation: We benchmarked 13 commercial and open-source LLMs.

- **Sci2Pol-Corpus:** (i) Filtering and Evaluation of Candidate Pairs: We employed **GPT-o3** to identify 639 high-quality science-policy pairs from 140,000 candidates. (ii) In-Context Polishing of Policy Briefs: We used **GPT-o3** with three expert-written examples in an in-context setup to revise raw policy documents into standard, structured policy briefs.

All LLM-generated content used in data or evaluation workflows was curated or reviewed by human experts. We take full responsibility for all content in this paper, including that generated or evaluated by LLMs. No LLM is credited as an author.

## B  DATASET LINKS

We provide the links for the datasets:

- Sci2Pol-Bench Dataset:  `https://huggingface.co/datasets/Weimin2000/Sci2Pol-Bench`

- Sci2Pol-Corpus Dataset:  `https://huggingface.co/datasets/Weimin2000/Sci2Pol-Corpus`

## C    PRELIMINARIES

In this section, we present preliminaries on the policy brief (Appendix C.1) and compare it with the full scientific paper, paper introduction, and paper abstract (Appendix C.2).

### C.1    POLICY BRIEF

The policy brief is first introduced by *Nature Energy* to provide concise, policy-relevant summaries written by the original researchers. Each brief cites its source article using clear metadata (e.g., "based on: title doi"), enabling reliable paper-brief alignment. It includes five parts:

- **Title:** A concise one-line headline that states the policy issue.

- **Policy Problem:** A short paragraph (fewer than five sentences) framing the societal risk.

- **Scientific Research Findings:** One or two compact paragraphs (about 150 words) clearly summarizing the study's core quantitative results and key empirical insights.

- **Scientific Research Study Methods:** A single paragraph (about 100 words) briefly explaining the dataset and modelling approach in clear, accessible, and lay terms.

- **Policy Implications:** Four to six bullet points (about 25 words each), drawn strictly from the paper's authors, highlighting concrete conclusions directly relevant to policymakers. No added recommendations, speculation, or external examples.

### C.2    COMPARISON WITH FULL SCIENTIFIC PAPER, PAPER INTRODUCTION AND ABSTRACT

We provide a comparison between the policy brief and the full paper, introduction, and abstract.

- **Purpose:** A research paper provides full evidence and methods for experts; its introduction frames the knowledge gap and aims; the abstract compresses aims, methods, and key results. A policy brief removes technical detail and distills only policy-relevant insights for decision makers.

- **Structure:** Papers follow IMRaD (Introduction, Methods, Results, Discussion) with references; abstracts compress these into one paragraph. Policy briefs replace IMRaD with Policy Problem, Scientific Research Findings, Scientific Research Study Methods, and Policy Implications.

- **Tone and Jargon:** Papers, introductions, and abstracts use technical language and equations. A policy brief avoids jargon and equations, relying on plain prose accessible to non-experts.

- **Focus:** Full papers emphasize evidence and methodological rigor; introductions stress scholarly significance; abstracts distill what is done and found. Policy briefs highlight why findings matter for policy, and repeat only the author's stated implications.

- **Audience Takeaway:** Researchers consult full papers for replicable detail, introductions for rationale, and abstracts for a quick overview. Policymakers rely on briefs to grasp the problem, key evidence, methodological credibility, and policy implications.

# D LIMITATIONS, FUTURE WORK, AND BROADER IMPACT

We present the limitations and future work (Appendix D.1), and the broader impact (Appendix D.2).

## D.1 LIMITATIONS AND FUTURE WORK

We have the following limitations and related future work:

- Sci2Pol-Bench and Sci2Pol-Corpus remain modest in size. Unlike efforts that rely on synthetic summaries or crowd-sourced approximations, our datasets reflect the communicative intent and expertise of scientists writing for policy audiences. We will continue to incorporate newly published briefs and their corresponding papers to expand coverage.

- Although supervised fine-tuning on Sci2Pol-Corpus yields consistent gains, it does not yet surpass state-of-the-art commercial models, e.g., Grok. Future work should explore improved training strategies and better use of Sci2Pol-Corpus to close this gap.

- The existing 85 published policy briefs come from *Nature Energy*, *Nature Climate Change*, *Nature Cities*, *Nature Sustainability*, and the *Journal of Health and Social Behavior (JHSB)*. Expanding Sci2Pol-Bench and Sci2Pol-Corpus to additional disciplines and languages will enable broader generalization as new paper-brief pairs appear.

- 20 samples in Tasks 11 and 13 are drawn from the *Journal of Health and Social Behavior (JHSB)*. For these cases, the *Policy Problem* and *Study Methods* sections are revised by GPT-o3 based on the *Research Problem & Data* section. This step may not be as reliable as for the remaining 65 samples. However, our experts review them, and we chose to retain these cases because they account for more than 20% of the total sample set.

- While Sci2Pol-Bench is intentionally designed around single-paper policy briefs, this focus also imposes an inherent limitation. Our benchmark does not capture the broader genre of multi-source policy synthesis, where policymakers integrate evidence from many studies. As a result, our benchmark does not assess, nor does our dataset enhance, a model's capacity to aggregate heterogeneous sources or integrate multiple lines of evidence.

## D.2 BROADER IMPACT

This work advances understanding of the critical connection between scientific research and policy-making. Science provides evidence and authoritative knowledge essential for informed decisions and sustaining public trust. Its role grows as pressing challenges such as climate change, public health crises, and rapid technological change demand the timely integration of new findings.

Sci2Pol-Bench directly addresses a persistent bottleneck in this pipeline: policymakers' difficulty in translating dense, nuanced scientific evidence into actionable guidance. By decomposing the brief writing process into five stages and providing 18 fine-grained tasks, it offers the first systematic evaluation framework tailored to scientific-to-policy communication. This benchmark exposes weaknesses in even frontier language models and creates opportunities to build models that are not only accurate but also clear, consistent, and persuasive for policy audiences.

Sci2Pol-Corpus complements this evaluation framework by supplying the first domain-specific training dataset for policy brief generation. Curated from millions of policy documents and refined through quality filtering and in-context revision, it provides 639 high-quality papers to brief pairs. Fine-tuning on this corpus leads to consistent gains across Sci2Pol-Bench, even enabling smaller open source models to surpass frontier-scale models. This demonstrates the importance of targeted supervision for capturing the communicative intent of scientific-to-policy translation.

Together, these resources have a significant and lasting societal impact worldwide. They empower scientists, policymakers, and institutional leaders with practical tools to measure and improve the policy relevance of scientific communication. They strengthen accountability and reduce serious risks of misinformation by encouraging accurate, structured, and transparent communication of evidence. They also highlight the vital and growing role of science in shaping societal outcomes and help ensure that policy decisions remain grounded in rigorous research rather than speculation.

At a time of political polarization, resource constraints, and global competition, the ability to generate clear, accurate, and actionable policy briefs is vital. Sci2Pol-Bench and Sci2Pol-Corpus provide foundational infrastructure for developing models that meet this challenge, supporting evidence-based decision-making and enhancing public trust in science and governance.

# E  RELATED WORKS

We review related work on benchmarks and datasets in the scientific and political domains.

## E.1  SCIENTIFIC AND POLITICAL BENCHMARKS

In recent years, LLMs (Team et al., 2024; Liu et al., 2024a; Touvron et al., 2023; Achiam et al., 2023; Bai et al., 2023) have attracted significant attention due to their impressive performance. There have been several benchmarks to evaluate their performance in scientific and political domains.

**Scientific Benchmarks.**  In the scientific domain, benchmarks such as SciRIFF (Wadden et al., 2024), MMLU-Pro (Wang et al., 2024), SciInstruct (Zhang et al., 2024a), SciLitLLM (Li et al., 2024c), and SciRepEval (Singh et al., 2023) test models on tasks like summarization, question answering, and claim verification. These benchmarks focus on instruction-following and comprehension of scientific content. Most use single-step tasks with scientific inputs, typically full papers or extended passages.

**Political Benchmarks.**  In the political domain, benchmarks assess how models reason about political science, ideology, and value alignment. Political Science QA (Li et al., 2024b) tests factual knowledge and reasoning. Röttger et al. (2024) and Ren et al. (2024) probe value orientation and political opinions. Motoki et al. (2024) measure political bias. These tasks often use short prompts under 300 tokens and rely on multiple-choice formats. Political-LLM (Li et al., 2024b) contributes a useful taxonomy and discussion of use cases, but offers no benchmark or annotated data.

However, existing scientific and political benchmarks overlook the dual challenge of science-informed policy communication: understanding complex research and translating it into actionable language. SciRIFF and SciInstruct focus on instruction-following for scientific tasks, while MMLU-Pro tests expert-level reasoning with multiple-choice questions. Political benchmarks such as ValueBench or Political-LLM probe ideology and values but do not address policy generation or scientific grounding. To fill this gap, Sci2Pol-Bench evaluates LLMs on generating policy briefs from full-length scientific papers, pairing them with expert-written briefs and introducing a five-stage pipeline. We compare against three representative efforts. SciRIFF and MMLU-Pro are the closest scientific benchmarks: SciRIFF covers single-step tasks like summarization, QA, and claim verification, while MMLU-Pro measures reasoning breadth but not policy translation. Political-LLM provides the first principled framework for computational political science, offering a taxonomy but no annotated tasks. Together, these works form the strongest prior art in scientific comprehension and political reasoning. Sci2Pol-Bench builds on them but uniquely combines scientific fidelity with direct policy relevance, making it the first benchmark to evaluate LLMs on producing accurate, actionable policy briefs.

## E.2  SCIENTIFIC AND POLITICAL DATASETS

Beyond evaluation benchmarks, several open-source fine-tuning datasets adapt LLMs to scientific and political domains through supervised instruction tuning or continued pre-training.

**Scientific Datasets.**  Researchers have curated domain-specific corpora to enhance scientific reasoning. SciInstruct (Zhang et al., 2024a) provides curated instructions across physics, chemistry, math, and formal proofs, improving models' performance on college-level problems. SciLitLLM (Li et al., 2024c) combines continual pre-training on research papers with supervised tuning. It introduces SciLitIns, a collection of instructions targeting underrepresented fields. Large-scale corpora have also been used. For example, Galactica (Taylor et al., 2022) is trained on 48 million science documents. This shows how domain-specific training endows models with broad scientific knowledge. Together, these datasets demonstrate the effectiveness of specialized fine-tuning.

**Political Datasets.**  In the political domain, fine-tuning datasets emphasizes factual knowledge, ideology, explicit opinions, and bias alignment. PoliTune (Agiza et al., 2024) creates ideology-specific instruction data from social media, while Vendetti et al. (2025) fine-tune the existing LLMs on curated Reddit discourse to model polarized political commentary. These efforts show how targeted fine-tuning can steer models toward particular political knowledge or stances.

However, existing datasets in both domains overlook the dual challenge of science-informed policy communication. Scientific datasets like SciInstruct and SciLitIns strengthen technical reasoning but exclude crucial policy text, while political datasets such as PoliTune emphasize ideological alignment yet ignore underlying scientific evidence. Sci2Pol-Corpus fills this important gap by linking full-length research papers to expert-written policy briefs. It trains advanced models to extract key findings and translate them into clear, actionable recommendations for diverse policymakers.

## F  How LLMs Fail in Policy Brief Generation

In this section, we present representative examples that highlight the limitations of existing LLMs in scientific-to-policy brief generation. Using the prompt in Table 59, we instruct different LLMs to generate policy briefs from the scientific papers listed in Appendix J, and then compare their outputs with the expert-written published briefs. The results reveal issues in four key areas: contextual depth, hallucination risk, readability and tone, and actionability. We provide one illustrative example for each issue in Appendix F.1, Appendix F.2, Appendix F.3, and Appendix F.4.

### F.1  Contextual Depth

Table 4 is an example where LLM output contains a contextual-depth issue. The issue is evident because the published brief gives precise quantitative findings such as "an additional 0.5 GtCO$_2$/yr by 2030 and 1.9 GtCO$_2$/yr by 2050" and compares these against Paris Agreement scenarios to highlight the "CDR gap". However, the LLM output only provides a broad summary about reliance on land-based removals and a lack of novel technologies. It does not capture the specific numerical evidence and scenario-based reasoning that are central to the paper's findings.

Table 4: **Expert- vs. LLM-written Policy Brief Excerpts: Contextual Depth Issue.** Paper: *The carbon dioxide removal gap* (Lamb et al., 2024a). LLM: GPT-4o. Presented Brief Section: Scientific Research Findings.

| Expert-written Brief | LLM Output |
|---|---|
| In our study we found that compared with 2020, the most ambitious national proposals for CDR imply an additional 0.5 GtCO2yr-1 of removals by 2030, and 1.9 GtCO2yr-1 by 2050. Compared with CDR scaling in Paris Agreement-consistent scenarios, we found that these national CDR proposals tend to fall short by hundreds of megatonnes of carbon dioxide in 2030 to several gigatonnes of carbon dioxide in 2050, highlighting a 'CDR gap'. However, we find that the most ambitious proposals do come close to levels in a low-energy-demand scenario where CDR requirements are minimized, suggesting that if countries pledge more ambitious emissions reductions consistent with these scenarios, the CDR gap will be closed. As levels of reporting vary, our evaluation of proposed CDR does assume that a number of countries simply maintain their current levels of (conventional) removals. In addition, it remains unknown to what extent firm CDR policies will follow these proposals. | Research indicates that current national proposals for CDR are not aligned with integrated assessment model scenarios, which project the levels necessary to achieve the 1.5°C temperature limit. The study shows that while afforestation and land-based removal methods are in use, there is a lack of commitment to scaling novel CDR technologies like bioenergy carbon capture and storage (BECCS) or direct air carbon capture and storage (DACCS). Most proposed CDR efforts remain heavily reliant on land-based removals, which are not sufficient on their own to meet the required targets. |

## F.2 HALLUCINATION RISK

Table 5 shows a risk of hallucination. In this case, the LLM reads the paper as endorsing public or cooperative ownership, progressive tax swaps, cross-border subsidies, and profit-cap rules. The paper does not state these. The human brief stays close to the source and lists the core points: trillion-dollar scale, windfall profits under unregulated markets, trade-offs from profit caps, regional distribution effects, and the near-term need to scale NETs. The LLM adds new policies and causal claims, so it may mislead readers.

Table 5: **Expert- vs. LLM-written Policy Brief Excerpts: Hallucination Issue.** Paper: *Inequality repercussions of financing negative emissions* (Andreoni et al., 2024b). LLM: Qwen3-8B. Presented Brief Section: Policy Implication.

| Expert-written Brief | LLM Output |
|---|---|
| <ul><li>In a net-zero emissions world, NETs could become a trillion-dollar business globally.</li><li>If financed through an unregulated carbon market, the owners of these companies would enjoy windfall profits, potentially leading to a large increase in economic inequality.</li><li>Market regulation, such as profit caps, could reduce the inequality increase, but at the risk of stimulating too much or too little carbon removal - policymakers should be aware of this trade-off.</li><li>Concentrating removal efforts in the Global North or transferring resources to the Global South could, to some extent, offset the increase in inequality at the global level.</li><li>These dynamics mostly apply to a net-zero and post-net-zero world. The current priority of policymakers should remain to provide adequate resources to scale up NETs towards technology maturity.</li></ul> | 1. Equitable Ownership Structures: Policymakers should consider public ownership or cooperative frameworks for NETs to prevent disproportionate benefits to private capital holders. This could mitigate the ownership effect and ensure that profits from carbon removal are distributed more fairly.
2. Progressive Taxation and Redistribution: Replacing regressive carbon taxes with progressive income taxes after net-zero, or using redistribution mechanisms, could counteract inequality risks. Pre-net-zero taxation reforms might also reduce adverse impacts, though they could compromise economic growth.
3. International Financial Transfers: Implementing cross-border carbon removal credits or subsidies for low-income countries could address global inequality. However, these transfers may still benefit high-income groups within recipient nations, necessitating targeted redistribution policies.
4. Regulating Profit Margins: Introducing profit caps or alternative pricing schemes for NETs within a single carbon market could balance incentivizing deployment with equity. Yet, such measures must avoid distorting markets or discouraging investment.
5. Integrated Policy Design: Climate strategies must explicitly account for distributional impacts in policy frameworks. This includes designing carbon pricing, ownership rules, and revenue allocation mechanisms to align with equity goals while maintaining the viability of NETs as part of a low-carbon transition. |

## F.3 READABILITY AND TONE

Table 6 is an example where the LLM output has readability and tone issues. The expert-written brief is logically ordered, uses active voice, and speaks to policymakers. It defines the study design, maps three strategies to plain descriptions, and explains how the team built and compared scenarios. By contrast, the LLM output contains unexplained jargon in bullet points. It also introduces acronyms without setup, which raises the barrier for non-expert readers.

Table 6: **Expert- vs. LLM-written Policy Brief Excerpts: Readability and Tone Issue.** Paper: *Demand-side strategies enable rapid and deep cuts in buildings and transport emissions to 2050* (van Heerden et al., 2025a). LLM: DeepSeek-V3. Presented Brief Section: Scientific Research Study Methods.

| Expert-written Brief | LLM Output |
|---|---|
| Our study uses quantitative scenarios generated with global integrated assessment models (IAMs) that are specifically designed to capture the interactions within and between the energy system, the economy, and the environment. We developed three scenarios that each capture the impact of a specific strategy targeting energy use. In the first scenario end users reduce or change energy-using activities, in the second they adopt more efficient technologies, and in the third they switch to electrification or alternative energy carriers as substitutes to fossil fuels. Using a structured questionnaire, we also gathered inputs from policymakers and experts in demand sectors to refine the scenarios. We quantitatively interpreted and quantified the scenarios through 2050 using seven IAMs. Finally, we compared the resulting emission reductions per strategy across the set of models and scenarios and analysed the results to better understand the main factors driving these reductions. | The study used an ensemble of seven global integrated assessment models (IAMs) to simulate three intervention strategies: 1. Activity-focused (ACT): Policies promoting shared mobility, reduced travel, and smaller living spaces. 2. Technology-optimizing (TEC): Enhanced building insulation, vehicle efficiency standards, and retrofits. 3. Electrification-focused (ELE): Widespread adoption of heat pumps, electric vehicles, and renewable energy. Scenarios were compared against a "current policies" baseline and a 1.5°C-aligned pathway, accounting for regional variations and model uncertainties. |

## F.4 ACTIONABILITY

Table 7 shows weak actionability. The LLM states broad aims: "reconsider approaches", "build public support", and "prioritize durable reforms". It does not say who does what, when, or how to track success. The expert-written brief gives steps and targets: start with the most damaging fuels, especially coal; invest in transit and support EVs; use fuel-efficiency rules; and choose measures that survive political shocks. The LLM repeats goals but omits scope, sequence, and levers, so a policymaker cannot act.

Table 7: **Expert- vs. LLM-written Policy Brief Excerpts: Actionability Issue.** Paper: *Fossil fuel subsidy reforms have become more fragile* (Mahdavi et al., 2025a). LLM: Gemma3-12B. Presented Brief Section: Policy Implication.

| Expert-written Brief | LLM Output |
| --- | --- |
| <ul><li>While fossil fuel subsidies are harmful, efforts to remove them fail more than 90% of the time. Low-cost fuel is popular and hard to change.</li><li>Reforms are only beneficial if they endure. While price reforms are fragile, other kinds of policies — such as investments in clean transportation and support for electric vehicles — are more durable.</li><li>Since subsidies are politically difficult to remove, governments should concentrate their efforts on the most damaging types of fuel, especially coal.</li><li>Governments should look for alternative ways to reduce the demand for subsidized fuel, for example, by using regulatory measures to enhance fuel efficiency, investing in public transit and promoting electric vehicles.</li></ul> | The findings suggest that current strategies for reducing fossil fuel subsidies are failing. Policymakers should reconsider their approaches, prioritizing:<ul><li>Long-Term Strategies: Focus on policies that reduce demand for subsidized fuel (e.g., fuel efficiency standards, electric vehicle incentives, improved public transit) alongside, rather than in place of, subsidy reductions.</li><li>Political Feasibility: Recognize the political challenges of subsidy reform and build public support through clear communication, targeted compensation for vulnerable populations, and a gradual, phased approach.</li><li>Durable Reforms: Prioritize reforms like shifting to floating prices that are not easily eroded by inflationary pressures or exchange rate fluctuations, even if initially less popular.</li></ul> |

# G    EXPERIMENTAL DETAILS

We provide: (i) the detailed experimental settings for LLM evaluation on Sci2Pol-Bench and supervised fine-tuning (Appendix G.1); (ii) the performance of 13 LLMs across all Sci2Pol-Bench tasks (Appendix G.2); and (iii) the supervised fine-tuning results of three LLMs (Appendix G.3).

## G.1    DETAILED EXPERIMENTAL SETTINGS

We present the detailed experimental settings.

**Evaluate LLMs on Sci2Pol-Bench.**  We evaluate 13 LLMs through API calls. For Grok, GPT-4o, and Claude, we use their respective private APIs. For all other models, we rely on the Novita API.

**Supervised Fine-tuning.**  Our supervised fine-tuning uses LLaMA-Factory (Zheng et al., 2024). We apply lightweight Low-rank Adaptation (LoRA) (Shen et al., 2022) under the following setup. The base models are LLaMA-3.1-8B-Instruct, Gemma-3-12B-IT, and Gemma-3-27B-IT. LoRA uses rank 8, alpha 32, and dropout 0.05, applied to the query and value projection matrices of all attention layers. Training runs for six epochs with a batch size of 8, a cosine learning rate schedule, a peak learning rate of 1e-4, and a 5% warmup phase. FlashAttention-2 (Dao, 2024) and the Liger kernel (Hsu et al., 2025) are enabled for efficient long-context training. We split Sci2Pol-Corpus into 95% training and 5% validation, and select the checkpoint with the lowest validation loss. We run all experiments on 4 NVIDIA A100 80GB Tensor Core GPUs.

### G.2 LLM Performance on Sci2Pol-Bench

We show the detailed performance of 13 LLMs across all tasks in Table 8.

Our experiments lead to three main findings:

- Current LLMs struggle to generate high-quality policy briefs from scientific papers, even though they likely encounter academic content during pretraining. These results reveal a persistent gap between general language capability and task-specific policy reasoning.

- Sci2Pol-Bench exposes new weaknesses not captured by traditional LLM benchmarks. As shown in Table 2 and Table 8, tasks in Autocompletion and Understanding produce the widest F1 range across models, highlighting instability in predicting coherent and grounded continuations.

- Larger models generally perform better, but size alone doesn't guarantee robustness. For instance, Gemma-3-12B outperforms the larger Gemma-3-27B on several tasks, e.g., Task 2 (see Table 2 and Table 8). Similarly, commercial models that dominate benchmarks like GPQA (Phan et al., 2025) often underperform on early-stage Sci2Pol tasks such as Task 1 and Task 4.

Table 8: **Detailed Performance of 13 LLMs on Sci2Pol-Bench.**

| Model | Autocompletion | | | | Understanding | | Summarization |
|---|---|---|---|---|---|---|---|
| | T1 (F1) | T2 (F1) | T3 (F1) | T4 (F1) | T5 (F1) | T6 (F1) | T7 (Gem.) |
| Grok-3-beta | 71.05±2.77 | 63.76±2.79 | 33.12±2.92 | 35.15±3.08 | 90.42±0.80 | 69.83±1.64 | **83.45**±0.05 |
| DeepSeek-R1 | 47.92±2.91 | 44.42±2.83 | **40.45**±3.53 | **46.26**±3.18 | 89.83±0.78 | 83.39±1.29 | 82.91±0.04 |
| Qwen3-235B | 53.14±2.95 | 51.75±2.81 | 39.64±3.20 | 44.33±3.16 | 87.81±0.76 | **86.58**±1.11 | 77.48±0.15 |
| DeepSeek-V3 | 41.62±2.90 | 38.39±3.26 | 39.91±3.28 | 38.23±2.79 | 88.99±0.84 | 69.72±1.72 | 80.30±0.05 |
| GPT-4o | **73.74**±2.81 | 63.33±2.82 | 31.59±3.27 | 40.01±3.09 | 90.56±0.80 | 63.77±1.84 | 75.63±0.06 |
| Claude-3.7-Sonnet | 60.13±2.53 | 44.08±3.26 | 34.38±2.93 | 37.67±3.27 | **93.72**±0.73 | 66.41±1.65 | 83.38±0.05 |
| Gemma-3-27B | 64.52±3.11 | 58.80±2.74 | 25.56±2.85 | 25.51±2.63 | 82.07±1.09 | 53.57±1.75 | 77.61±0.05 |
| Mistral-Large | 62.46±2.81 | 55.01±3.01 | 29.64±3.38 | 29.25±2.46 | 89.45±0.82 | 63.08±1.65 | 81.90±0.05 |
| LLaMA-3.3-70B-IT | 72.51±2.35 | **70.99**±2.68 | 32.26±2.72 | 36.89±3.14 | 87.62±0.99 | 60.66±1.76 | 75.15±0.06 |
| LLaMA-4-Maverick | 55.41±3.04 | 43.82±2.78 | 25.27±2.88 | 30.46±2.90 | 86.10±0.82 | 81.52±1.21 | 75.48±0.06 |
| Gemma-3-12B | 64.18±3.02 | 60.11±3.18 | 25.47±2.59 | 22.09±2.39 | 88.19±0.88 | 51.04±1.68 | 73.81±0.05 |
| Qwen3-8B | 33.49±2.86 | 32.46±2.70 | 33.76±3.00 | 40.90±2.95 | 87.24±0.99 | 74.45±1.42 | 75.80±0.17 |
| LLaMA-3.1-8B-IT | 37.53±2.92 | 32.08±2.47 | 16.44±2.41 | 22.44±2.30 | 55.08±1.42 | 40.41±1.66 | 67.38±0.05 |

| Model | Summarization | | | Generation | | | |
|---|---|---|---|---|---|---|---|
| | T8 (Gem.) | T9 (Gem.) | T10 (Gem.) | T11 (Gem.) | T12 (Gem.) | T13 (Gem.) | T14 (Gem.) |
| Grok-3-beta | **86.88**±0.05 | **83.55**±0.05 | 79.15±0.05 | 92.35±0.63 | **79.50**±0.86 | 82.67±1.40 | **89.38**±1.06 |
| DeepSeek-R1 | 86.30±0.04 | 76.73±0.04 | 77.36±0.04 | 92.77±0.74 | 79.10±0.98 | 88.02±1.52 | 84.64±1.44 |
| Qwen3-235B | 80.38±0.15 | 74.63±0.15 | 75.60±0.15 | **93.55**±0.85 | 77.74±1.39 | **90.40**±1.46 | 86.67±1.24 |
| DeepSeek-V3 | 80.95±0.05 | 76.90±0.05 | 77.72±0.05 | 91.50±0.93 | 78.24±1.46 | 90.29±0.99 | 87.62±1.28 |
| GPT-4o | 75.20±0.06 | 71.38±0.06 | 74.70±0.06 | 89.17±0.98 | 72.69±1.05 | 72.81±1.50 | 78.36±1.54 |
| Claude-3.7-Sonnet | 86.38±0.05 | 77.85±0.05 | **83.23**±0.05 | 81.29±3.37 | 66.62±3.34 | 68.14±3.95 | 77.08±3.58 |
| Gemma-3-27B | 75.98±0.05 | 68.57±0.05 | 76.03±0.05 | 91.33±0.87 | 75.23±1.12 | 88.12±1.32 | 88.54±1.24 |
| Mistral-Large | 79.90±0.05 | 74.75±0.05 | 77.72±0.05 | 87.64±0.92 | 73.02±1.06 | 64.10±1.93 | 77.53±1.88 |
| LLaMA-3.3-70B-IT | 71.85±0.06 | 66.15±0.06 | 71.75±0.06 | 81.40±1.44 | 71.55±1.19 | 60.02±2.07 | 70.86±2.01 |
| LLaMA-4-Maverick | 75.48±0.06 | 70.60±0.06 | 68.33±0.06 | 82.37±1.59 | 74.52±0.95 | 62.45±1.74 | 82.53±1.50 |
| Gemma-3-12B | 71.18±0.05 | 65.45±0.05 | 76.71±0.05 | 85.60±1.02 | 70.22±1.15 | 70.69±2.29 | 84.46±1.29 |
| Qwen3-8B | 77.48±0.17 | 72.08±0.17 | 70.94±0.17 | 91.59±1.00 | 72.62±1.17 | 77.50±1.76 | 79.91±1.84 |
| LLaMA-3.1-8B-IT | 67.38±0.05 | 59.03±0.05 | 63.88±0.05 | 82.35±1.55 | 64.62±1.65 | 55.12±1.90 | 67.74±1.89 |

| Model | Generation | Verification | | | Average | Rank |
|---|---|---|---|---|---|---|
| | T15 (Gem.) | T16 (F1) | T17 (F1) | T18 (F1) | | |
| Grok-3-beta | **89.58**±0.94 | 98.60±0.42 | 59.26±1.67 | **98.48**±0.50 | **77.01**±1.20 | 1 |
| DeepSeek-R1 | 79.23±1.63 | 94.55±0.86 | 59.74±1.61 | 97.23±0.69 | 75.05±1.34 | 2 |
| Qwen3-235B | 75.65±1.57 | 95.36±0.73 | 58.51±1.65 | 97.41±0.60 | 74.81±1.34 | 3 |
| DeepSeek-V3 | 83.51±1.65 | 98.97±0.37 | 59.52±1.61 | 97.94±0.57 | 73.35±1.33 | 4 |
| GPT-4o | 68.93±1.31 | 98.61±0.39 | 60.25±1.47 | 97.49±0.59 | 72.12±1.32 | 5 |
| Gemma-3-27B | 80.89±1.26 | 98.25±0.46 | 59.88±1.52 | 94.74±0.95 | 71.40±1.28 | 6 |
| Claude-3.7-Sonnet | 74.82±3.82 | 98.35±0.47 | 57.92±1.47 | 93.44±1.17 | 71.38±1.99 | 7 |
| Mistral-Large | 73.15±1.31 | 95.57±0.66 | 59.22±1.56 | 90.83±1.12 | 70.23±1.38 | 8 |
| LLaMA-3.3-70B-IT | 65.60±1.41 | **99.07**±0.32 | **61.10**±1.59 | 96.96±0.69 | 69.58±1.37 | 9 |
| LLaMA-4-Maverick | 72.86±1.11 | 96.94±0.61 | 57.99±1.60 | 97.54±0.65 | 68.87±1.31 | 10 |
| Qwen3-8B | 67.32±1.69 | 92.26±0.78 | 57.51±1.44 | 95.85±0.82 | 68.51±1.39 | 11 |
| Gemma-3-12B | 75.71±1.43 | 95.85±0.59 | 57.43±1.75 | 94.26±0.84 | 68.47±1.35 | 12 |
| LLaMA-3.1-8B-IT | 59.05±1.55 | 88.57±1.08 | 54.40±1.64 | 85.79±1.08 | 56.63±1.43 | 13 |

## G.3 SUPERVISED FINE-TUNING ON SCI2POL-CORPUS

We report the detailed supervised fine-tuning performance of three LLMs in Table 9.

Table 9: **Detailed Performance of LLMs with Supervised Fine-tuning (SFT) on Sci2Pol-Corpus.**

| Model | Autocompletion | | | | Understanding | | Summarization |
|---|---|---|---|---|---|---|---|
| | T1 (F1) | T2 (F1) | T3 (F1) | T4 (F1) | T5 (F1) | T6 (F1) | T7 (Gem.) |
| **LLaMA-3.1-8B-IT** | 37.53±2.92 | 32.08±2.47 | 16.44±2.41 | 22.44±2.30 | 55.08±1.42 | 40.41±1.66 | 67.38±0.05 |
| **LLaMA-3.1-8B-SFT** | 38.82±3.13 | 33.73±3.03 | 27.45±2.80 | 25.10±2.63 | 42.58±1.36 | 46.10±1.59 | 81.25±1.30 |
| **Gemma-3-12B** | 64.18±3.02 | 60.11±3.18 | 25.47±2.59 | 22.09±2.39 | 88.19±0.88 | 51.04±1.68 | 73.81±0.05 |
| **Gemma-3-12B-SFT** | 63.14±2.99 | 56.47±2.97 | 27.45±2.75 | 25.49±2.74 | 88.67±0.88 | 50.40±1.61 | 87.47±1.12 |
| **Gemma-3-27B** | 64.52±3.11 | 58.80±2.74 | 25.56±2.85 | 25.51±2.63 | 82.07±1.09 | 53.57±1.75 | 77.61±0.05 |
| **Gemma-3-27B-SFT** | 69.02±2.98 | 55.29±2.92 | 27.45±2.82 | 29.80±2.86 | 80.08±1.14 | 54.80±1.59 | **89.30**±1.08 |
| **DeepSeek-V3** | 41.62±2.90 | 38.39±3.26 | **39.91**±3.28 | 38.23±2.79 | 88.99±0.84 | **69.72**±1.72 | 80.30±0.05 |
| **GPT-4o** | **73.74**±2.81 | **63.33**±2.82 | 31.59±3.27 | **40.01**±3.09 | **90.56**±0.80 | 63.77±1.84 | 75.63±0.06 |

| Model | Summarization | | | Generation | | | |
|---|---|---|---|---|---|---|---|
| | T8 (Gem.) | T9 (Gem.) | T10 (Gem.) | T11 (Gem.) | T12 (Gem.) | T13 (Gem.) | T14 (Gem.) |
| **LLaMA-3.1-8B-IT** | 67.38±0.05 | 59.03±0.05 | 63.88±0.05 | 82.35±1.55 | 64.62±1.65 | 55.12±1.90 | 67.74±1.89 |
| **LLaMA-3.1-8B-SFT** | 71.17±1.45 | 76.00±1.14 | 84.72±1.10 | 86.66±1.11 | 71.10±1.47 | 74.19±1.83 | 83.41±1.61 |
| **Gemma-3-12B** | 71.18±0.05 | 65.45±0.05 | 76.71±0.05 | 85.60±1.02 | 70.22±1.15 | 70.69±2.29 | 84.46±1.29 |
| **Gemma-3-12B-SFT** | 79.17±1.39 | 81.75±1.22 | **88.38**±1.03 | 89.76±1.00 | 71.93±1.39 | 70.92±2.08 | 84.18±1.52 |
| **Gemma-3-27B** | 75.98±0.05 | 68.57±0.05 | 76.03±0.05 | 91.33±0.87 | 75.23±1.12 | 88.12±1.32 | **88.54**±1.24 |
| **Gemma-3-27B-SFT** | **85.42**±1.15 | **83.40**±1.03 | 87.30±1.00 | **91.76**±0.90 | **78.26**±1.35 | 67.34±2.45 | 86.65±1.53 |
| **DeepSeek-V3** | 80.95±0.05 | 76.90±0.05 | 77.72±0.05 | 91.50±0.93 | 78.24±1.46 | **90.29**±0.99 | 87.62±1.28 |
| **GPT-4o** | 75.20±0.06 | 71.38±0.06 | 74.70±0.06 | 89.17±0.98 | 72.69±1.05 | 72.81±1.50 | 78.36±1.54 |

| Model | Generation | Verification | | | | |
|---|---|---|---|---|---|---|
| | T15 (Gem.) | T16 (F1) | T17 (F1) | T18 (F1) | Average | Gain |
| **LLaMA-3.1-8B-IT** | 59.05±1.55 | 88.57±1.08 | 54.40±1.64 | 85.79±1.08 | 56.63±1.43 | - |
| **LLaMA-3.1-8B-SFT** | 72.76±1.75 | 93.29±0.84 | 53.20±1.59 | 95.29±0.81 | 64.27±1.70 | **+7.64** |
| **Gemma-3-12B** | 75.71±1.43 | 95.85±0.59 | 57.43±1.75 | 94.26±0.84 | 68.47±1.35 | - |
| **Gemma-3-12B-SFT** | 76.06±1.66 | 97.18±0.55 | 57.70±1.56 | 92.57±1.03 | 71.59±1.64 | **+3.12** |
| **Gemma-3-27B** | 80.89±1.26 | 98.25±0.46 | 59.88±1.52 | 94.74±0.95 | 71.40±1.28 | - |
| **Gemma-3-27B-SFT** | **83.65**±1.75 | 97.76±0.51 | 60.00±1.54 | 94.43±0.84 | **73.43**±1.64 | **+2.03** |
| **DeepSeek-V3** | 83.51±1.65 | **98.97**±0.37 | 59.52±1.61 | **97.94**±0.57 | 73.35±1.33 | - |
| **GPT-4o** | 68.93±1.31 | 98.61±0.39 | **60.25**±1.47 | 97.49±0.59 | 72.12±1.32 | - |

# H    ADDITIONAL EXPERIMENTAL ANALYSIS

In this section, we examine the following aspects in Sci2Pol-Bench and Sci2Pol-Corpus. (1) We provide evidence for the limitations of BERTScore and ROUGE when applied to Tasks 11-15 (Appendix H.1). (2) We provide the human baseline for Sci2Pol-Bench (Appendix H.2). (3) We provide the analysis of common failure modes of top LLMs (Appendix H.3). (4) We provide a detailed analysis of what supervised fine-tuning learns (Appendix H.4). (5) We test the impact of prompt length on performance for Tasks 1-4 (Appendix H.5). (6) We assess the reliability of the Gemini-2.5-Pro-based reference-free judge by alignwing with human evaluation (Appendix H.6). (7) We contrast section-by-section versus full-brief generation to explain the need for Tasks 11-14 in addition to Task 15 (Appendix H.7). (8) We analyze whether models show a tendency to over-endorse by studying the distribution of false positives and false negatives in Tasks 16 and 18 (Appendix H.8). (9) We validate that in-context polishing does not introduce information leakage (Appendix H.9). (10) We evaluate potential circularity in benchmark construction by comparing GPT and DeepSeek families on Task 16 (Appendix H.10). (11) We do the saturation analysis (Appendix H.11). (12) We compare brief generation from abstracts, introductions, and full papers to analyze trade-offs in context length (Appendix H.12). Together, these studies clarify the robustness and fairness of our Sci2Pol-Bench and Sci2Pol-Corpus.

## H.1    LIMITATIONS OF BERTSCORE AND ROUGE SCORES FOR TASKS 11-15

We provide evidence for why BERTScore and ROUGE scores fail to evaluate Tasks 11-15.

Consider one example: the LLaMA-3.1-8B-Instruct-generated policy brief for the scientific paper *How Central Banks Address Climate and Transition Risks* (Paper (Shears et al., 2025a); Brief (Shears et al., 2025b)). For BERTScore, we demonstrate that deleting large portions of the candidate brief hardly changes the score against the true brief. Using Table 10, we define "brief completeness" by removing sections of the candidate brief to create 75%, 50%, 25%, and title-only variants. Table 11 then reports the BERTScore trend as completeness falls.

For ROUGE scores, we show that simple grammatical or paraphrasing changes cause the scores to drop sharply, even when meaning is preserved. Table 12 presents example texts, and Table 13 reports ROUGE-1/2/L scores between them. The results reveal low scores despite semantic equivalence, highlighting ROUGE's sensitivity to surface form.

Together, these findings illustrate two failure modes: BERTScore remains high under major deletions, while ROUGE collapses under harmless paraphrases. Consequently, for Tasks 11-15, we rely on task-specific LLM-judge scores, which verify section coverage, reasoning flow, and evidence linkage.

Table 10: **Deriving Completeness Subsets from LLM Outputs.** Sections are removed sequentially in the order: Policy Implications → Methods → Findings → Policy Problem. The table indicates which sections remain at each completeness level. Percentages reflect the proportion of content retained.

| Completeness | Policy Problem | Findings | Methods | Policy Implications |
|---|---|---|---|---|
| **100% (Full)** | ✓ | ✓ | ✓ | ✓ |
| **75%** | ✓ | ✓ | ✓ | ✗ |
| **50%** | ✓ | ✓ | ✗ | ✗ |
| **25%** | ✓ | ✗ | ✗ | ✗ |
| **Title Only** | ✗ | ✗ | ✗ | ✗ |

Table 11: **BERTScore under Progressive Section Deletions from an LLM Brief.** Scores remain high despite missing sections. See Table 10 for the definition of percentage levels.

| Brief Completeness | BERT Precision | BERT Recall | BERT F1 Score |
|---|---|---|---|
| **Full Brief** | 0.8689 | 0.8599 | 0.8644 |
| **75%** | 0.8721 | 0.8612 | 0.8666 |
| **50%** | 0.8829 | 0.8551 | 0.8688 |
| **25%** | 0.8828 | 0.8174 | 0.8489 |
| **Title Only** | 0.8738 | 0.7839 | 0.8264 |

Table 12: **Original vs. Grammar-alternated Policy Implications.** We present two semantically equivalent versions of the same "Policy Implications" section side by side: the left column is the original expert wording, and the right column rewrites sentences by alternating grammar/phrasing only (no change in meaning). This pairing is used to evaluate whether shallow changes alone depress ROUGE scores.

| Original Text | Grammar-alternated (Rephrased) Text |
| --- | --- |
| **Policy Implications** | **Policy Implications** |
| • Central banks vary substantially in the extent to which they re-risk stranded asset and physical climate risks and de-risk clean energy investments. | • Substantial variation exists across central banks in how they re-risk stranded asset and physical climate risks while de-risking clean energy investments. |
| • Central bank actions on climate risks are positively associated with their country's climate policy stringency and public concern with climate change and less with its underlying economic risks. | • Their actions on climate risks are linked more strongly with national climate policy stringency and public concern over climate change, and less strongly with economic fundamentals. |
| • Despite their autonomy, central banks do not substitute for the lack of national climate policy but complement existing national policies promoting the clean energy transition. | • National climate policy is not replaced by central banks, even with their autonomy; rather, it is supported and complemented in advancing the clean energy transition. |
| • The political nature of central bank actions to manage transition and physical risks raises concerns about unmanaged risks in the global economy, specifically stranded asset risks. | • The political character of central bank efforts to handle transition and physical risks raises concerns about unmanaged threats to the global economy, especially stranded asset risks. |
| • A central bank climate index could increase transparency of the risk mitigation gap; international institutions governing central banks could set standards for climate and transition risk management. | • Transparency of the risk-mitigation gap could be improved by a central bank climate index, and international institutions governing central banks could set standards for climate and transition risk management. |

Table 13: **ROUGE on Original vs. Grammar-alternated Paraphrase.** We report ROUGE-1/2/L F1 scores between the two columns in Table 12. Despite identical meaning, scores are low. These illustrate ROUGE's sensitivity to word order and phrasing rather than semantic equivalence.

| Pair | ROUGE-1 F1 | ROUGE-2 F1 | ROUGE-L F1 |
| --- | --- | --- | --- |
| **Original vs. Rephrased** | 0.4058 | 0.1606 | 0.2319 |

## H.2    HUMAN BASELINE FOR SCI2POL-BENCH

We recruit two university-affiliated evaluators and compensate each with $100. Because full evaluation is time-intensive, we do not assess all questions. Instead, we randomly sample 25 questions from each task in Tasks 1–10 and 16–18. For Tasks 11–15, we randomly sample three questions each, using the same three underlying scientific papers across all five tasks to maintain consistency. Due to the limited number of evaluation questions, we report only the final results and do not conduct bootstrap significance testing. The resulting evaluations, together with the state-of-the-art Grok-3-beta results, are shown in Table 14. These results indicate that current LLMs still have meaningful room for improvement.

Table 14: **Detailed Human Baseline Evaluation.** We present the evaluation results obtained from two independent human evaluators.

| Model | Autocompletion | | | | Understanding | | Summarization |
|---|---|---|---|---|---|---|---|
| | T1 (F1) | T2 (F1) | T3 (F1) | T4 (F1) | T5 (F1) | T6 (F1) | T7 (Gem.) |
| **Evaluator 1** | 92.00 | 84.00 | 80.00 | 84.00 | 100.00 | 92.00 | 94.20 |
| **Evaluator 2** | 96.00 | 92.00 | 76.00 | 80.00 | 96.00 | 92.00 | 91.60 |
| **Grok-3-beta** | 71.05±2.77 | 63.76±2.79 | 33.12±2.92 | 35.15±3.08 | 90.42±0.80 | 69.83±1.64 | 83.45±0.05 |

| Model | Summarization | | | Generation | | | |
|---|---|---|---|---|---|---|---|
| | T8 (Gem.) | T9 (Gem.) | T10 (Gem.) | T11 (Gem.) | T12 (Gem.) | T13 (Gem.) | T14 (Gem.) |
| **Evaluator 1** | 96.20 | 91.00 | 89.00 | 95.67 | 87.24 | 88.74 | 96.84 |
| **Evaluator 2** | 94.60 | 92.20 | 87.20 | 94.19 | 89.12 | 87.32 | 91.23 |
| **Grok-3-beta** | 86.88±0.05 | 83.55±0.05 | 79.15±0.05 | 92.35±0.63 | 79.50±0.86 | 82.67±1.40 | 89.38±1.06 |

| Model | Generation | Verification | | | |
|---|---|---|---|---|---|
| | T15 (Gem.) | T16 (F1) | T17 (F1) | T18 (F1) | Average |
| **Evaluator 1** | 93.45 | 100.00 | 84.00 | 100.00 | 91.57 |
| **Evaluator 2** | 95.12 | 100.00 | 92.00 | 100.00 | 91.48 |
| **Grok-3-beta** | 89.58±0.94 | 98.60±0.42 | 59.26±1.67 | 98.48±0.50 | 77.01±1.20 |

### H.3 ANALYSIS OF COMMON FAILURE MODES

In this section, we summarize the most salient failure modes of strong general-purpose LLMs by task group and connect them to our design targets.

**Autocompletion (Tasks 1–4).** On scientific and policy autocompletion and sentence reordering, leading models often rely on shallow lexical cues rather than discourse structure. For Tasks 1–2, models frequently choose continuations that share surface overlap (entities, topical words) but violate local rhetorical flow (e.g., jumping to limitations or policy implications prematurely). For Tasks 3–4, they can correctly identify the "introductory" sentence but struggle to order the remaining sentences when all share similar vocabulary. These failures reflect limited pattern alignment to domain-specific discourse patterns and weak implicit modeling of how scientific and policy arguments unfold. Our design, which exposes models to well-structured paper-to-brief mappings, is designed to strengthen discourse-sensitive pattern alignment and implicit skills for ordering and continuations in this genre.

**Understanding (Tasks 5–6).** For sentence-level classification (Task 5), frontier models often conflate *Policy Implications* with *Recommendations*, or mislabel methodological descriptions as findings when they contain numerical results. This indicates insufficient specialization to Sci2Pol rhetorical roles and weak style awareness of policy-brief section functions. In contrast, performance on general scientific knowledge (Task 6) is comparatively strong and less affected by our domain-specific SFT; we treat Task 6 as a sanity check that specialization to Sci2Pol does not destroy broad knowledge. Our corpus, built around explicit section labels and section-specific prompts, directly targets improved pattern alignment and stylistic control over these roles.

**Summarization (Tasks 7–10).** For policy-oriented summarization, leading models systematically underperform along two axes. First, failures in factual grounding: models often hallucinate policy framings, quantitative effects, or stakeholder groups that are not present in the selected paragraphs, especially in Tasks 7 (Policy Problem) and 10 (Policy Implications). Second, failures in style and pattern alignment: generated outputs drift toward generic "science communication" prose, ignoring the requested section type (e.g., mixing methods into findings, or inserting recommendations in the policy problem). Tasks 8–9 reveal additional issues with functional-length adaptation: models either compress complex methods and findings into vague topical summaries, or over-elaborate with boilerplate explanations not grounded in the input text. Our in-context polishing prompt and SFT objective explicitly penalize these behaviors by requiring section-specific outputs, professional policy-brief style, and strict factual alignment with the target paper.

**Generation (Tasks 11–15).** When asked to generate sections or full briefs from the full paper, strong models exhibit compounding errors. For Tasks 11–14, they often produce sections that (i) are not cleanly separated, (ii) rely on generic policy tropes rather than paper-specific evidence, or (iii) introduce external context and speculative recommendations beyond the study's scope. These reflect weaknesses in factual grounding, style discipline, and implicit skill at mapping paper structure into the four-section brief template. For full-brief generation (Task 15), models sometimes maintain local coherence within paragraphs but fail at global pattern alignment: section ordering is irregular, headings are omitted or merged, and later sections contradict earlier claims. Our design addresses these issues by (i) training on high-quality, section-aligned paper–brief pairs, (ii) enforcing a stable four-part template in the polishing prompt, and (iii) emphasizing functional length and section-specific content during optimization, thereby improving both pattern alignment and implicit structural skills.

**Verification (Tasks 16–18).** In verification tasks, frontier models tend to over-accept plausible-sounding claims, leading to high recall but poor precision. For Task 16, models often label partially supported or overstated claims as fully supported, revealing limited sensitivity to scope qualifiers and caveats in the source paper. For Tasks 17–18, they struggle to distinguish subtly contradicted policy implications from valid ones, especially when the implication is globally reasonable but locally misaligned with the provided evidence. These failure modes are rooted in weak factual grounding and insufficient disciplined reasoning about what is actually entailed by the text. While Sci2Pol SFT is not trained directly on verification labels, the polishing objective, which enforces strict factual alignment and discourages speculative or extrapolatory language in generated briefs, is designed to mitigate these tendencies by cultivating more conservative, evidence-driven behaviors that transfer to verification settings.

### H.4 ANALYSIS OF IMPROVEMENTS FROM SFT

In this section, we analyze the improvements induced by supervised fine-tuning (SFT). Based on the design of the in-context polishing prompt (Table 77), we identify four aspects of learning: factual grounding, style, pattern alignment, and implicit skill acquisition. A detailed comparison of base and fine-tuned outputs is provided in Table 15, with additional analyses of leading-model shortcomings in Tables 16 and 17.

**Factual Grounding.** The in-context polishing prompt explicitly requires "strict factual alignment" with the target scientific paper and provides multiple high-quality examples. This setup strongly encourages the model to improve factual grounding. Specifically, the model learns to:

- **Rely exclusively on the target paper.** The prompt prohibits speculation and external examples, reinforcing adherence to the provided scientific evidence.
- **Capture all major findings.** SFT encourages summaries that match the scope of the study, avoiding both omission of central findings and unnecessary broadening.
- **Avoid hallucinations.** Exposure to accurate exemplars and strict alignment requirements reduces the likelihood of invented facts or unwarranted claims.

**Style.** We expose models to policy-brief writing conventions, including professional tone, academic rigor, and a standardized four-part layout (Policy Problem, Scientific Research Findings, Scientific Research Study Methods, and Policy Implications). Through this structure, the model learns:

- **Consistent four-section structure.** The SFT model generates outputs that more reliably follow the required organization.
- **Policy-brief writing style.** Fine-tuned outputs exhibit concise, declarative phrasing, policy relevance, and precise causal language.
- **Improved higher-level organization.** Transitions between sections become more coherent, with clearer logical connections between findings and implications.
- **Reduction of verbosity or filler.** The prompt explicitly disallows padding or oversimplification, teaching the model to maintain informational density without unnecessary elaboration.

**Pattern Alignment.** In Table 77, we provide three (paper, brief) examples and a fixed output schema, enabling the model to learn generalizable structural patterns without memorizing specific text. The model acquired:

- **Abstract communicative patterns.** These include framing a policy problem, narratively summarizing scientific evidence, and deriving policy implications strictly from the study's results.
- **Cross-paper generalization.** Although each exemplar concerns a different scientific domain, their shared functional structure teaches the model to transfer these patterns to unseen papers.

Importantly, the prompt design discourages memorization of surface-level form:

- The model does not reproduce specific sentences from the samples.
- The vocabulary, phrasing, and order of points vary across target papers, indicating structural rather than textual memorization.

**Implicit Skill Acquisition.** Beyond explicit patterns, SFT teaches several latent skills that contribute to improved performance:

- **Functional-length adaptation.** The model adjusts the depth of discussion to match the complexity of the target scientific paper, rather than imitating sample length.
- **Disciplined scientific reasoning.** The SFT model more consistently avoids speculation, external examples, or causal claims not supported by the paper, and provides accurate interpretations of empirical results.
- **Alignment between findings and policy implications.** The fine-tuned model produces implications that more directly follow from the paper's findings, yielding a clearer and more logically grounded causal chain.

Table 15: **Qualitative Comparison of Gemma-27B vs. Gemma-27B-SFT on Task 15.** Column 3 provides an analysis of what the model learns during supervised fine-tuning.

| Gemma-27B Output | Gemma-27B-SFT Output | Analysis of Improvements |
|---|---|---|
| "Even when rooftop solar is free for low-income households, take-up remains limited because qualified homeowners lack information, face high transaction costs, and may distrust programme providers. Existing referral schemes—which efficiently connect providers with eligible clients—typically rely on financial rewards alone, leaving many referrals unrealised and widening clean-energy equity gaps." | "Subsidized energy assistance programs, like those offering low-income solar installations, are vital for promoting energy justice and an equitable clean energy transition. However, these programs often suffer from low participation rates, leaving eligible households underserved. This under-subscription stems not only from economic barriers but also from non-economic factors like lack of information, complex processes, and distrust, resulting in inefficient allocation of resources and hindering progress towards equitable access to clean energy benefits." | **Factual Grounding + Style:** SFT output maintains stronger factual alignment with the paper's framing and avoids speculative or generic claims. It exhibits a more formal policy-brief tone ("energy justice", "equitable clean energy transition"), introduces clear causal chains, and avoids compression of complex factors into short sentences. The structure is clearer and more policy-oriented. |
| "Adding a small, unconditional US$1 gift (reciprocity appeal) raised referrals by 2.6-fold compared with a standard reward-only postcard and improved referral quality... Both interventions produced durable effects, continuing to outperform the control nine months after the campaign, indicating that they shifted neither timing nor total referrals." | "Adding a small, non-contingent gift ($1) alongside a referral reward reminder... and simplifying the referral process... increased referral rates by 2.6 to 5.2 times compared to simply reminding participants of the reward. This translated to 5.2 times more solar contracts at a lower cost per contract... The quality of referrals... remained comparable to the control group, and the increased volume of referrals still resulted in a substantial number of qualified leads." | **Factual Precision:** SFT output reports the quantitative findings more faithfully, reflecting correct magnitudes and their policy relevance (e.g., cost per contract, quality of referrals). **Pattern Alignment:** Follows the narrative structure seen in the training samples—findings grouped by mechanism, then scale, then interpretation. **No Memorization:** Different phrasing and structure from samples; reasoning aligns with the target paper. |
| "Researchers partnered with GRID Alternatives to conduct a large, pre-registered randomised controlled trial... Eleven days later, a reminder postcard was sent. Referral behaviour... was tracked for 17 weeks and nine months. Poisson and logistic regressions... were used to estimate treatment effects." | "Researchers partnered with a non-profit administering a low-income solar program and conducted a randomized controlled trial... Data on referral rates, referral quality... and resulting solar contracts were collected... and analyzed using statistical methods including logistic regression and Poisson count models. The study accounted for potential confounding factors through stratification during randomization and robust statistical analysis." | **Implicit Skill Acquisition:** SFT output expands the methods narrative with clearer explanation of data collection, statistical approaches, and rationale—matching the "narrative methods" requirement in the SFT prompt. **Style Enforcement:** Avoids bullet points, uses narrative flow, mirrors sample briefs' level of technical detail. |
| "Pairing financial rewards with reciprocity and simplification multiplies programme reach at minimal incremental cost, making it a practical complement to larger subsidies... Programme managers can tailor interventions to their objectives: reciprocity alone yields higher-quality referrals, while reciprocity + simplification maximises volume and cost-effectiveness." | "Integrate Behavioral Insights... Small Incentives Can Yield Large Returns... Streamline Processes... Targeted Outreach... Cost-Effectiveness... Program Evaluation." | **Style + Structure:** SFT output produces a cleaner, bullet-pointed policy implications section, matching the required format ("bullet points only"). **Pattern Generalization:** SFT consistently aligns implications with findings—no speculative statements, each point tied to an observed effect. **Improved Policy Orientation:** Uses the framing style of the sample briefs (action verbs, policy directives). |

Table 16: **Qualitative Limitations of DeepSeek-V3 on Task 15.** The limitations include factual grounding, stylistic control, pattern alignment, and implicit skill acquisition.

| DeepSeek-V3 Output | Analysis of Limitations |
|---|---|
| "Climate-tech startups are critical for accelerating the commercialization of innovative technologies needed to meet global climate goals. However, these startups face high failure rates and chronic underinvestment due to long development timelines, high capital requirements, and misaligned investor expectations." | **Factual Grounding:** The model introduces generic and speculative claims (e.g., "chronic underinvestment," "misaligned investor expectations") that do not appear in the target paper. It expands beyond the study context, weakening alignment with the evidence. **Style:** Tone resembles an op-ed or general commentary, not a structured policy brief; lacks concise problem framing. **Pattern Alignment:** Does not follow the requested four-section structure and drifts toward thematic discussion rather than evidence-based summary. **Implicit Skills:** Fails to preserve the functional depth or specificity expected from a scientifically grounded brief. |
| "Corporate investment is strongly associated with startup success (110% higher exit rate) but also with failure (62% higher failure rate), suggesting corporations selectively fund higher-risk, high-reward ventures." | **Factual Grounding:** The model fabricates numerical results ("110% higher exit rate," "62% higher failure rate"), which do not appear in the original paper. **Style:** Uses bullet-like enumeration in Findings, deviating from the narrative style required. **Pattern Alignment:** Introduces causal interpretation ("high-risk, high-reward ventures") not supported by the study, violating evidence-based reporting. **Implicit Skills:** Lacks disciplined reasoning; interprets correlation causally. |
| "Applied Cox Proportional Hazards models to assess how funding sources correlate with startup exits and failures." | **Factual Grounding:** Substitutes the study's actual methods with invented ones (the real paper uses descriptive correlations and regression-based comparisons, not Cox models). **Style:** Overly technical for policy brief norms; deviates from the narrative methods requirement. **Pattern Alignment:** Fails to mirror the structure used in the exemplars, which emphasize high-level methodological narrative rather than model-heavy detail. **Implicit Skills:** Does not adapt the level of technicality appropriately, violating "functional-length" guidance. |
| "Target high-risk sectors underserved by private capital (e.g., nuclear, geothermal)... Create platforms... Mitigate risks of private investment... Leverage regional strengths." | **Factual Grounding:** Several points are speculative and not grounded in the paper's empirical findings (e.g., nuclear or geothermal sectors are not mentioned). **Style:** Bullet points are overly long, mixing policy recommendations unspecified in the study. **Pattern Alignment:** Introduces external examples and normative proposals that diverge from evidence-based implications. **Implicit Skills:** Fails to maintain alignment between Findings $\rightarrow$ Implications; proposed policies are not traceable back to the study's results. |

Table 17: **Qualitative Limitations of GPT-4o on Task 15.** The limitations include factual grounding, stylistic control, pattern alignment, and implicit skill acquisition.

| GPT-4o Output | Analysis of Limitations |
|---|---|
| "California faces a significant challenge in reducing greenhouse gas (GHG) emissions from its transportation sector, primarily due to high dependence on oil extraction." | **Factual Grounding:** The model reframes the policy problem incorrectly—the paper discusses oil-extraction emissions and community impacts, not transportation-sector dependence. **Style:** Tone is explanatory and generic, not aligned with compact, policy-brief framing. **Pattern Alignment:** Does not follow the structured policy problem template (missing specific grounding in setbacks, excise taxes, and carbon taxes). **Implicit Skills:** Shows poor adherence to the task requirement of rephrasing the study's actual motivation. |
| "Among these, setbacks yield the highest health benefits and equity gains... but also lead to more substantial worker compensation losses compared to excise and carbon taxes." | **Factual Grounding:** Omits key quantitative details and nuances (e.g., study's nuanced 2045 target scenarios; absence of explicit percentages). **Style:** Narrative is broad rather than specific; lacks explicit causal reasoning tied to the paper's model results. **Pattern Alignment:** Loses the ordering convention used in exemplars (Findings $\rightarrow$ quantitative evidence $\rightarrow$ constraints). **Implicit Skills:** Simplifies complex findings, failing to preserve functional-length or depth appropriate to the study. |
| "The study employs a comprehensive methodology combining an empirical oil-production model, an air pollution dispersion model, and an employment input-output model." | **Factual Grounding:** The original paper does not use an input-output model; methodology is partially hallucinated. **Style:** Uses vague terms ("comprehensive methodology"), lacking the structured narrative required in SFT. **Pattern Alignment:** Deviates from exemplar tone, which emphasizes structured description over meta-evaluation. **Implicit Skills:** Does not constrain technical detail to the appropriate depth; misses mention of limitations as done in samples. |
| "Policymakers must consider integrating supply-side policies with demand-side strategies to ensure comprehensive and coordinated reductions in statewide and global GHG emissions." | **Factual Grounding:** Introduces policy recommendations not supported by the paper (demand-side strategies are outside the study's scope). **Style:** Shifts to normative messaging inappropriate for a scientific policy brief. **Pattern Alignment:** Does not adhere to the exemplar requirement that policy implications be strictly derived from the Findings. **Implicit Skills:** Violates "strict factual alignment"—adding novel arguments inconsistent with paper evidence. |

## H.5 IMPACT OF PROMPT LENGTH FOR TASKS 1-4

For Tasks 1-4, we test whether two sentences of context are sufficient for a model to predict the next sentence. Using Task 1 as an example, we evaluate 50 samples with 2-, 3-, and 4-sentence prompts, nesting shorter prompts within the 4-sentence version for fairness. Results in Table 18 suggest that prompt length has a limited effect on model accuracy.

Table 18: **Impact of Prompt Length for Task1.**

| Model | Task 1 | | |
|---|---|---|---|
| | **Len = 2 (F1)** | **Len = 3 (F1)** | **Len = 4 (F1)** |
| **Grok-3-beta** | 80.22±5.19 | 87.20±5.02 | 82.18±4.99 |
| **DeepSeek-R1** | 71.10±7.09 | 72.32±6.72 | 75.04±6.49 |
| **Qwen3-235B** | 40.84±6.60 | 44.70±7.73 | 50.08±8.71 |
| **GPT-4o** | 64.08±8.19 | 65.84±7.62 | 61.94±7.52 |
| **Claude-3.7-Sonnet** | 82.60±4.95 | 80.18±6.01 | 82.30±5.24 |
| **LLaMA-3.3-70B-IT** | 46.06±7.52 | 54.68±7.80 | 56.52±8.23 |
| **Mistral-Large** | 49.70±7.15 | 49.68±8.62 | 49.42±7.62 |
| **DeepSeek-V3** | 62.88±6.98 | 68.34±7.25 | 66.84±7.10 |
| **LLaMA-4-Maverick** | 60.18±7.23 | 59.58±7.46 | 60.02±6.93 |
| **Gemma-3-27B** | 56.68±7.96 | 57.96±7.85 | 55.86±8.01 |
| **Gemma-3-12B** | 46.88±6.94 | 50.58±7.83 | 43.78±7.82 |
| **Qwen3-8B** | 50.16±7.72 | 58.28±7.57 | 56.38±7.82 |
| **LLaMA-3.1-8B-IT** | 31.18±6.41 | 35.92±6.80 | 27.84±5.78 |

## H.6 HUMAN–LLM JUDGE AGREEMENT EVALUATION

In this section, we evaluate whether the LLM-based judge align with human judgments. We conduct two tests to assess agreement between humans and the LLM judge.

- **Direct human scoring (Appendix H.6.1).** We ask two human evaluators to directly score model outputs and compare their averaged scores with those produced by the Gemini judge.
- **Confusion-matrix validation Appendix H.6.2.** We take the top five and bottom five LLaMA-4-Maverick summaries (ranked by Gemini-2.5-Pro on Task 7), have a policy expert label them as "good" or "bad", and compute a confusion matrix to assess judge reliability.

### H.6.1 DIRECT HUMAN SCORING

We randomly sample 10 cases from Tasks 9 and 11-15 for human evaluation (using the same 10 cases across Tasks 11–15). Due to the limited number of evaluation questions, we report only the final results and do not conduct bootstrap significance testing. The results show that Gemma-27B-SFT consistently outperforms DeepSeek-V3 and GPT-4o, aligning closely with our reference-based evaluation. For completeness, we present one representative example for Tasks 9 and 15 along with brief analyses in Tables 20 to 22.

Table 19: **Direct Human Scoring.**

| | Task 9 | | Task 11 | | Task 12 | |
|---|---|---|---|---|---|---|
| **Model** | **Gemini** | **Human** | **Gemini** | **Human** | **Gemini** | **Human** |
| **Gemma-3-27B-SFT** | 83.40±1.03 | 82.00 | 91.76±0.90 | 89.20 | 78.26±1.35 | 75.10 |
| **DeepSeek-V3** | 76.90±0.05 | 74.00 | 91.50±0.93 | 88.40 | 78.24±1.46 | 75.60 |
| **GPT-4o** | 71.38±0.06 | 70.00 | 89.17±0.98 | 86.30 | 72.69±1.05 | 71.30 |

| | Task 13 | | Task 14 | | Task 15 | |
|---|---|---|---|---|---|---|
| **Model** | **Gemini** | **Human** | **Gemini** | **Human** | **Gemini** | **Human** |
| **Gemma-3-27B-SFT** | 67.34±2.45 | 65.20 | 86.65±1.53 | 83.25 | 83.65±1.75 | 80.00 |
| **DeepSeek-V3** | 90.29±0.99 | 84.50 | 87.62±1.28 | 84.50 | 83.51±1.65 | 79.10 |
| **GPT-4o** | 72.81±1.50 | 68.20 | 78.36±1.54 | 78.25 | 68.93±1.31 | 65.90 |

Table 20: **Examples of Task 9 Outputs Across Different Models.**

**Question:**

You are given a passage from a scientific paper that describes part of the research study's methodology. Summarize the method or approach using policy-brief style sentences. Your output should:
- Describe the model, data, or procedure mentioned in the passage
- Use clear and accessible language (technical terms are allowed when necessary)
- Focus only on what is present in the passage.

**Scientific Text:**

Our research methodology is informed by the principles underpinning ethical Australian Indigenous research outlined in the Australian Institute of Aboriginal and Torres Strait Islander Studies Code of Ethics for Aboriginal and Torres Strait Islander Research. Our research team is committed to the principles of Indigenous self-determination, Indigenous leadership, impact and value, sustainability and accountability. V.N.D. is senior Aboriginal researcher at Tangentyere Research Hub in Mparntwe (Alice Springs) and a visiting Indigenous fellow at the Australian National University (ANU) Centre for Aboriginal Economic Policy Research. M.K. is senior policy manager at Tangentyere Research Hub and a visiting fellow at ANU's Centre for Aboriginal Economic Policy Research.

**Summary:**

**Answers (Study Methods Summarization)**

| (i) Gemma-27B-SFT | (ii) DeepSeek-V3 | (iii) GPT-4o |
|---|---|---|
| Potential control areas were digitally delineated from 300 m raster data on land cover, development, elevation and coastal proximity, excluding cells within or near existing or protected coastal zones, to mimic planners' 1982 information and drawing process. | To identify plausible control areas that could have been designated as part of the Coastal Barrier Resources System (CBRS) in 1982 but were not, researchers used an automated process mirroring the original manual boundary-drawing approach. They relied on historical data—including aerial photos, topographic maps, and 300-meter-resolution grids of land cover, development, elevation, and coastal proximity—to replicate the information available to planners at the time. Only coastal grid cells within 2 km of the shore were considered, while excluding areas that were fully water, already part of the CBRS or other protected areas, or within 2 km of existing CBRS units. This method ensured control areas were comparable to actual CBRS units in terms of geographic and developmental characteristics. | The study identifies potential control areas for Coastal Barrier Resources System (CBRS) designation from 1982 by mimicking the original selection criteria through an automated approach. This method involves analyzing aerial photographs, topographic maps, and 300-meter resolution gridded data on land cover, development, elevation, and proximity to the coast. Only areas within 2 kilometers of the coastline are considered, excluding entirely water-covered cells, those within existing or planned CBRS units, protected areas, and areas near CBRS units to prevent overlap of effects. |

**Analysis:**

Gemma-27B-SFT provides the strongest policy-brief style summary. Its response is the most concise and the most aligned with policy-focused communication norms, distilling the method into a single clear statement without unnecessary operational details. Compared with DeepSeek-V3 and GPT-4o, which produce much longer and more technical descriptions, Gemma-27B-SFT captures the essential methodological mechanism with precision and clarity. This makes Gemma-27B-SFT's output more accessible, more readable, and better suited for policy briefs that prioritize streamlined, high-level methodological reporting.

Table 21: **Examples of Task 15 Outputs Across Different Models — Part 1 (Problem & Findings).** Scientific Paper: "What is driving the drug overdose epidemic in the United States?" (Thombs et al., 2020b)

| (i) Gemma-27B-SFT | (ii) DeepSeek-V3 | (iii) GPT-4o |
|---|---|---|
| **What is driving the drug overdose epidemic in the United States?** **Policy Problem** Despite decades of U.S. policy intervention, drug-overdose deaths continue to rise, exceeding 70,000 in 2017. The crisis now reflects the interaction of two structural forces: (1) increasingly potent and widely available opioids, both licit and illicit, and (2) deepening economic inequality that erodes stability for the poorest households. State-level analyses from 2006–2017 show that opioid availability predicts overdose mortality, whereas educational attainment does not. The most powerful inequality predictor is the declining income share of the bottom 20%. Without addressing both supply-side drug availability and structural economic deprivation, overdose mortality will likely continue climbing and spreading into other substances, including cocaine and psychostimulants. **Scientific Research Findings** • The income share held by the bottom 20% is the strongest inequality predictor of drug-related mortality. • Increasing the bottom-quintile income share corresponds to reduced overdose mortality. • States with higher opioid prescription rates have higher drug-related mortality on average. | **What is driving the drug overdose epidemic in the United States?** **Policy Problem** Despite two decades of escalating intervention, the United States continues to face a lethal drug-overdose epidemic that claimed more than 70,000 lives in 2017 alone—an age-adjusted rate of 21.7 deaths per 100,000, more than triple the 1999 level. New evidence shows that the crisis is no longer driven solely by prescription-opioid misuse or by "deaths of despair" among less-educated workers; instead, it reflects a dangerous convergence of (1) an abundant, increasingly synthetic opioid supply (licit and illicit) and (2) widening income inequality that strips the bottom-quintile of earners of economic and social buffers. State-level analysis (2006–2017) reveals that while higher opioid-prescribing rates predict higher overdose mortality, educational attainment does not, and the strongest inequality predictor is low income share for the bottom 20 percent—not gains at the top. Current policy, which centers on regulating prescribers and expanding treatment, therefore tackles only half the problem; without addressing the structural deprivation that fuels demand for high-risk substances, overdose deaths—and their spill-over to stimulants like cocaine and methamphetamine—will likely continue to rise. **Scientific Research Findings** • Of the four measures of income inequality that we used, the share of income going to the bottom 20% was the key inequality measure associated with drug-related mortality. • An increase in the share of income going to the bottom 20% of earners was associated with a reduction in drug-related mortality. • States with a higher opioid prescription rate, on average, had higher drug-related mortality. | **What is driving the drug overdose epidemic in the United States?** **Policy Problem** The United States is facing a devastating drug overdose epidemic, which remains a significant public health crisis. In 2017, the country witnessed 70,237 drug overdose deaths, with opioids being a significant contributor, accounting for 67.8% of these fatalities. While the focus has often been on opioids, cocaine and psychostimulants are increasingly involved, highlighting the evolving nature of this epidemic characterized by polysubstance use. This crisis reflects broader socioeconomic disparities and necessitates an understanding of its complex drivers to inform effective public health policies. **Scientific Research Findings** 1. Educational attainment does not significantly predict overdose mortality. 2. Opioid prescription rates are positively associated with drug mortality. 3. Income inequality is a key determinant: lower income share for the bottom 20% correlates with higher overdose mortality. |

Table 22: **Examples of Task 15 Outputs Across Different Models — Part 2 (Methods, Implications, Analysis).** Scientific Paper: "What is driving the drug overdose epidemic in the United States?" (Thombs et al., 2020b)

| (i) Gemma-27B-SFT | (ii) DeepSeek-V3 | (iii) GPT-4o |
|---|---|---|
| **Scientific Research Study Methods** The study links CDC WONDER mortality data, CDC opioid-prescribing data, and ACS inequality and education statistics for all 50 states and DC (2006–2017; 611 observations). All continuous variables were logged and appropriately centered. A two-level within–between random-intercept model disentangles temporal variation within states from structural differences across states. Covariates include median household income, manufacturing employment share, census-region indicators, and quadratic time trends. **Policy Implications** Effective prevention requires targeting both drug supply and structural inequality. Policymakers should regulate opioid manufacturing, distribution, and prescribing practices, while implementing redistributive reforms that raise the economic position of the lowest-income households. Strengthened safety nets, progressive taxation, wage supports, and healthcare reform can mitigate structural vulnerability and reduce the population-level risk of overdose mortality. | **Scientific Research Study Methods** We compiled a balanced panel of all 50 states plus DC for 2006–2017 (N = 611 state-years) by merging CDC WONDER drug-overdose mortality rates with (i) CDC opioid-prescribing rates, (ii) ACS educational-attainment and income-distribution data (shares of the top 5%, top 20%, bottom 20%, and the Gini), and (iii) state economic structure and region controls. All continuous variables were logged and centred. Two-level random-intercept models separated within- and between-state variation. **Policy Implications** Our findings suggest that an adequate prevention response will require both limiting the availability of opiates and redistributing income, especially to earners at the bottom of the income distribution. Policy makers should strengthen regulations on opioid manufacturing, distribution, and prescription, while also addressing structural economic inequality to reduce vulnerability in low-resource communities. | **Scientific Research Study Methods** The research uses a two-level random intercept model with state-level data from 2006–2017, covering all 50 states and DC. This method allows for the simultaneous analysis of within-state and between-state variation in education, opioid prescriptions, income inequality, and drug mortality. **Policy Implications** Policies should tighten regulation of opioid prescribing and marketing, strengthen monitoring systems, and address structural economic inequality. Redistributive policies—such as progressive taxation, higher minimum wages, and expanded social services—could improve economic stability among vulnerable groups and reduce overdose risk. |

**Analysis:** Gemma-27B-SFT provides the strongest overall policy brief. It is the most concise, maintains a clean policy-brief structure, and links empirical findings directly to actionable policy recommendations. DeepSeek-V3 is detailed but overly long and less suited to policymaking audiences; GPT-4o is clear but less structurally rigorous. Gemma therefore provides the most interpretable and policy-relevant response.

### H.6.2 CONFUSION-MATRIX VALIDATION

In this part, we evaluate Gemini's scoring of LLaMA-4-Maverick outputs on Task 7 as a toy example. Specifically, we select the five samples that receive the highest scores and the five that receive the lowest scores from Gemini-2.5-Pro. We then ask our political expert to review these ten samples and determine which are "good" or "bad" summarizations (5 "good" and 5 "bad" summarizations). We show the confusion matrix in Table 23, and the results show that our reference-free method is indeed reliable.

Table 23: **Confusion Matrix of the Gemini-2.5-Pro-based Reference-free Judge.** We evaluate Gemini's scoring of LLaMA-4-Maverick outputs on Task 7 as a toy example. Specifically, we select the five samples that receive the highest scores and the five that receive the lowest scores from Gemini-2.5-Pro. We then ask our political expert to review these ten samples and determine which are "good" or "bad" summarizations (5 "good" and 5 "bad" summarizations). The confusion matrix shows the reliability of this method.

| Expert Judgment | Predicted Good | Predicted Bad |
|---|---|---|
| **Good** | 5 | 0 |
| **Bad** | 0 | 5 |

### H.7 WHY TASKS 11-14 BEYOND TASK 15: SECTION VS. FULL BRIEF GENERATION

We provide a concrete example comparing LLM output generated section-by-section versus as a full policy brief. Specifically, we use LLaMA-3.1-8B-Instruct on the paper *European battery self-sufficiency* (Link et al., 2025a). We show the results in Table 24 and Table 25. This example illustrates why Tasks 11–14 are necessary in addition to Task 15 in Sci2Pol-Bench.

- Task 15 evaluates end-to-end policy brief generation, but Tasks 11-14 aim to capture complementary dimensions of model ability. When LLMs generate briefs one section at a time, they often include more technical details: precise figures, dataset sizes, or methodological constructs. This section-wise mode highlights a model's capacity to reproduce fine-grained scientific evidence. However, the sections may drift in terminology, exaggerate findings, or fail to connect implications back to results. The outputs are accurate in isolation but lack global coherence.

- In contrast, full-brief generation produces more fluent and coherent outputs. The narrative flows smoothly from the policy problem to the findings, methods, and implications, often with organized recommendations and a consistent tone. Yet this approach tends to sacrifice technical precision. Numbers are simplified, mechanisms skipped, and caveats dropped. These lead to overconfident or generic claims that weaken policy relevance.

- Thus, Tasks 11-14 complement Task 15 by disentangling precision from coherence. Section-level generation stresses factual grounding and attention to detail, while full-brief generation stresses holistic structure and readability. Evaluating both provides a fuller picture of LLM performance and reveals trade-offs that Task 15 alone cannot capture. These comparisons justify the inclusion of Tasks 11-14 as critical complements to Task 15.

Table 24: **Section-by-section vs. Full-brief Generation**. We use LLaMA-3.1-8B-Instruct on the paper *European battery self-sufficiency* (Link et al., 2025a). The full brief (Task 15) shows smoother narrative and coherence, while the section-by-section outputs (Tasks 11–14) surface fine-grained technical precision.

| Task 15: Full brief | Tasks 11–14: Sections |
|---|---|
| **Policy Problem:** The European Union's ambition to strengthen national capabilities and self-sufficiency in battery production is at risk due to uncertain demand and production growth rates. The study highlights the need for stakeholders to accelerate production capacities and reckon with demand growth post-2030 to support Europe's battery prospects. *(Sets stakes succinctly and frames the policy arc; coherence-first framing without tying to modelling choices yet.)* | **Task 11 (Policy Problem):** The European Union's ambition to strengthen national capabilities and self-sufficiency in battery production is at odds with the growing demand for batteries in the region. By 2030, European battery demand is projected to exceed **1.0 TWh/yr**, far outpacing domestic production capacity, which will require highly ambitious growth rates of 31–68%/yr. Our research suggests that while Europe can cover at least **50–60%** of its demand via domestic production by 2030, **90% self-sufficiency** seems feasible but far from certain. This exposes Europe to the risk of domestic production shortfalls, decreased competitiveness, and reliance on imports, threatening to undermine the region's net-zero goals and climate change mitigation efforts. Immediate attention is needed from policymakers and industry leaders to overcome these production challenges, strengthen supply chains, and ensure sustainable battery development. *(Numbers and risks front-loaded; standalone problem statement precise but disconnected from other sections.)* |
| **Scientific Research Findings:** The study uses probabilistic modeling to project future battery demand and domestic production in Europe, covering the EU, EFTA, and the United Kingdom. The results indicate that:
• European demand is likely to exceed **1.0 TWh/yr by 2030**, outpacing domestic production.
• Domestic production capacity is more likely to fall behind terawatt hour scales by 2030, with momentum increasing after 2025.
• At least **50–60%** of demand can be covered by domestic production by 2030.
• **90% self-sufficiency** by 2030 seems feasible but far from certain.
*(Aggregates key quantities clearly for readability; scenario granularity and caveats are compressed compared to section-level detail.)* | **Task 12 (Findings):** The study finds that Europe is likely to experience ambitious growth in battery demand, reaching at least **1.0 TWh by 2030**, with a high probability of surpassing 1.0 TWh by 2026. However, domestic production capacities are more likely to fall behind terawatt hour scales by 2030, with momentum increasing after 2025. Despite this, it is very likely that Europe can cover at least **50–60%** of its demand via domestic production by 2030, and even **90% self-sufficiency** seems feasible but far from certain. The study emphasizes the urgent need for reliable policies to ensure domestic production and develop a comprehensive battery ecosystem, including localization of raw material production and recycling. To achieve this, stakeholders must accelerate the materialization of production capacities and consider the potential for ambitious growth rates in production capacity. *(High technical precision in probabilities, thresholds, and timelines; but no linking back to policy problem or methods.)* |

Table 25: **Section-by-section vs. Full-brief Generation (Continued from Table 24).**

| Task 15: Full brief (continued) | Tasks 11–14: Sections (continued) |
| --- | --- |
| **Scientific Research Study Methods:** The study employed a **probabilistic modeling** approach, using the latest empirical data to project future battery demand and domestic production in Europe. The model was implemented in **Python**, and the results were based on a **Monte Carlo simulation (N=1,000)** to construct feasibility spaces and classify findings by probability. *(Method summarized compactly; omits modelling artifacts and domain specifics that appear explicitly in Task 13.)* | **Task 13 (Methods):** This study projects future battery demand and domestic production capacity in Europe using **probabilistic modelling**. The research design combines empirical data with a **Monte Carlo simulation (N=1,000)** to create feasibility spaces and probabilistic statements. The model evaluates Europe's prospects towards battery self-sufficiency and was implemented in **Python**. The study focuses on high-energy batteries for battery electric vehicles (BEVs), which are the main demand driver. European demand is projected to increase, but domestic production capacity is likely to fall behind, indicating a risk of domestic production shortfalls. The study also analyzes the European battery value chain, including raw material sourcing and recycling, and highlights the importance of reliable industrial policies to support Europe's competitiveness in the battery industry. *(Technical scaffolding explicit here: modelling approach, simulation size, demand drivers. This detail is absent in Task 15's smoother summary.)* |

| Task 15: Full brief (continued) | Tasks 11–14: Sections (continued) |
| --- | --- |
| **Policy Implications:**
• Accelerate production capacities to meet growing demand and support Europe's battery prospects.
• Reckon with demand growth post-2030 to ensure a stable and sustainable battery ecosystem.
• Implement industrial policies that balance trade protectionism and global competitiveness to create attractive, predictable home markets.
• Invest in local battery production to avoid shortages in domestic production and keep track of international developments.
• Prioritize establishing fully scaled and sustainable value chains.
• Use **net materialized production capacities** as a basis for projections rather than announced capacities.
*(Actionable storyline; recommendations not explicitly re-anchored to each numerical threshold or method assumption.)* | **Task 14 (Policy Implications):** To ensure a competitive and sustainable battery ecosystem in Europe, policymakers and industry stakeholders must take immediate action to support domestic production expansion and address potential risks. Our study highlights the need for industrial policies that balance trade protectionism and global competitiveness, creating attractive and predictable home markets that reduce risks for industry players. Reliable policies are crucial to lower risks and ensure certain predictability. To achieve **90% self-sufficiency by 2030**, stakeholders must accelerate the materialization of production capacities and consider demand growth post-2030. Investments in local battery production are necessary to avoid shortages in domestic production while keeping track of international developments. *(Concrete recommendations are explicit but disconnected from modelling details or numerical findings. Lacks coherence across sections.)* |

## H.8 OVER-ENDORSEMENT ANALYSIS ON TASKS 16 AND 18

We take Tasks 16 and 18 as examples to conduct an over-endorsement analysis by comparing predicted labels ("SUPPORT" or "CONTRADICT") against ground-truth annotations. Specifically, we compute the number of false positives—instances where the model predicts "SUPPORT" despite the true label being "CONTRADICT"—as well as false negatives, where the model fails to endorse a true "SUPPORT" case by predicting "CONTRADICT". For each task, we calculate the false positive rate (FPR), defined as the proportion of CONTRADICT-labeled instances that are incorrectly predicted as SUPPORT, and the false negative rate (FNR), defined as the proportion of SUPPORT-labeled instances that are incorrectly predicted as CONTRADICT. We also compute the ratio of false positives to false negatives (FPR/FNR ratio). A high FPR/FNR ratio indicates a model's propensity to over-endorse, potentially overstating the strength of scientific support behind science and policy claims. This analysis provides insight into models' decision biases, which is critical for assessing their reliability in science and policy-relevant contexts.

We show the detailed results in Table 26, and we have the following findings:

- The FPR/FNR ratios show consistent results across Tasks 16 and 18, with the exception of the model DeepSeek-V3, which exhibits divergent behavior.
- Most advanced models do not exhibit over-endorsement, whereas smaller models such as LLaMA-3.3-70B-IT, Gemma-3-27B, Gemma-3-12B, and LLaMA-3.1-8B-IT demonstrate a tendency to over-endorse scientific claims and policy implications.

Table 26: **Over-endorsement Analysis on Tasks 16 and 18.** We compute the number of false positives—instances where the model predicts "SUPPORT" despite the true label being "CONTRADICT"—as well as false negatives, where the model fails to endorse a true "SUPPORT" case by predicting "CONTRADICT". For each task, the false positive rate (FPR) is defined as the proportion of CONTRADICT-labeled instances that are incorrectly predicted as SUPPORT, and the false negative rate (FNR) is defined as the proportion of SUPPORT-labeled instances that are incorrectly predicted as CONTRADICT. We also show the ratio of false positives to false negatives (FPR/FNR ratio). A high FPR/FNR ratio indicates a model's propensity to over-endorse, potentially overstating the strength of scientific support behind science and policy claims.

| | Task 16 | | | Task 18 | | |
|---|---|---|---|---|---|---|
| **Model** | **FPR** | **FNR** | **FPR / FNR** | **FPR** | **FNR** | **FPR / FNR** |
| **Grok-3-beta** | 0.47 | 2.34 | 0.2000 | 0.57 | 2.56 | 0.2222 |
| **DeepSeek-R1** | 0.95 | 10.05 | 0.0930 | 1.72 | 3.99 | 0.4286 |
| **Qwen3-235B** | 0.00 | 8.88 | 0.0000 | 0.86 | 4.56 | 0.1875 |
| **GPT-4o** | 0.24 | 2.57 | 0.0909 | 0.29 | 4.84 | 0.0588 |
| **Claude-3.7-Sonnet** | 0.24 | 3.04 | 0.0769 | 0.29 | 2.85 | 0.1000 |
| **LLaMA-3.3-70B-IT** | 0.95 | 0.93 | 1.0000 | 4.01 | 2.28 | 1.7500 |
| **Mistral-Large** | 0.95 | 7.94 | 0.1176 | 2.58 | 16.24 | 0.1579 |
| **DeepSeek-V3** | 0.95 | 1.17 | 0.8000 | 2.29 | 1.71 | 1.3333 |
| **LLaMA-4-Maverick** | 0.71 | 5.37 | 0.1304 | 1.72 | 3.42 | 0.5000 |
| **Gemma-3-27B** | 2.13 | 1.4 | 1.5000 | 8.31 | 2.56 | 3.2222 |
| **Gemma-3-12B** | 5.92 | 2.34 | 2.5000 | 10.60 | 0.85 | 12.3333 |
| **Qwen3-8B** | 0.24 | 14.49 | 0.0161 | 1.43 | 7.12 | 0.2000 |
| **LLaMA-3.1-8B-IT** | 18.01 | 1.87 | 9.5000 | 18.34 | 2.85 | 6.4000 |

## H.9   Information Leakage Check of In-context Polishing in Section 3.3

When applying in-context polishing to revise policy briefs (Section 3.3), we use three reference samples from the 85 expert-written briefs. The three reference samples are

- *Reliable industrial policies required to support the ramp-up of European battery production* – Paper (Link et al., 2025a); Brief (Link et al., 2025b).
- *How central banks address climate and transition risks* – Paper (Shears et al., 2025a); Brief (Shears et al., 2025b).
- *Faster deployment of renewables stabilizes electricity prices in Europe* – Paper (Navia Simon and Diaz Anadon, 2025b); Brief (Navia Simon and Diaz Anadon, 2025a).

Although the prompt for GPT-o3 instructs the model to mimic only writing style and format (Table 77), there remains a risk of information leakage. To validate that this step does not introduce leakage, we perform three ablation studies:

- Excluding the three samples from Task 15 and then using Task 15 for justification (Appendix H.9.1).
- Using the three latest published paper-brief pairs (detailed in Appendix J.4) as the in-context examples to polish the dataset, and subsequently using this newly polished dataset for supervised tuning (Appendix H.9.2).
- Using an additional 8 newly published samples (after May 10th, 2025, detailed in Appendix J.3) as the new samples for Task 15, and testing the model performance on these 8 samples with Task 15 (Appendix H.9.3).

### H.9.1   Excluding Three Samples from Justification

We exclude the three samples from Task 15 and use Task 15 for justification. We show the results in Table 27. Gains remain stable, and this shows that in-context polishing does not induce leakage.

Table 27: **Information Leakage Check of In-context Polishing in Section 3.3.** Performance on Task 15 with all 85 pairs vs. with the three reference samples removed (82 pairs). Gains remain stable, and this shows that in-context polishing does not induce leakage.

| Model | 85 Expert-written Pairs | | 82 Expert-written Pairs | |
|---|---|---|---|---|
| | T15 (Reference-based Score) | Gain | T15 (Reference-based Score) | Gain |
| **LLaMA-3.1-8B-IT** | 59.05±1.55 | - | 58.77±1.62 | - |
| **LLaMA-3.1-8B-SFT** | 72.76±1.75 | **+13.71** | 72.32±1.90 | **+13.55** |
| **Gemma-3-12B** | 75.71±1.43 | - | 75.49±1.47 | - |
| **Gemma-3-12B-SFT** | 76.06±1.66 | **+0.35** | 76.10±1.75 | **+0.61** |
| **Gemma-3-27B** | 80.89±1.26 | - | 80.30±1.30 | - |
| **Gemma-3-27B-SFT** | 83.65±1.75 | **+2.76** | 83.54±1.76 | **+3.24** |

### H.9.2   In-Context Polishing with Three New Published Pairs

We use the three latest published paper-brief pairs as the three in-context examples to polish the dataset, and use this newly polished dataset to do supervised tuning. The three examples include:

- *Nutritional outcomes of irrigation expansion* – Paper (Mehta et al., 2025a); Brief (Mehta et al., 2025b).
- *Managing development choices is essential to reduce coastal flood risk in China* – Paper (Wang et al., 2025c); Brief (Wang et al., 2025b).
- *Heat pumps can help alleviate residential energy insecurity in the USA* – Paper (Ye et al., 2025a); Brief (Ye et al., 2025b).

We show the results in Table 28. Gains remain stable, and this shows that in-context polishing does not induce leakage.

Table 28: **Information Leakage Check 2 of In-context Polishing in Section 3.3.** Performance on Task 15 using three new non-overlapping in-context polishing examples.

| | Original 3 In-context Samples | | New 3 In-context Samples | |
| --- | --- | --- | --- | --- |
| Model | T15 (Reference-based Score) | Gain | T15 (Reference-based Score) | Gain |
| **LLaMA-3.1-8B-IT** | 59.05±1.55 | - | 59.05±1.55 | - |
| **LLaMA-3.1-8B-SFT** | 72.76±1.75 | **+13.71** | 72.91±1.96 | **+13.86** |
| **Gemma-3-12B** | 75.71±1.43 | - | 75.71±1.43 | - |
| **Gemma-3-12B-SFT** | 76.06±1.66 | **+0.35** | 76.31±1.65 | **+0.60** |
| **Gemma-3-27B** | 80.89±1.26 | - | 80.89±1.26 | - |
| **Gemma-3-27B-SFT** | 83.65±1.75 | **+2.76** | 83.43±1.80 | **+2.54** |

### H.9.3    EVALUATING ON NEW EIGHT TEST SAMPLES

We use another eight newly published paper-brief pairs to generate Task 15. These include all the new peer-reviewed pairs up to Nov. 12th, 2025. We list the details of these eight new pairs in Appendix J.3. We show the results in Table 29. Gains remain stable, and this shows that in-context polishing does not induce leakage.

Table 29: **Information Leakage Check 3 of In-context Polishing in Section 3.3.** Performance on Task 15 with eight new samples.

| | Original 85 Expert-written Pairs | | New 8 Expert-written Pairs | |
| --- | --- | --- | --- | --- |
| Model | T15 (Reference-based Score) | Gain | T15 (Reference-based Score) | Gain |
| **LLaMA-3.1-8B-IT** | 59.05±1.55 | - | 58.43±1.12 | - |
| **LLaMA-3.1-8B-SFT** | 72.76±1.75 | **+13.71** | 72.02±1.36 | **+13.59** |
| **Gemma-3-12B** | 75.71±1.43 | - | 74.81±1.12 | - |
| **Gemma-3-12B-SFT** | 76.06±1.66 | **+0.35** | 75.58±1.43 | **+0.77** |
| **Gemma-3-27B** | 80.89±1.26 | - | 81.02±1.62 | - |
| **Gemma-3-27B-SFT** | 83.65±1.75 | **+2.76** | 83.21±1.46 | **+2.19** |
| **DeepSeek-V3** | 83.51±1.65 | - | 82.52±1.25 | - |
| **GPT-4o** | 68.93±1.31 | - | 67.53±1.59 | - |

## H.10 Athlete as Judge Testing: GPT Family

A common concern in benchmark design is potential circularity when GPT family models e.g., GPT-o3) generate intermediate data, such as classification labels or gold prompts for tasks where other GPT family models (e.g., GPT-4o) are later evaluated. This "athlete as judge" setup could favor models from the same family because of shared training signals and writing style. To test this, we design two experiments:

- Cross-lineage evaluation: Evaluate the performance of GPT-4o and DeepSeek-V3 on datasets generated by GPT-o3 or a different lineage model, DeepSeek-R1 (Appendix H.10.1).
- Model substitution: Replace GPT-o3 with MiniMax-M2 in the data construction process (Appendix H.10.2).

### H.10.1 Potential Circularity: GPT vs. DeepSeek Family

To test whether this issue affects Sci2Pol-Bench, we conduct a controlled comparison on Task 16. We evaluate GPT-4o and DeepSeek-V3 on datasets generated either by GPT-o3 or by a different lineage model, DeepSeek-R1. As shown in Table 30, DeepSeek-V3 consistently outperforms GPT-4o, and both models drop 10-12 points when prompts come from DeepSeek-R1 instead of GPT-o3. DeepSeek-R1 prompts are harder. This demonstrates that any benefit from GPT-o3 prompts applies broadly across models rather than giving GPT-family models a special advantage, and it confirms that the "athlete as judge" concern does not affect our setting.

Table 30: **GPT vs. DeepSeek Family on Task 16.** DeepSeek-V3 consistently outperforms GPT-4o, and both models drop 10-12 points when prompts come from DeepSeek-R1 instead of GPT-o3. DeepSeek-R1 prompts are harder. This demonstrates that any benefit from GPT-o3 prompts applies broadly across models rather than giving GPT family models a special advantage, and it confirms that the "athlete as judge" concern does not affect our setting.

| Model | Dataset (GPT-o3) | Dataset (DeepSeek-R1) |
|---|---|---|
| **GPT-4o** | $98.61 \pm 0.39$ | $86.93 \pm 1.26$ |
| **DeepSeek-V3** | $98.97 \pm 0.37$ | $89.00 \pm 1.06$ |

### H.10.2 Model Substitution by MiniMax-M2

GPT-o3 is utilized to generate data for Tasks 5, 11, 13, 16, and 18, and to polish all corpus documents. To assess the influence of this model choice, we design an ablation study using Tasks 11 and 13 as illustrative examples. The experiment proceeds as follows: (i) We replicate the original data construction method, but replace the data generation model (GPT-o3) with MiniMax-M2. (ii) We then evaluate the performance of GPT-4o, DeepSeek-V3, and Mistral-Large on the data constructed by MiniMax-M2 for Tasks 11 and 13. (iii) The evaluation utilizes a dual LLM Judge setup, employing both Gemini and MiniMax-M2. The full results are presented in Table 31. These results demonstrate consistency when employing different large language models in the dataset construction process.

Table 31: **Model Performance on Tasks 11 and 13 with Dataset Constructed by MiniMax-M2.**

| | Gemini as Judge | | | | MiniMax-M2 as Judge | | | |
|---|---|---|---|---|---|---|---|---|
| **Data Construction** | **GPT-o3** | | **MiniMax-M2** | | **GPT-o3** | | **MiniMax-M2** | |
| **Model** | **T11** | **T13** | **T11** | **T13** | **T11** | **T13** | **T11** | **T13** |
| **DeepSeek-V3** | 91.50±0.93 | 90.29±0.99 | 91.10±1.01 | 89.93±0.94 | 54.32±2.44 | 65.56±1.08 | 54.98±2.34 | 65.78±1.11 |
| **GPT-4o** | 89.17±0.98 | 72.81±1.50 | 89.53±0.94 | 71.97±1.43 | 52.01±2.40 | 62.54±1.24 | 51.78±2.34 | 62.83±1.29 |
| **Mistral-Large** | 87.64±0.92 | 64.10±1.93 | 87.02±0.97 | 63.98±1.34 | 51.71±1.56 | 61.64±1.46 | 50.94±1.36 | 61.03±1.44 |

## H.11 SATURATION TESTING

We do the saturation analysis as in Figure 4. We report the relationship between Sci2Pol-Bench performance and three commonly used model characteristics: (a) model size, (b) pre-training FLOPs, and (c) release date. Although several model developers do not publicly disclose all training details, the approximations used here follow standard practice in the LLM evaluation literature and provide reliable, order-of-magnitude comparisons.

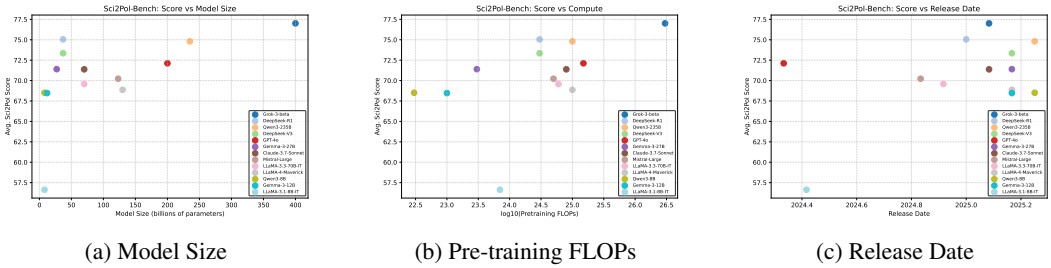

(a) Model Size      (b) Pre-training FLOPs      (c) Release Date

Figure 4: **Satuation Testing: Model Size, Pre-training FLOPs, and Release Date.**

**Score vs. Model Size.** Very large models (e.g., GPT-4o) perform strongly, but several mid-sized models such as DeepSeek-V3 match or exceed them, indicating that parameter count alone does not determine policy-reasoning quality. Smaller models ($< 10B$) consistently underperform, suggesting a minimum capacity threshold. For proprietary models that do not release full architectural details, we use the closest publicly stated or family-level parameter estimate. These values are widely used by open-source leaderboards and preserve the correct relative placement among models.

**Score vs. Pre-training FLOPs.** Although the highest-FLOPs models generally achieve the strongest results, many models in the $10^{24}$–$10^{25}$ FLOP range overlap substantially, showing diminishing returns once compute passes a sufficient scale. Several developers no longer publish exact FLOP counts. Therefore, we follow the standard estimation heuristic FLOPs $\approx 6 \times N_{\text{params}} \times T_{\text{tokens}}$ using known or reported model sizes, token counts, and architectural multipliers. Even when exact numbers are unavailable, the order-of-magnitude estimates are robust and do not affect the qualitative scaling trends.

**Score vs. Release Date.** The third panel visualizes performance relative to the earliest publicly documented release month. While newer models (2024–2025) generally occupy the upper performance range, the significant variation observed within yearly cohorts indicates that recency is not the dominant factor. Instead, our supervised fine-tuning results indicate that alignment strategies and the specific data mixture used in training show a stronger influence on Sci2Pol-Bench performance.

Overall, the three panels show that Sci2Pol-Bench performance does not follow a simple scaling law. Model size, compute, and release date each matter, but do not fully explain cross-model differences. Instead, high performance is often associated with improved alignment strategies, targeted supervised finetuning, and richer training mixtures. These are factors that help models reason about scientific evidence and policy trade-offs beyond what raw scale alone can provide.

### H.12 Comparison of Writing from an Abstract, an Introduction, and a Full Paper

We compare policy brief generation when the input is the abstract, the introduction, or the full scientific paper. Writing from the full paper requires a much longer input context. This increases inference cost, fine-tuning cost, and technical difficulty. At the same time, it offers richer information for grounding policy-relevant insights.

We first present a concrete example using Grok-3-beta to generate policy briefs for the paper *How central banks address climate and transition risks* (Shears et al., 2025a), based on three input scopes: (i) the full paper, (ii) the introduction only, and (iii) the abstract only. We show the generated outputs in Table 32 and Table 33. The outputs enable a detailed comparison across input settings as follows.

- When a model writes from the abstract only, it captures the headline claim, the main outcome, and the high-level implications. The tone reads confident, and the structure feels clean. However, the draft lacks mechanism, methods detail, and scope limits. Numbers, when present, drift toward round values, and uncertainty language fades. Policy advice turns generic, because the draft does not anchor actions to specific evidence or design choices. The abstract frames motivation and result, but it does not carry the caveats that make a brief credible, so the model tends to overstate precision and under-specify risks.

- When a model writes from the introduction only, it provides better context. The draft states the policy stakes, defines terms, and maps the debate. The flow improves, and the problem framing reads stronger. Yet, the introduction centers motivation and gaps, not verification. Methods remain thin, and heterogeneity across cases stays offstage. Claims may adopt the paper's persuasive setup, which can tilt the brief toward advocacy language. Without the full evidence base, the draft still softens uncertainty, skips mechanism, and treats recommendations at the slogan level, not at the instrument level.

- When a model writes from the full paper, it can link evidence to tools. The draft cites methods, describes mechanisms, and carries uncertainty statements. It notes limits, and it distinguishes what is feasible from what is likely. The narrative supports actionable advice, because the draft can name concrete instruments, timing, and trade-offs. Here, the main risks shift. The model can drown readers in detail, mix minor numbers into the main thread, or copy framing that belongs in a technical appendix. If the paper is dense, the model may misread tables or blend results, so careful summarization still matters.

Table 32: **Task 16 Outputs Across Input Scopes.** We use Grok-3-beta to generate the policy brief for *How central banks address climate and transition risks* (Shears et al., 2025a) from three input scopes: (i) full paper, (ii) introduction only, and (iii) abstract only. This comparison supports our analysis in Appendix H.12. The table illustrates how source scope affects Task 15 output quality.

| (i) Full Paper | (ii) Introduction Only | (iii) Abstract Only |
|---|---|---|
| **Policy Problem** | | |
| Climate change and the global transition away from fossil fuels present substantial risks to economic and financial stability. These include "transition risks" (e.g., losses on fossil fuel assets becoming obsolete, known as stranded assets) and "physical risks" (e.g., economic damage from storms, droughts). Central banks, responsible for financial stability, have begun addressing these climate-related risks. However, their actions vary significantly across countries, leading to uncertainty about whether these crucial institutions are effectively managing the underlying economic dangers or are primarily influenced by other factors. This inconsistency creates a potential gap in mitigating systemic financial risks related to climate change. | Climate change and the global transition to clean energy create significant risks for the financial system, including "stranded assets" (devalued fossil fuel investments) and the uncertainties of new clean energy technologies. These risks threaten the stability of national and global economies. Central banks have been identified as key institutions for managing these financial risks, yet their response has been highly inconsistent across countries. This uneven approach creates a critical policy challenge: understanding what drives central banks to act on climate risk is essential for ensuring financial stability and supporting an orderly energy transition. | Electricity price volatility, driven by unpredictable fluctuations in global fossil fuel markets, poses a significant economic risk to households, businesses, and national economies. As Europe transitions its energy system, a critical question for policymakers is whether a high share of intermittent renewables, like solar and wind, will increase or decrease this price instability. Understanding this relationship is essential for designing resilient and cost-effective long-term energy strategies. |
| **Scientific Research Findings** | | |
| The study reveals considerable differences in how central banks in major economies (OECD and G20) are managing climate risks. Researchers categorized actions into "re-risking" (integrating climate risks into financial supervision, targeting fossil fuels and physical impacts) and "de-risking" (reducing investment risks for clean energy). Key findings indicate:
* Central bank actions are not significantly associated with the country's economic exposure to transition risks, such as the size of its fossil fuel sector or the growth rate of its renewable energy industry.
* Exposure to physical climate risks showed a statistically significant, though unexpected, positive correlation only with de-risking activities (promoting clean energy), not with re-risking (managing existing climate/fossil fuel risks).
* Instead, central bank climate actions are significantly linked to the political context:
* Countries with stronger national climate policies tend to have central banks more engaged in re-risking fossil fuel investments and physical climate impacts.
* Countries with higher public concern about climate change tend to have central banks more involved in de-risking clean energy investments.
* This suggests central banks may be reinforcing national policy directions rather than acting independently as purely technocratic risk managers or compensating for weak national climate policies. | The study reveals that central bank actions on climate risk are not primarily driven by the magnitude of the underlying economic threats. Instead, political factors are the most significant driver.
* Political Will is Crucial: Central banks are far more likely to implement climate risk management measures in countries that already have strong national climate policies and where public opinion shows high concern about climate change.
* Economic Risks are Overlooked: There is no significant statistical link between the scale of a country's transition risks (such as exposure to stranded fossil fuel assets or the need for clean energy investment) and the actions taken by its central bank.
* A "Risk Mitigation Gap" Exists: The disconnect between economic risk exposure and central bank action means that financial systems in countries with high climate-related economic risks but low political will are likely being left unprotected. | The research demonstrates that renewable energy sources have a powerful stabilizing effect on electricity prices. Key findings include:
* By 2030, under current national energy plans, the European power system will be significantly less vulnerable to natural gas price shocks. A €1 increase in the price of natural gas is projected to raise the average electricity price by only €1, a substantial reduction from the current impact of €1.40.
* More ambitious renewable deployment yields even greater stability. Increasing the deployment of solar and wind by 30% beyond the 2030 targets would slash the sensitivity to gas prices even further, to less than €0.50 for every €1 increase.
* This price stabilization effect creates a tangible economic benefit for society, which the study terms an "insurance value." By buffering consumers and industries from fossil fuel price spikes, renewables provide a form of economic insurance. |

Table 33: **Task 15 Outputs Across Input Scopes (Continued from Table 32).**

| (i) Full Paper | (ii) Introduction Only | (iii) Abstract Only |
| --- | --- | --- |
| **Scientific Research Study Methods** | | |
| The findings are based on an original dataset compiled by the researchers, documenting climate-related policies enacted by central banks across 47 Organization for Economic Co-operation and Development (OECD) and Group of 20 (G20) countries up to August 2023. Policies were systematically classified as primarily "re-risking," "de-risking," or both. Composite scores for re-risking and de-risking activity were calculated for each country, weighting policies based on their potential impact and cost. The researchers then used statistical linear regression analysis to test the correlation between these scores and various national indicators, including measures of economic risk (fossil fuel sector size, financial sector size, renewable energy growth, physical climate risk exposure) and political factors (climate policy stringency index, public opinion data on climate concern). The analysis controlled for other country-level variables like GDP, central bank independence, and institutional mandates. | The researchers conducted a systematic analysis of 47 central banks across the Organization for Economic Co-operation and Development (OECD) and Group of 20 (G20) nations. They created a comprehensive new dataset of central bank actions and developed a novel classification system to distinguish between two key functions: 1. Re-risking: Actions that integrate the financial risks of fossil fuel investments into standard risk management, discouraging "brown" assets. 2. De-risking: Actions that reduce the investment risks associated with new clean energy technologies, facilitating "green" assets. The study then statistically tested whether these actions were more closely associated with measures of economic risk or with indicators of political pressure, such as existing climate laws and public polling data. | The findings are based on a sophisticated computer simulation of the European power markets as they are projected to operate in 2030. The model incorporated the energy infrastructure and renewable capacity targets outlined in Europe's National Energy and Climate Plans. To test the resilience of this future system, researchers subjected the model to the real-world historical variability of weather patterns, electricity demand, and volatile fossil fuel prices. The study introduced a new metric, "$\beta$-sensitivity," to precisely measure how much the average electricity price would change in response to a one-euro change in the price of natural gas. |
| **Policy Implications** | | |
| The research suggests that policymakers should not expect central banks to proactively fill gaps left by inadequate national climate policy, particularly concerning the risks of stranded fossil fuel assets. Key implications include: * Central banks tend to complement, not substitute: Central bank actions on climate risk appear influenced by, and aligned with, existing national climate policy strength and public sentiment. They are more likely to amplify existing political will than to counteract its absence. * Potential risk mitigation gap: In countries with significant exposure to climate-related financial risks (especially transition risks) but weak climate policies or low public concern, central banks may not be taking sufficient action, leaving substantial risks unmanaged. * Transparency and international standards may help: Increasing transparency about central bank climate actions (e.g., via comparative indices) could create pressure on laggards. Furthermore, developing stronger international standards or best practices through bodies like the Bank for International Settlements (BIS) or the Financial Stability Board (FSB) could encourage more consistent climate risk management across countries, regardless of domestic political headwinds. | These findings suggest that policymakers cannot assume central banks will act as autonomous, technocratic first responders to climate-related financial threats. * National Governments Must Lead: Central banks tend to *reinforce* existing national climate policy, not substitute for it. To ensure financial stability, governments must enact clear, strong, and credible climate and energy transition policies. This provides the political legitimacy central banks appear to require before they will act. * Do Not Depend on Central Banks to Fill Policy Gaps: Hopes that central banks could compensate for a lack of political action on decarbonization are misplaced. Their role is more responsive than proactive. * Close the Risk Mitigation Gap: In jurisdictions with weak climate policy, other financial regulators and government bodies must step in to address the unmanaged financial risks of the energy transition. Relying solely on the central bank in these situations could leave the economy vulnerable. * Clarify Central Bank Mandates: Governments can empower central banks by explicitly including the management of climate-related financial risks within their official mandates, thereby reducing ambiguity and concerns over "mission creep." | The "insurance value" of renewables is a public good that is not currently recognized or rewarded by market mechanisms. This leads to an undervaluing of renewable energy in investment and policy decisions. To correct this, policymakers should: 1. Explicitly factor the price-stabilizing benefits of renewables into policy and planning. Cost-benefit analyses for energy projects and support schemes should go beyond direct generation costs to include the economic value of reduced price volatility. 2. Pursue and potentially exceed current 2030 renewable energy targets. The research provides a strong economic case that accelerating the deployment of solar and wind power is a direct investment in national economic stability and resilience against fossil fuel market shocks. 3. Design market and support mechanisms that recognize this insurance value. This could inform the structure of renewable energy auctions or other incentives, ensuring that the full societal benefit of renewables is captured, leading to a more efficient and secure energy transition. |

# I CROSS-DOMAIN GENERALIZATION TESTING

We test the model's generalization capacity across domains outside of its primary training focus. Due to the scarcity of readily available peer-reviewed scientific paper-policy brief pairs in other domains, we conduct an exploratory evaluation using one representative paper from each of the following important fields: AI, Robotics, Mathematics, Economics, and Humanities. The generated policy briefs are then subjected to human evaluation. We test three models: Gemma-27B-SFT, DeepSeek-V3, and GPT-4o. The resulting evaluation indicates that our supervised fine-tuning successfully generalizes to other domains.

We provide the details of 5 papers as follows:

- AI: "Generative AI needs adaptive governance (Reuel and Undheim, 2024)". We show the results in Tables 34 and 35
- Robotics: "Anticipatory policy design for robots in public spaces (Mintrom et al., 2025)". We show the results in Tables 36 and 37
- Mathematics: "Educational policy to improve mathematics instruction at scale: Conceptualizing contextual factors (Ryve and Hemmi, 2019)". We show the results in Tables 38 and 39.
- Economics: "Financial inclusion, the shadow economy and economic growth in developing economies (Younas et al., 2022)". We show the results in Tables 40 and 41.
- Humanities: "Science cited in policy documents: Evidence from the Overton database (Fang et al., 2024)". We show the results in Tables 42 and 43.

Table 34: **AI: "Generative AI needs adaptive governance (Reuel and Undheim, 2024)".**

| (i) Gemma-27B-SFT | (ii) DeepSeek-V3 | (iii) GPT-4o |
|---|---|---|
| **Policy Problem**
Generative artificial intelligence (AI) is developing with unprecedented speed, scale, and capacity to augment human work. Its ability to create realistic text, code, images, and other content means it is rapidly becoming a general-purpose technology that will reshape economies and societies. Existing "one-and-done" regulatory approaches—disclosure, registration, and fixed risk-management rules—are ill-suited to this dynamic environment. Rigid rules risk obsolescence, misalignment, and stifling innovation, while leaving gaps for new harms to emerge. Policymakers need a governance system that can keep pace with the technology, balance agility with accountability, and incorporate new knowledge as it becomes available.

**Scientific Research Findings**
The authors demonstrate that generative AI differs from previous AI generations in four key ways: it is more accessible, more complex, more expensive, and evolves faster. These characteristics create a "pacing problem" in which laws and regulations cannot keep up. Drawing on environmental governance literature, they show that "adaptive governance"—characterized by flexibility, learning, and stakeholder co-evolution—is essential for managing such rapidly changing systems. They define an adaptive AI governance framework that assigns roles to governments, industry, academia, civil society, and citizens, and outlines shared activities (sharing best practices, collaborating, using insights, monitoring, informing, adapting) and actor-specific activities (financing, anticipating, challenging, training, innovating). | **Policy Problem**
Generative AI poses unprecedented governance challenges due to its rapid development, broad societal impact, and ability to augment human cognition and creativity. Traditional regulatory approaches—static, slow-moving, and top-down—are ill-suited to address AI's dynamic risks (e.g., bias, misinformation, labor disruption) and emergent capabilities. Policymakers face a "pacing problem," where rigid rules risk irrelevance or misalignment with evolving technology.

**Scientific Research Findings**
Key insights from the paper include: - Generative AI's rapid adoption, complexity, and high costs concentrate expertise and oversight within industry, sidelining academia and civil society. - Current governance (e.g., the EU AI Act) relies on fixed metrics (e.g., compute thresholds) that may quickly become obsolete as algorithms improve. - Adaptive governance—iterative, multi-stakeholder, and evidence-based—has succeeded in domains like climate change and public health, offering a model for AI. | **Policy Problem**
The paper addresses the societal and strategic challenge of effectively governing generative AI technologies that evolve rapidly and impact human performance, economic structures, and societal norms. Traditional governance systems struggle to keep pace with the transformative nature of these technologies, posing risks in regulation, societal impacts, and foundational governance structures.

**Scientific Research Findings**
Key findings highlight that generative AI, due to its rapid advancement and broad applicability, challenges traditional governance frameworks. Existing regulatory strategies, which are often static and inflexible, fail to accommodate the dynamic nature of AI development, leading to potential risks of regulatory misalignment and outdated policies. The paper proposes that adaptive governance, which is flexible and evolves along with AI technologies, is necessary to manage these challenges. |

Table 35: **AI: "Generative AI needs adaptive governance (Reuel and Undheim, 2024)".**

| (i) Gemma-27B-SFT | (ii) DeepSeek-V3 | (iii) GPT-4o |
|---|---|---|
| **Scientific Research Study Methods** The study combines conceptual analysis, literature review, and case study reasoning. It synthesises insights from AI development trajectories, investment trends, research output, and public discourse. It draws on environmental governance theory and agile software development to articulate the principles of adaptive governance. The authors illustrate how an adaptive framework could be operationalised through examples such as multi-stakeholder governance committees embedded in government, structured policy review cycles, regulatory sandboxes, and dedicated AI governance research funding. The EU AI Act is presented as a contrast case to show why static governance struggles with emergent foundation models. | **Scientific Research Study Methods** The paper synthesizes case studies (e.g., EU AI Act, COVID-19 responses) and proposes a framework derived from environmental governance and agile methodologies. It defines: - **Actors**: Governments, industry, academia, civil society, and citizens. - **Shared activities (SCUMIA)**: Sharing best practices, collaboration, monitoring, and adaptation. - **Actor-specific tasks (FACTI)**: Financing, anticipating risks, challenging norms, training, and innovation. | **Scientific Research Study Methods** The researchers employed a mixed-methods approach, drawing from case studies, current policy analyses, and interdisciplinary frameworks. The study involved reviewing AI governance strategies across different global entities and theoretical frameworks from parallel fields like environmental and public health governance. This comprehensive analysis informed the development of the proposed adaptive governance framework. |
| **Policy Implications** 1. **Shift from static to adaptive regulation:** Adopt governance structures that can be updated rapidly, drawing on stakeholder input and new evidence. 2. **Establish inclusive governance bodies:** Embed permanent, cross-sectoral committees within government agencies. 3. **Mandate iterative policy cycles:** Design regulations with review periods, pilot programmes, and revision pathways. 4. **Invest in AI governance research:** Fund independent AI safety and ethics research and ensure computational access for academia and civil society. 5. **Promote transparency and accountability:** Require interoperable risk frameworks and mandatory incident disclosure. 6. **Cultivate AI literacy:** Educate citizens to support democratic participation and responsible use. | **Policy Implications** 1. **Replace Static Laws with Iterative Processes**: - Establish government-led "adaptive committees" with rotating expert input to review and update regulations (e.g., biannual revisions). - Pilot regulatory sandboxes to test governance measures in controlled environments. 2. **Multi-Stakeholder Collaboration**: - Mandate cross-sector transparency (e.g., incident registries, compute/resource disclosures) to inform policy. - Fund independent oversight bodies (akin to the IPCC) to assess risks and avoid regulatory capture. 3. **Capacity Building**: - Invest in public AI literacy and workforce training to empower citizen engagement. - Require corporate boards to include AI ethics officers and share risk assessments. 4. **Global Coordination**: - Promote international standards for interoperability and safety, learning from nuclear/biotech governance. **Downsides and Mitigations** - *Regulatory uncertainty*: Provide clear roadmaps for phased policy updates. - *Insufficient oversight*: Layer third-party audits with government accountability. - *Industry dominance*: Fund public-sector AI R&D to balance private influence. | **Policy Implications** The findings suggest a shift towards adaptive governance models is essential for the effective regulation of generative AI. Policymakers are encouraged to adopt a dynamic approach that emphasizes flexibility, interactivity, and rapid responsiveness. Such governance should engage multiple stakeholders, including governments, industry, academia, civil society, and citizens, in a collaborative effort to continually refine and adapt policies to technological advancements and societal needs. Adaptive governance can reduce risks, ensure AI's beneficial development, and maintain public trust. Additionally, investing in regulatory R&D and establishing robust monitoring and feedback processes will support the sustainability and effectiveness of AI governance frameworks. |

**Analysis:**
Gemma-27B-SFT delivers the strongest policy-brief performance. It provides a more coherent problem framing, a more detailed and conceptually accurate description of adaptive governance mechanisms, and significantly more actionable policy recommendations. DeepSeek-V3 is detailed but verbose, and GPT-4o is clear but omits key theoretical and operational components. Gemma-27B-SFT therefore achieves the best balance of accuracy, clarity, and policy relevance.

Table 36: **Robotics: "Anticipatory policy design for robots in public spaces (Mintrom et al., 2025)".**

| **(i) Gemma-27B-SFT** | **(ii) DeepSeek-V3** | **(iii) GPT-4o** |
|---|---|---|
| **Policy Problem** Robots are rapidly entering streets, parks, malls, and civic infrastructure, creating a major socio-technical transition. Current governance remains overwhelmingly reactive, concentrating on immediate safety and privacy risks while neglecting aesthetics, co-creation, equitable access, and long-term innovation trajectories. Without anticipatory policy design that integrates citizen perspectives and broader social wellbeing goals, robot deployments risk uneven benefits, reduced public trust, and missed opportunities for positive transformation. | **Policy Problem** Robots are increasingly present in public spaces across major Asian cities, offering benefits like efficiency and convenience but also posing risks related to safety, privacy, equity, and social disruption. Current policies in leading cities (Hong Kong, Seoul, Shanghai, Singapore, Taipei, Tokyo) primarily address safety and productivity but often neglect broader societal impacts, such as aesthetics, equitable access, and participatory governance. Without anticipatory policy frameworks, rapid robot adoption may exacerbate inequalities, erode public trust, or create unintended harms. | **Policy Problem** The integration of robots into public spaces presents both opportunities and challenges for society. While robots hold the potential to enhance productivity, convenience, and innovation, they also raise significant concerns around safety, privacy, ethics, and equitable access. Cities around the world, particularly in Asia, are at the forefront of facing these challenges as robot density increases. Without proactive and anticipatory policy design, there is a risk of negative social and economic consequences, such as exacerbated inequalities and erosion of public trust. |
| **Scientific Research Findings** A comparative review of six Asian cities (Hong Kong, Seoul, Shanghai, Singapore, Taipei, Tokyo) shows that policies overwhelmingly emphasize safety and, secondarily, privacy. Broader concerns central to socio-technical transitions are inconsistently addressed: only Seoul and Tokyo reference equitable access, and only Shanghai explicitly includes co-creation. Aesthetics, distribution of productivity gains, and systemic innovation rarely appear despite their importance for public legitimacy and long-term value. This pattern reveals a governance gap between technical regulation and desired social outcomes. | **Scientific Research Findings** 1. Safety and Privacy: Policies universally address physical safety but inconsistently cover data privacy and ethical risks (e.g., facial recognition biases). 2. Productivity vs. Social Costs: Robots are promoted for economic gains, but their impact on jobs, public space aesthetics, and human interactions is understudied. 3. Equity Gaps: Few cities (e.g., Seoul, Tokyo) explicitly address equitable access, risking exclusion of marginalized groups. 4. Co-Creation Deficits: Only Shanghai emphasizes co-design with citizens; most policies lack mechanisms for public input. 5. Innovation Focus: Systemic innovation is prioritized in tech development but not in ensuring inclusive governance or long-term societal benefits. | **Scientific Research Findings** The research highlights that current policy frameworks in six Asian cities—Hong Kong, Seoul, Shanghai, Singapore, Taipei, and Tokyo—primarily focus on issues of safety and productivity. These cities have made varying degrees of progress in addressing privacy and ethics, systemic innovation, and equitable access. However, there is a distinct lack of emphasis on aesthetics and co-creation, which are crucial for fostering positive human-robot interactions and ensuring broader community acceptance. |

Table 37: **Robotics: "Anticipatory policy design for robots in public spaces (Mintrom et al., 2025)".**

| (i) Gemma-27B-SFT | (ii) DeepSeek-V3 | (iii) GPT-4o |
|---|---|---|
| **Scientific Research Study Methods** The study systematically analyzes legislation, regulations, and strategy documents governing robots in public spaces across six Asian cities. Using a seven-topic framework (safety, privacy/ethics, productivity, aesthetics, co-creation, equitable access, systemic innovation), each policy document was assessed using a three-tier scoring system: directly addressed, signalled, or not mentioned. Documents were translated when necessary, and comparative narratives were produced to reveal cross-jurisdictional patterns and gaps. | **Scientific Research Methods** The study analyzed 30 policy documents across six high-robot-density Asian cities using a seven-topic framework: 1. Safety 2. Privacy/Ethics 3. Productivity 4. Aesthetics 5. Co-Creation 6. Equitable Access 7. Systemic Innovation Policies were scored based on explicit attention to each topic, supplemented by media analysis and cross-jurisdictional comparisons. | **Scientific Research Study Methods** The study employed a comparative analysis of existing public policy and regulatory settings across six major Asian cities. By reviewing legislation, regulations, and strategic documents, the researchers assessed these cities using a seven-topic analytical framework: safety, privacy and ethics, productivity, aesthetics, co-creation, equitable access, and systemic innovation. The methodology involved thorough policy searches using government websites and translating policy documents where necessary to ensure comprehensive analysis. |
| **Policy Implications** 
 • Embed anticipatory governance tools such as foresight exercises, horizon scanning, and iterative policy review cycles. 
 • Expand policy scope beyond safety and privacy to explicitly address aesthetics, co-creation, equitable access, and systemic innovation. 
 • Institutionalize co-design through participatory workshops, neighbourhood pilots, and continuous citizen feedback loops. 
 • Promote inclusive innovation via regulatory sandboxes, research incentives, and data-sharing mechanisms. 
 • Encourage international and cross-city learning, drawing on practices in Seoul, Tokyo, and Shanghai. | **Policy Implications** 1. Expand Policy Scope: Move beyond safety/productivity to address ethics (e.g., bias audits), aesthetics, and equitable access. 2. Co-Design with Citizens: Adopt participatory methods (e.g., citizen juries, design labs) to shape robot deployment rules. 3. Equity Safeguards: Mandate accessibility features (e.g., for disabilities) and subsidize robot services in underserved areas. 4. Anticipatory Governance: Establish regulatory sandboxes to test robot impacts and update policies iteratively. 5. Cross-City Learning: Create platforms for Asian cities to share best practices on inclusive robot governance. 

 **Conclusion** Asian cities are pioneers in robot deployment but must shift from reactive to anticipatory policymaking. Prioritizing co-design, equity, and ethical oversight will ensure robots enhance–not undermine–public wellbeing. Policymakers should treat robots as social, not just technological, innovations. | **Policy Implications** The findings suggest that anticipatory policy design is essential for maximizing the benefits and minimizing the risks associated with robots in public spaces. Policymakers should focus on developing inclusive co-design processes that broaden stakeholder participation, including diverse resident voices. Engagement in proactive policy learning and innovation diffusion across jurisdictions can lead to more effective management of socio-technical transitions. By prioritizing co-creation and aesthetics alongside established considerations like safety and productivity, policymakers can foster environments where robots contribute to social and economic wellbeing. Expanding anticipatory efforts to incorporate these neglected areas will help ensure that the integration of robots into public spaces preserves public trust, enhances community experiences, and promotes equitable access to technological advancements. |

**Analysis:**
Gemma-27B-SFT provides the strongest brief. It captures the full socio-technical transition outlined in the paper, integrates all seven analytical topics, and links findings to concrete anticipatory governance mechanisms. DeepSeek-V3 is comprehensive but more descriptive and repetitive. GPT-4o is clear but omits key cross-city contrasts and understates co-creation and equity dimensions. Gemma-27B-SFT offers the best balance of accuracy, breadth, and policy relevance.

Table 38: **Mathematics: "Educational policy to improve mathematics instruction at scale: Conceptualizing contextual factors (Ryve and Hemmi, 2019)".**

| (i) Gemma-27B-SFT | (ii) DeepSeek-V3 | (iii) GPT-4o |
|---|---|---|
| **Policy Problem** Large-scale improvement of mathematics instruction in Sweden is hindered by a persistent misalignment between national educational policies and actual classroom practice. Despite strong teacher autonomy and progressive intentions, reforms frequently fail to achieve meaningful instructional change because they do not account for three key contextual features: teachers' system-level autonomy, their traditionally reactive role in classrooms, and the long-standing culture of invisible pedagogy that favors implicit guidance over explicit instructional leadership. Without policies that directly address these contextual constraints, high-level initiatives risk limited uptake and minimal impact. | **Policy Problem** The paper addresses the challenge of improving mathematics instruction at scale, particularly within specific educational contexts like Sweden. A key issue is the gap between educational policies and their effective implementation in classrooms, influenced by both explicit factors (e.g., curriculum, teacher training) and implicit contextual factors (e.g., teacher positioning, pedagogical traditions). Policymakers and educators often struggle to account for these contextual elements, leading to mismatches between policy intentions and classroom realities. | **Policy Problem** The educational system faces a critical challenge in improving mathematics instruction at a large scale. Traditional methods have often failed to account for the varied contextual factors present in different educational environments, resulting in ineffective policy implementation. Addressing these challenges is key to enhancing student outcomes in mathematics. |
| **Scientific Research Findings** Evidence from the five-year "Count on Vasteras" project shows that systemic instructional improvement depends on recognizing three interrelated contextual factors: (1) Sweden's empowerment-oriented educational system positions teachers with high autonomy but limited structural support for sustained improvement; (2) classroom norms position teachers as reactive helpers rather than proactive instructional leaders; and (3) invisible pedagogy, characterized by weak curriculum framing and implicit expectations, obscures core instructional work. When reforms introduced new roles, explicit instructional tools, structured collegial learning events, and routines that made pedagogy visible, teachers reported greater clarity, improved collaboration, and enhanced student learning outcomes. | **Scientific Research Findings** The study identifies three critical **implicit contextual factors** that shape the success of large-scale educational reforms in mathematics: 1. **Positioning of Teachers in the Educational System** – In Sweden, teachers have significant autonomy, but this can lead to variability in instruction and a lack of cohesive implementation of new policies. 2. **Positioning of Teachers in the Classroom** – Swedish teachers are often reactive rather than proactive, focusing on student-led learning without structured teacher guidance, which may hinder deep mathematical understanding. 3. **Traditions of Visible vs. Invisible Pedagogy** – Swedish education emphasizes "invisible pedagogy," with minimal explicit instruction and weak framing of learning objectives, making it difficult to align reforms with existing norms. Key empirical insights: - Existing professional development (PD) often fails to address these contextual factors. - Teachers appreciate structured tools (e.g., the **Five Practice Model**) when they empower rather than dictate practice. - Textbook reliance and weak lesson planning norms hinder systemic change. | **Scientific Research Findings** The study highlights critical contextual factors affecting mathematics instruction at scale, including: 1. The positioning of teachers within the educational system, which is influenced by empowerment versus control policies. 2. The role of teachers within the classroom, where proactive engagement in setting learning goals and orchestrating class discussions is essential. 3. Traditions of visible versus invisible pedagogy, influencing how educational strategies are implemented and perceived. |

Table 39: **Mathematics: "Educational policy to improve mathematics instruction at scale: Conceptualizing contextual factors (Ryve and Hemmi, 2019)".**

| (i) Gemma-27B-SFT | (ii) DeepSeek-V3 | (iii) GPT-4o |
|---|---|---|
| **Scientific Research Study Methods** The study applies educational design research over 2012–2017 across 38 schools (about 400 teachers and 10,000 students). Guided by Cobb and Jackson's learning design framework, researchers collected longitudinal teacher surveys, municipality-wide student assessments, textbook and syllabus analyses, classroom observations, and interviews with teachers and teacher educators. These data informed iterative refinements to professional development materials, new instructional tools, new organizational routines, and new teacher-support roles. | **Scientific Research Study Methods** The study employed a **mixed-methods approach** within the **"Count on Vasteras" (CoV) project (2012–2017)**, a large-scale reform effort in Sweden: - **Surveys & Interviews**: Pre-, mid-term, and post-intervention questionnaires from ~400 teachers to assess perceptions of PD and classroom practices. - **Student Assessments**: Yearly tests to measure mathematical proficiency across ~10,000 students. - **Textbook & Policy Analysis**: Examination of Swedish curriculum materials and teacher guides to identify gaps in support for teacher-led instruction. - **Iterative Design Research**: Adjustments to PD programs based on real-time feedback and contextual barriers. | **Scientific Research Study Methods** This research builds on theoretical frameworks and empirical studies conducted during the "Count on Vasteras" project, which involved collaboration with Swedish schools. Utilizing educational design research, the study iteratively developed strategies by analyzing teacher surveys, student math tests, and textbook content, alongside interviews and observational studies. This approach revealed patterns and deeper insights into the contextual factors impacting educational policy implementation. |
| **Policy Implications** 1. Embed new instructional leadership roles, such as Mathematics Mentors and Heads of Mathematics, in ways that align with Sweden's empowerment culture. 2. Introduce explicit instructional tools, lesson-planning supports, and formative assessment routines that make teacher work visible and strengthen proactive classroom leadership. 3. Preserve teacher autonomy while providing clearer learning goals, structured supports, and coherence across curriculum materials. 4. Require all large-scale reforms to begin with a systematic contextual analysis of teacher norms, classroom practices, and pedagogical traditions. 5. Build distributed instructional expertise through networks of school-level leaders who can adapt tools to local conditions and sustain reforms over time. | **Policy Implications** 1. **Adapt PD Programs to Teacher Autonomy** – Reform efforts should empower teachers by providing frameworks (e.g., goal-setting tools) rather than rigid scripts, respecting their professional agency while guiding instructional improvement. 2. **Strengthen Teacher Leadership** – Introduce roles like "Heads of Mathematics" to facilitate peer collaboration and bridge policy-practice gaps. 3. **Make Pedagogy More Visible** – Shift toward structured, explicit teaching strategies (e.g., problem-solving frameworks) while retaining student engagement elements. 4. **Context-Sensitive Policy Design** – Policymakers should analyze implicit cultural factors (e.g., teacher positioning, pedagogical norms) before scaling interventions. 5. **Foster Cross-System Learning** – Promote exchange between researchers, teachers, and policymakers to iteratively refine reforms based on local realities. | **Policy Implications** To improve mathematics instruction on a large scale, policymakers should: - Recognize and integrate the empowerment of teachers within policy frameworks, allowing them flexibility and agency in instruction while providing clear, supportive tools and strategies. - Facilitate the active engagement of teachers in the classroom by introducing frameworks that require strategic planning and the proactive orchestration of classroom activities. - Balance invisible pedagogical traditions with explicit instructional strategies that clearly define teacher roles and responsibilities. Implementing visible pedagogies can improve accountability and learning outcomes. |

**Analysis:**
Gemma-Gemma-27B-SFT provides the strongest policy brief. It offers a clearer synthesis of the three contextual factors, relates them directly to the mechanisms of instructional improvement, and presents actionable, context-aware policy recommendations aligned with the paper's evidence. DeepSeek-V3 is informative but more descriptive and diffuse, and GPT-4o is clear but omits several critical mechanisms. Overall, Gemma-27B-SFT achieves the best balance of accuracy, structure, and policy relevance.

Table 40: **Economics: "Financial inclusion, the shadow economy and economic growth in developing economies (Younas et al., 2022)".**

| (i) Gemma-27B-SFT | (ii) DeepSeek-V3 | (iii) GPT-4o |
|---|---|---|
| **Policy Problem**
Financial exclusion and widespread informal economic activity undermine sustainable economic growth in developing countries. Low access to formal banking, credit, and savings systems reduces investment and productivity, while a large shadow economy constrains revenue collection, weakens institutions, and limits the reach of macroeconomic policy. Policymakers require guidance on how financial inclusion and shadow-economy dynamics jointly shape growth prospects.

**Scientific Research Findings**
Using a panel of 30 developing economies (2008–2017), the study finds:

• Financial inclusion strongly and consistently increases GDP per capita across all econometric specifications.

• Shadow economy size has a large negative effect on GDP per capita in most regions.

• Bidirectional Granger causality exists between financial inclusion and economic growth, and between shadow economy size and economic growth, indicating a mutually reinforcing dynamic.

• These relationships vary in magnitude across regions, reflecting institutional and structural differences. | **Policy Problem**
Developing economies face significant challenges in achieving sustainable economic growth due to limited access to formal financial services and a large shadow economy (informal economic activities unreported to authorities). Financial exclusion and informal economic activities hinder productivity, tax revenues, and equitable development, exacerbating poverty and inequality. Policymakers need evidence-based strategies to promote financial inclusion and curb the shadow economy to foster inclusive growth.

**Scientific Research Findings**
The study analyzed data from 30 developing economies (2008–2017) and found:

• Financial inclusion (measured by access to banking services, credit, and savings) has a positive and statistically significant impact on economic growth (GDP per capita). A 1% increase in financial inclusion boosts growth by 15.9% on average.

• The shadow economy negatively affects growth: a 1% increase in informal economic activity reduces GDP per capita by 53.6%. Regions with larger shadow economies (e.g., Latin America, Sub-Saharan Africa) exhibited slower growth.

• Bidirectional causality exists: Economic growth also drives financial inclusion, while a larger shadow economy stifles formal financial sector development. | **Policy Problem**
Developing economies face a two-pronged challenge impacting their economic growth: low levels of financial inclusion and a substantial shadow economy. These factors collectively impede investments, resource allocation, and socio-economic advancements, crucial for sustainable development in these regions.

**Scientific Research Findings**
Recent research spanning from 2008 to 2017 illustrates that financial inclusion significantly enhances economic growth in developing economies. Conversely, the shadow economy detracts from economic performance, posing challenges for systemic integration into formal financial systems. Notably, there is a bidirectional causality between financial inclusion and economic growth, as well as between shadow economy activities and economic growth. |

Table 41: **Economics: "Financial inclusion, the shadow economy and economic growth in developing economies (Younas et al., 2022)".**

| (i) Gemma-27B-SFT | (ii) DeepSeek-V3 | (iii) GPT-4o |
|---|---|---|
| **Scientific Research Study Methods** The authors use harmonized panel data for 30 developing economies (2008–2017) and apply three complementary approaches:

• A fixed-effects panel OLS model to estimate short-run relationships while controlling for time-invariant heterogeneity.

• A two-step difference GMM estimator to address endogeneity, dynamic adjustment, and serial correlation.

• A Dumitrescu–Hurlin panel Granger causality test to examine directional relationships among financial inclusion, shadow economy size, and economic growth.

Data come from the World Development Indicators, IMF Financial Access Survey, and the Medina–Schneider shadow economy database.

**Policy Implications**

• **Expand financial inclusion**: Promote access to banking, credit, mobile money, and savings tools; reduce intermediation costs; and strengthen digital financial infrastructure.

• **Shrink the shadow economy**: Simplify regulatory requirements, improve tax enforcement, strengthen rule of law, and reduce corruption to incentivize formalization.

• **Coordinate reforms**: Because financial inclusion, shadow economy activity, and growth reinforce each other, policies should be integrated rather than isolated.

• **Tailor regional strategies**: Differences in institutional quality and economic structure require region-specific approaches. | **Scientific Research Study Methods** - Data Sources: World Bank Development Indicators, IMF Financial Access Survey, and Medina & Schneider's shadow economy database. - Methodologies: - Panel regression models (fixed effects) to isolate regional impacts. - Dynamic panel analysis (Arellano-Bond GMM) to address endogeneity and lagged effects. - Granger causality tests to confirm bidirectional relationships. - Variables: GDP per capita (dependent), financial inclusion index, shadow economy size (% of GDP), and controls (inflation, trade, unemployment).

**Policy Implications**
1. **Expand Financial Inclusion**: - Digital financial services: Leverage mobile banking and fintech to reach unbanked populations. - Affordable credit: Support microfinance and SME lending to integrate informal businesses into the formal sector. - Financial literacy programs: Educate citizens on using formal financial tools.
2. **Reduce the Shadow Economy**: - Simplify tax systems: Lower compliance burdens to incentivize formalization. - Strengthen governance: Combat corruption and improve transparency to rebuild trust in institutions. - Labor market reforms: Provide social protections to reduce reliance on informal work.
3. **Regional Tailoring**: Policies should address regional disparities (e.g., higher shadow economies in Latin America vs. lower financial inclusion in Sub-Saharan Africa). | **Scientific Research Study Methods** The study utilized data from the World Development Indicators, IMF's Financial Access Survey, and Medina and Schneider's global database across 30 developing economies. Employing panel data analysis, the researchers conducted a panel ordinary least square (OLS) fixed effect, a two-step differenced generalized method of moments (GMM), and panel Granger causality approaches to derive robust findings.

**Policy Implications** These insights advocate for policies that enhance financial inclusion through increased access to banking services, credit facilities, and financial literacy programs. Simultaneously, strategies to shrink the shadow economy, such as improving regulatory frameworks and reducing corruption, are critical. By doing so, developing economies can foster economic stability and growth. This dual approach not only addresses immediate financial access challenges but also strengthens long-term economic structures, helping transitioning economies achieve broader development objectives. |

**Analysis:**
Gemma-27B-SFT produces the strongest policy brief. It offers the clearest causal explanation, integrates all empirical findings, and presents the most actionable and region-sensitive policy recommendations. DeepSeek-V3 is comprehensive but more verbose, and GPT-4o is concise but omits methodological and regional detail. Gemma-27B-SFT achieves the best balance between clarity, completeness, and policy relevance.

Table 42: **Humanities: "Science cited in policy documents: Evidence from the Overton database (Fang et al., 2024)".**

| (i) Gemma-27B-SFT | (ii) DeepSeek-V3 | (iii) GPT-4o |
|---|---|---|
| **Policy Problem**
Despite growing calls for evidence-based policymaking, it remains unclear how extensively scientific research is used in policy documents and which fields, publication types, and research topics policymakers rely on most. Without a systematic understanding of this uptake, governments and research systems cannot assess the real-world influence of science, allocate funding effectively, or design strategies that increase the policy relevance of research outputs.

**Scientific Research Findings**
Matching 18.3 million Web of Science publications (2010–2019) with Overton's global database of policy documents shows that only 3.9% of publications are cited at least once in policy texts. Policy citations are delayed: publications from 2010 have 6.47% coverage, while 2019 publications show only 0.78%. Reviews (8.2% coverage) and articles (4.4%) attract far more policy attention than other document types. Across fields, Social Sciences and Humanities lead (12.34% coverage), followed by Life and Earth Sciences (5.96%) and Biomedical and Health Sciences (5.72%). Physical Sciences and Engineering (0.62%) and Mathematics/Computer Science (0.93%) are seldom cited. Within each major field, policy attention is uneven at the micro-field level, indicating selective relevance of specific research topics. | **Policy Problem**
Bridging Science and Policy: Insights from the Overton Database on Research Utilization in Policymaking The translation of scientific research into actionable policy is a critical challenge for governments and organizations. Policymakers often struggle to identify and integrate relevant scientific evidence into decision-making due to the sheer volume of research and disparities in its policy relevance. This study addresses the gap in understanding how often and in what ways scientific publications are cited in policy documents, revealing patterns that could improve evidence-based policymaking.

**Scientific Research Findings**
- Only 3.9% of 18 million scientific publications (2010–2019) were cited in policy documents, with an average of 0.09 citations per paper. - Policy citations show a time lag: Older publications (e.g., from 2010) are 8 times more likely to be cited than recent ones (e.g., 2019). - Document types matter: Reviews (8%) and articles (4.4%) are most cited, suggesting policymakers favor synthesized or comprehensive research. - Field disparities: - Social Sciences and Humanities (SSH): Highest policy engagement (12.3% cited). - Life/Earth Sciences (LES) and Biomedical/Health Sciences (BHS): Moderate engagement ( 6% cited). - Physical Sciences/Engineering (PSE) and Mathematics/Computer Science (MCS): Low engagement (<1% cited). | **Policy Problem**
Utilizing Scientific Research in Policy: Examining Evidence from the Overton Database The intricate relationship between scientific research and policy development presents a societal challenge: ensuring that scientific findings effectively inform policy decisions. This issue closely relates to understanding how scientific publications are utilized within policy documentation, a key step in enhancing evidence-based policymaking.

**Scientific Research Findings**
The study indicates a notably limited use of scientific publications in policy documents, with only 3.9% of the 18 million analyzed publications being cited at least once. Social Sciences and Humanities (SSH) research received the most citations from policy documents, with 12.34% coverage, followed by Life and Earth Sciences and Biomedical and Health Sciences. Reviews and articles are most frequently cited, suggesting a tendency towards comprehensive and synthesized research outputs. |

Table 43: **Humanities: "Science cited in policy documents: Evidence from the Overton database (Fang et al., 2024)".**

| (i) Gemma-27B-SFT | (ii) DeepSeek-V3 | (iii) GPT-4o |
|---|---|---|
| **Scientific Research Study Methods** The study integrates 18.3 million WoS publications with Overton's global database of policy documents using DOI matching (June 2020 snapshot). Subject classification is performed with the CWTS publication-level system, mapping papers to 4,013 micro-fields and five macro-fields. Three indicators quantify policy uptake: Coverage (share cited), Density (citations per publication), and Intensity (citations per cited publication). VOSviewer visualizations highlight topic clusters receiving disproportionate policy attention. | **Scientific Research Study Methods** - Data source: Analyzed 18.3 million Web of Science publications (2010–2019) matched with policy citations from Overton, a global policy document database. - Metrics: Measured policy impact using: - Coverage (% of papers cited), - Density (average citations per paper), - Intensity (average citations per cited paper). - Field classification: Used the CWTS/Leiden Ranking system to categorize publications into five broad fields (e.g., SSH, BHS). - Visualization: Employed VOSviewer to map citation patterns across research topics. | **Scientific Research Study Methods** This analysis utilized the Overton database, the world's largest assembly of policy documents, to track citations of scientific publications indexed by the Web of Science from 2010 to 2019. The researchers employed bibliometric analysis and visualization tools like VOSviewer to assess citation patterns, focusing on coverage, density, and intensity of policy document citations across publication years, document types, and scientific fields. |
| **Policy Implications** • Funding strategies should consider the strong policy pull of SSH, BHS and LES relative to PSE and MCS. • Reviews and articles dominate policy citations, indicating that policymakers rely heavily on synthetized and peer-reviewed evidence. • Policy citation metrics accumulate slowly; evaluation frameworks should use longer assessment windows for policy impact. • Comprehensive policy-document indexes like Overton are essential infrastructure; broader coverage and openness would enhance research-policy tracking. • Field-level inequalities suggest the need for targeted knowledge-translation initiatives to support underrepresented disciplines in reaching policy audiences. | **Policy Implications** 1. Targeted Research Funding: Prioritize funding for SSH, LES, and BHS fields, which show higher policy relevance, but ensure interdisciplinary integration to address complex societal challenges. 2. Timeliness of Evidence: Policymakers should account for the citation delay (3+ years) when seeking evidence, or invest in systems to flag emerging high-impact research. 3. Support for Review Articles: Encourage synthesis of research (e.g., systematic reviews) to bridge gaps between academia and policy, as these are highly cited. 4. Database Improvements: Expand policy citation tracking to include non-DOI publications and non-English research to reduce bias in visibility. 5. Evaluation Metrics: Incorporate policy citations (e.g., via Overton or Altmetric) into research assessment frameworks to incentivize policy-relevant science. | **Policy Implications** The findings highlight potential strategies to enhance the utilization of scientific research in policymaking. Emphasizing review articles and fostering interdisciplinary research, particularly in areas such as SSH and environmental sciences, could increase policy engagement. Policymakers should consider these fields' relevance and contribute to reducing the delay in the uptake of recent scientific findings. Furthermore, addressing the underrepresentation of certain fields in policy documents can optimize research impact and foster a more balanced approach to evidence-based policy development. |

**Analysis:**
Gemma-27B-SFT provides the most policy-ready and analytically complete brief. It incorporates specific numerical findings, interprets field-level disparities, highlights micro-field variation, and links evidence to clear policy levers. DeepSeek-V3 is detailed but more narrative and less structured, while GPT-4o offers a concise overview but omits several central quantitative insights. Overall, Gemma-27B-SFT delivers the strongest balance of accuracy, completeness, and actionable relevance.

## J  DETAILS OF 85 EXPERT-WRITTEN PAPER-BRIEF PAIRS

We provide the detailed list of 85 pairs in Appendix J.1 and their yearly distribution in Appendix J.2. We also provide the detailed list of 3 new pairs for in-context polishing in Appendix J.4, and 8 new pairs for Task 15 in Appendix J.3.

### J.1  LIST OF 85 EXPERT-WRITTEN PAPER-BRIEF PAIRS

We document the 85 expert-written scientific paper-policy brief pairs included in Sci2Pol-Bench, citing each source to ensure transparency and reproducibility.

1. Paper (Link et al., 2025a) (DOI: 10.1038/s41560-025-01722-y); Brief (Link et al., 2025b) (DOI: 10.1038/s41560-025-01741-9); Nature Energy; Accessed: 2025-05-10.
2. Paper (Shears et al., 2025a) (DOI: 10.1038/s41560-025-01724-w); Brief (Shears et al., 2025b) (DOI: 10.1038/s41560-025-01725-9); Nature Energy; Accessed: 2025-05-10.
3. Paper (Navia Simon and Diaz Anadon, 2025b) (DOI: 10.1038/s41560-025-01704-0); Brief (Navia Simon and Diaz Anadon, 2025a) (DOI: 10.1038/s41560-025-01715-x); Nature Energy; Accessed: 2025-05-10.
4. Paper (van Heerden et al., 2025a) (DOI: 10.1038/s41560-025-01703-1); Brief (van Heerden et al., 2025b) (DOI: 10.1038/s41560-025-01721-z); Nature Energy; Accessed: 2025-05-10.
5. Paper (Millinger et al., 2025b) (DOI: 10.1038/s41560-024-01693-6); Brief (Millinger et al., 2025a) (DOI: 10.1038/s41560-024-01685-6); Nature Energy; Accessed: 2025-05-10.
6. Paper (Odenweller and Ueckerdt, 2025b) (DOI: 10.1038/s41560-024-01684-7); Brief (Odenweller and Ueckerdt, 2025a) (DOI: 10.1038/s41560-024-01682-9); Nature Energy; Accessed: 2025-05-10.
7. Paper (Caggiano et al., 2024b) (DOI: 10.1038/s41560-024-01603-w); Brief (Caggiano et al., 2024a) (DOI: 10.1038/s41560-024-01585-9); Nature Energy; Accessed: 2025-05-10.
8. Paper (O'Shaughnessy et al., 2024b) (DOI: 10.1038/s41560-024-01546-2); Brief (O'Shaughnessy et al., 2024a) (DOI: 10.1038/s41560-024-01575-x); Nature Energy; Accessed: 2025-05-10.
9. Paper (Kennedy et al., 2024a) (DOI: 10.1038/s41560-024-01530-w); Brief (Kennedy et al., 2024b) (DOI: 10.1038/s41560-024-01554-2); Nature Energy; Accessed: 2025-05-10.
10. Paper (Sitarz et al., 2024a) (DOI: 10.1038/s41560-024-01505-x); Brief (Sitarz et al., 2024b) (DOI: 10.1038/s41560-024-01545-3); Nature Energy; Accessed: 2025-05-10.
11. Paper (Link et al., 2024b) (DOI: 10.1038/s41560-024-01531-9); Brief (Link et al., 2024a) (DOI: 10.1038/s41560-024-01555-1); Nature Energy; Accessed: 2025-05-10.
12. Paper (White et al., 2024a) (DOI: 10.1038/s41560-023-01422-5); Brief (White et al., 2024b) (DOI: 10.1038/s41560-023-01433-2); Nature Energy; Accessed: 2025-05-10.
13. Paper (Wolske et al., 2023b) (DOI: 10.1038/s41560-023-01298-5); Brief (Wolske et al., 2023a) (DOI: 10.1038/s41560-023-01323-7); Nature Energy; Accessed: 2025-05-10.
14. Paper (Deshmukh et al., 2023a) (DOI: 10.1038/s41560-023-01259-y); Brief (Deshmukh et al., 2023b) (DOI: 10.1038/s41560-023-01273-0); Nature Energy; Accessed: 2025-05-10.
15. Paper (Gars et al., 2022a) (DOI: 10.1038/s41560-022-01122-6); Brief (Gars et al., 2022b) (DOI: 10.1038/s41560-022-01125-3); Nature Energy; Accessed: 2025-05-10.
16. Paper (Gruber et al., 2022a) (DOI: 10.1038/s41560-022-00994-y); Brief (Gruber et al., 2022b) (DOI: 10.1038/s41560-022-01031-8); Nature Energy; Accessed: 2025-05-10.
17. Paper (Longden et al., 2022a) (DOI: 10.1038/s41560-021-00942-2); Brief (Longden et al., 2022b) (DOI: 10.1038/s41560-021-00968-6); Nature Energy; Accessed: 2025-05-10.
18. Paper (Pachauri et al., 2021a) (DOI: 10.1038/s41560-021-00911-9); Brief (Pachauri et al., 2021b) (DOI: 10.1038/s41560-021-00939-x); Nature Energy; Accessed: 2025-05-10.
19. Paper (Hall et al., 2021b) (DOI: 10.1038/s41560-021-00781-1); Brief (Hall et al., 2021a) (DOI: 10.1038/s41560-021-00821-w); Nature Energy; Accessed: 2025-05-10.
20. Paper (Kitzing et al., 2020a) (DOI: 10.1038/s41560-020-00717-1); Brief (Kitzing et al., 2020b) (DOI: 10.1038/s41560-020-00733-1); Nature Energy; Accessed: 2025-05-10.
21. Paper (Bonan et al., 2020b) (DOI: 10.1038/s41560-020-00719-z); Brief (Bonan et al., 2020a) (DOI: 10.1038/s41560-020-00727-z); Nature Energy; Accessed: 2025-05-10.
22. Paper (Goldstein et al., 2020a) (DOI: 10.1038/s41560-020-00683-8); Brief (Goldstein et al., 2020b) (DOI: 10.1038/s41560-020-00691-8); Nature Energy; Accessed: 2025-05-10.
23. Paper (Mani et al., 2020a) (DOI: 10.1038/s41560-020-0596-7); Brief (Mani et al., 2020b) (DOI: 10.1038/s41560-020-0635-4); Nature Energy; Accessed: 2025-05-10.

24. Paper (Casey et al., 2020b) (DOI: 10.1038/s41560-020-0600-2); Brief (Casey et al., 2020a) (DOI: 10.1038/s41560-020-0622-9); Nature Energy; Accessed: 2025-05-10.

25. Paper (Kontokosta et al., 2020a) (DOI: 10.1038/s41560-020-0589-6); Brief (Kontokosta et al., 2020b) (DOI: 10.1038/s41560-020-0603-z); Nature Energy; Accessed: 2025-05-10.

26. Paper (Kaufmann and Connelly, 2020b) (DOI: 10.1038/s41560-020-0549-1); Brief (Kaufmann and Connelly, 2020a) (DOI: 10.1038/s41560-020-0563-3); Nature Energy; Accessed: 2025-05-10.

27. Paper (Braunholtz-Speight et al., 2020a) (DOI: 10.1038/s41560-019-0546-4); Brief (Braunholtz-Speight et al., 2020b) (DOI: 10.1038/s41560-020-0556-2); Nature Energy; Accessed: 2025-05-10.

28. Paper (Kar et al., 2019) (DOI: 10.1038/s41560-019-0429-8); Brief (Kar et al., 2020) (DOI: 10.1038/s41560-019-0536-6); Nature Energy; Accessed: 2025-05-10.

29. Paper (Liu and Rajagopal, 2019) (DOI: 10.1038/s41560-019-0430-2); Brief (Rajagopal and Liu, 2020) (DOI: 10.1038/s41560-019-0532-x); Nature Energy; Accessed: 2025-05-10.

30. Paper (White and Sintov, 2020a) (DOI: 10.1038/s41560-019-0507-y); Brief (White and Sintov, 2020b) (DOI: 10.1038/s41560-019-0515-y); Nature Energy; Accessed: 2025-05-10.

31. Paper (Rinscheid and Wüstenhagen, 2019b) (DOI: 10.1038/s41560-019-0460-9); Brief (Rinscheid and Wüstenhagen, 2019a) (DOI: 10.1038/s41560-019-0509-9); Nature Energy; Accessed: 2025-05-10.

32. Paper (Mays et al., 2019a) (DOI: 10.1038/s41560-019-0476-1); Brief (Mays et al., 2019b) (DOI: 10.1038/s41560-019-0502-3); Nature Energy; Accessed: 2025-05-10.

33. Paper (Egli et al., 2018) (DOI: 10.1038/s41560-018-0277-y); Brief (Egli et al., 2019) (DOI: 10.1038/s41560-019-0482-3); Nature Energy; Accessed: 2025-05-10.

34. Paper (Tiefenbeck et al., 2019b) (DOI: 10.1038/s41560-018-0282-1); Brief (Tiefenbeck et al., 2019a) (DOI: 10.1038/s41560-019-0480-5); Nature Energy; Accessed: 2025-05-10.

35. Paper (Azarova et al., 2018) (DOI: 10.1038/s41560-018-0105-4); Brief (Azarova et al., 2019) (DOI: 10.1038/s41560-019-0479-y); Nature Energy; Accessed: 2025-05-10.

36. Paper (Apostoleris et al., 2018) (DOI: 10.1038/s41560-018-0256-3); Brief (Apostoleris et al., 2019) (DOI: 10.1038/s41560-019-0481-4); Nature Energy; Accessed: 2025-05-10.

37. Paper (Mahdavi et al., 2025a) (DOI: 10.1038/s41558-025-02283-4); Brief (Mahdavi et al., 2025b) (DOI: 10.1038/s41558-025-02304-2); Nature Climate Change; Accessed: 2025-05-10.

38. Paper (Ogier et al., 2025b) (DOI: 10.1038/s41558-025-02291-4); Brief (Ogier et al., 2025a) (DOI: 10.1038/s41558-025-02292-3); Nature Climate Change; Accessed: 2025-05-10.

39. Paper (Liu et al., 2025b) (DOI: 10.1038/s41558-024-02237-2); Brief (Liu et al., 2025c) (DOI: 10.1038/s41558-024-02240-7); Nature Climate Change; Accessed: 2025-05-10.

40. Paper (Tang et al., 2024a) (DOI: 10.1038/s41558-024-02162-4); Brief (Tang et al., 2024b) (DOI: 10.1038/s41558-024-02145-5); Nature Climate Change; Accessed: 2025-05-10.

41. Paper (Druckenmiller et al., 2024a) (DOI: 10.1038/s41558-024-02082-3); Brief (Druckenmiller et al., 2024b) (DOI: 10.1038/s41558-024-02083-2); Nature Climate Change; Accessed: 2025-05-10.

42. Paper (Nowak et al., 2024b) (DOI: 10.1038/s41558-024-02054-7); Brief (Nowak et al., 2024a) (DOI: 10.1038/s41558-024-02055-6); Nature Climate Change; Accessed: 2025-05-10.

43. Paper (Lamb et al., 2024a) (DOI: 10.1038/s41558-024-01984-6); Brief (Lamb et al., 2024b) (DOI: 10.1038/s41558-024-01993-5); Nature Climate Change; Accessed: 2025-05-10.

44. Paper (Gasparini et al., 2024b) (DOI: 10.1038/s41558-024-01972-w); Brief (Gasparini et al., 2024a) (DOI: 10.1038/s41558-024-01959-7); Nature Climate Change; Accessed: 2025-05-10.

45. Paper (Duan et al., 2024a) (DOI: 10.1038/s41558-024-01952-0); Brief (Duan et al., 2024b) (DOI: 10.1038/s41558-024-01962-y); Nature Climate Change; Accessed: 2025-05-10.

46. Paper (Andreoni et al., 2024b) (DOI: 10.1038/s41558-023-01870-7); Brief (Andreoni et al., 2024a) (DOI: 10.1038/s41558-023-01871-6); Nature Climate Change; Accessed: 2025-05-10.

47. Paper (Linsenmeier et al., 2023a) (DOI: 10.1038/s41558-023-01710-8); Brief (Linsenmeier et al., 2023b) (DOI: 10.1038/s41558-023-01700-w); Nature Climate Change; Accessed: 2025-05-10.

48. Paper (Merfort et al., 2023a) (DOI: 10.1038/s41558-023-01697-2); Brief (Merfort et al., 2023b) (DOI: 10.1038/s41558-023-01711-7); Nature Climate Change; Accessed: 2025-05-10.

49. Paper (Cerf et al., 2023a) (DOI: 10.1038/s41558-023-01679-4); Brief (Cerf et al., 2023b) (DOI: 10.1038/s41558-023-01677-6); Nature Climate Change; Accessed: 2025-05-10.

50. Paper (Buck et al., 2023b) (DOI: 10.1038/s41558-022-01592-2); Brief (Buck et al., 2023a) (DOI: 10.1038/s41558-023-01614-7); Nature Climate Change; Accessed: 2025-05-10.

51. Paper (Harring et al., 2023a) (DOI: 10.1038/s41558-023-01597-5); Brief (Harring et al., 2023b) (DOI: 10.1038/s41558-023-01609-4); Nature Climate Change; Accessed: 2025-05-10.

52. Paper (Basheer et al., 2023a) (DOI: 10.1038/s41558-022-01556-6); Brief (Basheer et al., 2023b) (DOI: 10.1038/s41558-022-01557-5); Nature Climate Change; Accessed: 2025-05-10.

53. Paper (Iyer et al., 2022b) (DOI: 10.1038/s41558-022-01508-0); Brief (Iyer et al., 2022a) (DOI: 10.1038/s41558-022-01517-z); Nature Climate Change; Accessed: 2025-05-10.

54. Paper (de Ruig et al., 2022b) (DOI: 10.1038/s41558-022-01501-7); Brief (de Ruig et al., 2022a) (DOI: 10.1038/s41558-022-01502-6); Nature Climate Change; Accessed: 2025-05-10.

55. Paper (Bjørn et al., 2022a) (DOI: 10.1038/s41558-022-01379-5); Brief (Bjørn et al., 2022b) (DOI: 10.1038/s41558-022-01385-7); Nature Climate Change; Accessed: 2025-05-10.

56. Paper (Mildenberger et al., 2022a) (DOI: 10.1038/s41558-021-01268-3); Brief (Mildenberger et al., 2022b) (DOI: 10.1038/s41558-021-01270-9); Nature Climate Change; Accessed: 2025-05-10.

57. Paper (Budolfson et al., 2021a) (DOI: 10.1038/s41558-021-01217-0); Brief (Budolfson et al., 2021b) (DOI: 10.1038/s41558-021-01228-x); Nature Climate Change; Accessed: 2025-05-10.

58. Paper (Janssens et al., 2020) (DOI: 10.1038/s41558-020-0847-4); Brief (Janssens et al., 2021) (DOI: 10.1038/s41558-021-01201-8); Nature Climate Change; Accessed: 2025-05-10.

59. Paper (Moffette et al., 2021b) (DOI: 10.1038/s41558-020-00956-w); Brief (Moffette et al., 2021a) (DOI: 10.1038/s41558-021-01195-3); Nature Climate Change; Accessed: 2025-05-10.

60. Paper (Bechtel et al., 2020) (DOI: 10.1038/s41558-020-00914-6); Brief (Bechtel et al., 2021) (DOI: 10.1038/s41558-021-01202-7); Nature Climate Change; Accessed: 2025-05-10.

61. Paper (Peng et al., 2021b) (DOI: 10.1038/s41558-021-01128-0); Brief (Peng et al., 2021a) (DOI: 10.1038/s41558-021-01193-5); Nature Climate Change; Accessed: 2025-05-10.

62. Paper (Tran et al., 2024) (DOI: 10.1038/s44284-024-00116-7); Brief (Ivanov et al., 2024) (DOI: 10.1038/s44284-024-00128-3); Nature Cities; Accessed: 2025-05-10.

63. Paper (Diezmartínez et al., 2024b) (DOI: 10.1038/s44284-024-00121-w); Brief (Diezmartínez et al., 2024a) (DOI: 10.1038/s44284-024-00129-2); Nature Cities; Accessed: 2025-05-10.

64. Paper (Mollborn et al., 2025a) (DOI: 10.1177/00221465241255946); Brief (Mollborn et al., 2025b) (DOI: 10.1177/00221465251315281); Journal Of Health And Social Behavior; Accessed: 2025-05-10.

65. Paper (Moinester and Stanhope, 2024a) (DOI: 0.1177/00221465241230839); Brief (Moinester and Stanhope, 2024b) (DOI: 10.1177/00221465241269117); Journal Of Health And Social Behavior; Accessed: 2025-05-10.

66. Paper (Han et al., 2024a) (DOI: 10.1177/00221465231205266); Brief (Han et al., 2024b) (DOI: 10.1177/00221465241248972); Journal Of Health And Social Behavior; Accessed: 2025-05-10.

67. Paper (Dore et al., 2024a) (DOI: 10.1177/00221465231194043); Brief (Dore et al., 2024b) (DOI: 10.1177/00221465241226808); Journal Of Health And Social Behavior; Accessed: 2025-05-10.

68. Paper (McFarland et al., 2023a) (DOI: 10.1177/00221465221109202); Brief (McFarland et al., 2023b) (DOI: 10.1177/00221465221150307); Journal Of Health And Social Behavior; Accessed: 2025-05-10.

69. Paper (Masters et al., 2023a) (DOI: 10.1177/00221465231165284); Brief (Masters et al., 2023b) (DOI: 10.1177/00221465231171627); Journal Of Health And Social Behavior; Accessed: 2025-05-10.

70. Paper (Parbst and Wheaton, 2023a) (DOI: 10.1177/00221465231166334); Brief (Parbst and Wheaton, 2023b) (DOI: 10.1177/00221465231190977); Journal Of Health And Social Behavior; Accessed: 2025-05-10.

71. Paper (Czarnecki et al., 2023a) (DOI: 10.1177/00221465231172177); Brief (Czarnecki et al., 2023b) (DOI: 10.1177/00221465231209380); Journal Of Health And Social Behavior; Accessed: 2025-05-10.

72. Paper (McCabe, 2022a) (DOI: 10.1177/00221465211058152); Brief (McCabe, 2022b) (DOI: 10.1177/00221465221097453); Journal Of Health And Social Behavior; Accessed: 2025-05-10.

73. Paper (Vuolo et al., 2022a) (DOI: 10.1177/00221465211067209); Brief (Vuolo et al., 2022b) (DOI: 10.1177/00221465221112986); Journal Of Health And Social Behavior; Accessed: 2025-05-10.

74. Paper (Anderson and Ray-Warren, 2022a) (DOI: 10.1177/00221465221074915); Brief (Anderson and Ray-Warren, 2022b) (DOI: 10.1177/00221465221130917); Journal Of Health And Social Behavior; Accessed: 2025-05-10.

75. Paper (Augustine, 2021a) (DOI: 10.1177/0022146520979664); Brief (Augustine, 2021b) (DOI: 10.1177/0022146520986008); Journal Of Health And Social Behavior; Accessed: 2025-05-10.

76. Paper (Manzer and Bell, 2021a) (DOI: 10.1177/00221465211003232); Brief (Manzer and Bell, 2021b) (DOI: 10.1177/00221465211008328); Journal Of Health And Social Behavior; Accessed: 2025-05-10.

77. Paper (Berg et al., 2021a) (DOI: 10.1177/00221465211052568); Brief (Berg et al., 2021b) (DOI: 10.1177/00221465211055925); Journal Of Health And Social Behavior; Accessed: 2025-05-10.
78. Paper (Schnittker and Do, 2020a) (DOI: 10.1177/0022146519899115); Brief (Schnittker and Do, 2020b) (DOI: 10.1177/0022146520903969); Journal Of Health And Social Behavior; Accessed: 2025-05-10.
79. Paper (Owens, 2020a) (DOI: 10.1177/0022146520924810); Brief (Owens, 2020b) (DOI: 10.1177/0022146520926100); Journal Of Health And Social Behavior; Accessed: 2025-05-10.
80. Paper (Thombs et al., 2020a) (DOI: 10.1177/0022146520939514); Brief (Thombs et al., 2020b) (DOI: 10.1177/0022146520945607); Journal Of Health And Social Behavior; Accessed: 2025-05-10.
81. Paper (Bierman and Schieman, 2020a) (DOI: 10.1177/0022146520970190); Brief (Bierman and Schieman, 2020b) (DOI: 10.1177/0022146520968770); Journal Of Health And Social Behavior; Accessed: 2025-05-10.
82. Paper (Zhang et al., 2025b) (DOI: 10.1038/s41893-025-01533-9); Brief (Zhang et al., 2025a) (DOI: 10.1038/s41893-025-01535-7); Nature Sustainability; Accessed: 2025-05-10.
83. Paper (Azar, 2024a) (DOI: 10.1177/00221465241265435); Brief (Azar, 2024b) (DOI: 10.1177/00221465241291690); Journal Of Health And Social Behavior; Accessed: 2025-05-10.
84. Paper (Rapp et al., 2022a) (DOI: 10.1177/00221465211058153); Brief (Rapp et al., 2022b) (DOI: 10.1177/00221465211073836); Journal Of Health And Social Behavior; Accessed: 2025-05-10.
85. Paper (Ci et al., 2024b) (DOI: 10.1038/s41893-024-01471-y); Brief (Ci et al., 2024a) (DOI: 10.1038/s41893-024-01472-x); Nature Sustainability; Accessed: 2025-05-10.

### J.2 PUBLICATION YEAR DISTRIBUTION OF 85 POLICY BRIEFS

As shown in Table 44, the number of policy briefs grows steadily. This trend highlights the increasing availability of high-quality published data. This suggests strong potential for expanding Sci2Pol-Bench and Sci2Pol-Corpus in the future.

Table 44: **Publication Year Distribution of the 85 Expert-written Policy Briefs.**

| Year | 2019 | 2020 | 2021 | 2022 | 2023 | 2024 | 2025 |
|------|------|------|------|------|------|------|------|
| **Num. of Pairs** | 6 | 15 | 10 | 11 | 12 | 20 | 11 |

### J.3 LIST OF 8 NEW EXPERT-WRITTEN PAPER-BRIEF PAIRS

We document the new 8 expert-written scientific paper-policy brief pairs for Task 15.

1. Paper (Weber et al., 2025b) (DOI: 10.1038/s41893-025-01676-9); Brief (Weber et al., 2025a) (DOI: 10.1038/s41893-025-01680-z); Nature Sustainability; Accessed: 2025-05-10.
2. Paper (Minten et al., 2025b) (DOI: 10.1038/s41893-025-01618-5); Brief (Minten et al., 2025a) (DOI: 10.1038/s41893-025-01622-9); Nature Sustainability; Accessed: 2025-05-10.
3. Paper (Vivier et al., 2025b) (DOI: 10.1038/s41558-025-02348-4); Brief (Vivier et al., 2025a) (DOI: 10.1038/s41558-025-02342-w); Nature Climate Change; Accessed: 2025-05-10.
4. Paper (Cai et al., 2025) (DOI: 10.1038/s41893-025-01560-6); Brief (Wang et al., 2025a) (DOI: 10.1038/s41893-025-01582-0); Nature Sustainability; Accessed: 2025-05-10.
5. Paper (Hermansen et al., 2025a) (DOI: 10.1038/s41586-025-09259-6); Brief (Hermansen et al., 2025b) (DOI: 10.1038/d41586-025-02546-2); Nature; Accessed: 2025-05-10.
6. Paper (Sun et al., 2025b) (DOI: 10.1038/s41560-025-01821-w); Brief (Sun et al., 2025a) (DOI: 10.1038/s41560-025-01822-9); Nature Energy; Accessed: 2025-05-10.
7. Paper (Anshassi and Townsend, 2025b) (DOI: 10.1038/s41893-025-01607-8); Brief (Anshassi and Townsend, 2025a) (DOI: 10.1038/s41893-025-01597-7); Nature Sustainability; Accessed: 2025-05-10.
8. Paper (Dipoppa and Gulzar, 2024) (DOI: 10.1038/s41586-024-08046-z); Brief (Dipoppa and Gulzar, 2025) (DOI: 10.1038/d41586-025-02545-3); Nature; Accessed: 2025-05-10.

### J.4 LIST OF 3 NEW IN-CONTEXT EXPERT-WRITTEN PAPER-BRIEF PAIRS

We document the new 3 expert-written scientific paper-policy brief pairs for in-context polishing.

1. Paper (Mehta et al., 2025a) (DOI: 10.1038/s41893-025-01584-y); Brief (Mehta et al., 2025b) (DOI: 10.1038/s41893-025-01593-x); Nature Sustainability; Accessed: 2025-05-10.
2. Paper (Wang et al., 2025c) (DOI: 10.1038/s41558-025-02439-2); Brief (Wang et al., 2025b) (DOI: 10.1038/s41558-025-02418-7); Nature Climate Change; Accessed: 2025-05-10.
3. Paper (Ye et al., 2025a) (DOI: 10.1038/s41560-025-01845-2); Brief (Ye et al., 2025b) (DOI: 10.1038/s41560-025-01840-7); Nature Energy; Accessed: 2025-05-10.

## K  EXAMPLES FOR TASKS 1-18

In this section, we provide examples for Tasks 1-18, as show in Tables 45 to 62.

Table 45: **Example for Task 1: Scientific Text Autocompletion.**

---

**Prompt:** You are given the start of a paragraph from a scientific research paper, ending in "...". Choose the most coherent continuation from the options below.

Query:
We do not find an association between exposure to climate hazards and re-risking. Instead, we find that de-risking is positively and significantly correlated with higher exposure to physical climate risks. ...

Options:
A. This is puzzling and requires further analysis.
B. Similarly, a large financial sector increases the direct exposure of an economy to stranded asset risks, which could threaten financial stability.
C. It follows that if a country is highly exposed to physical climate risks, it would adopt these practices.
D. In fact, prior research has demonstrated a correlation between physical risks and central bank management of climate risks.
E. We measure climate hazard exposure by using the exposure component of the Notre Dame Global Adaptation Initiative (ND-Gain) country index.

Answer with the letter (A-E) corresponding to the best continuation. **Strictly follow this format– do not include any explanations or additional text.**

---

**Answer:** A

---

Table 46: **Example for Task 2: Political Text Autocompletion.**

---

**Prompt:**
You are given the start of a paragraph from a policy brief, ending in "...". Choose the most coherent continuation from the options below.

Query:
The rapid diffusion of battery electric vehicles, in addition to the decarbonization of the energy sector—requires an increasing number of batteries. However, the EU's goal to cover 90% of its battery demand from domestic production by 2030 is at risk, as projected demand will likely exceed 1.0 TWh per year and outpace production capacity despite highly ambitious growth rates. ...

Options:
A. If Europe fails to scale up production, it may face severe economic and geopolitical risks, due to increased dependence on external suppliers, weakened industrial competitiveness and potential for delayed decarbonization.
B. An urgent question is therefore whether Europe can realistically meet its future battery demand through domestic production, and what policy actions are needed to ensure success.
C. We find that European battery cell demand will likely surpass 1.0 TWh per year by 2030, whereas domestic production capacity is expected to fall short, creating a risk of supply constraints.
D. Although Europe can be expected to meet at least 50-60% of its demand through domestic production by 2030, achieving the EU's 90% self-sufficiency target is feasible but uncertain, as nearly half of our modelled scenarios fail to meet this target (Fig. 1).
E. If Europe wants more independence from battery cell imports, our findings highlight the urgency of accelerating production capacity expansion, scaling up a battery supply chain, and implementing strong industrial policies to support competitiveness and supply sovereignty.

Answer with the letter (A-E) corresponding to the best continuation. **Strictly follow this format–do not include any explanations or additional text.**

---

**Answer:** B

---

Table 47: **Example for Task 3: Scientific Sentence Reordering.**

---

**Prompt:**
You are given three shuffled sentences that originally formed a coherent paragraph from a scientific research paper. Your task is to determine the correct order by selecting the most logical and coherent sequence.

Shuffled Sentences:
A. Other key markets, such as the United States and China, have also set ambitious ZEV targets from the 2030s.
B. Accordingly, BEVs prevail in the future portfolios of car manufacturers and several European countries will enforce 100% zero-emission vehicle (ZEV) sales for cars by at least 2035, banning large-scale sales of conventional vehicles as sufficient quantities of sustainable fuels are unlikely.
C. While some studies have emphasized the difficulties involved in decarbonizing transport, there is robust evidence that battery electric vehicles (BEVs) will form the backbone of future low-carbon road transport.

Answer with a permutation of A, B, and C that best restores the original paragraph (e.g., BAC, CAB). **Strictly follow this format–do not include any explanations or additional text.**

---

**Answer:** CBA

---

Table 48: **Example for Task 4: Political Sentence Reordering.**

---

**Prompt:**
You are given three shuffled sentences that originally formed a coherent paragraph from a policy brief document. Your task is to determine the correct order by selecting the most logical and coherent sequence.

Shuffled Sentences:
A. Our approach accounts for uncertain ties such as construction delays, utilization rates, and evolving market conditions, and assesses corresponding raw material needs.
B. Our study uses probabilistic modelling to project future battery demand and domestic production in Europe and evaluates Europe's pathway towards battery self-sufficiency via probabilistic statements.
C. We independently model demand and supply using S-shaped diffusion curves based on historical data, actual announced production capacities, and practice-oriented findings about how these announced capacities materialize over time.

Answer with a permutation of A, B, and C that best restores the original paragraph (e.g., BAC, CAB). **Strictly follow this format–do not include any explanations or additional text.**

---

**Answer:** CAB

---

Table 49: **Example for Task 5: Sentence Classification.**

---

**Prompt:**
You are given a sentence or paragraph from a scientific research paper. Classify the text into one of the following five categories relevant to policy briefs:

1. Policy Problem - Describes the societal or strategic issue that the research aims to address.
2. Scientific Research Findings - Reports key empirical results, model outcomes, or discoveries from the study.
3. Scientific Research Study Methods - Details the data sources, methodologies, models, or analytical procedures used.
4. Policy Implications - Explains how the research findings can inform, influence, or support policy decisions.
5. None - Does not fit into any of the above categories.

Your response must be exactly one of the following:
Policy Problem | Scientific Research Findings | Scientific Research Study Methods | Policy Implications | None

Text:
Batteries are critical to mitigate global warming, with battery electric vehicles as the backbone of low-carbon transport and the main driver of advances and demand for battery technology. However, the future demand and production of batteries remain uncertain, while the ambition to strengthen national capabilities and self-sufficiency is gaining momentum. In this study, leveraging probabilistic modelling, we assessed Europe's capability to meet its future demand for high-energy batteries via domestic cell production. We found that demand in Europe is likely to exceed 1.0 TWh yr-1 by 2030 and thereby outpace domestic production, with production required to grow at highly ambitious growth rates of 31-68% yr-1. European production is very likely to cover at least 50-60% of the domestic demand by 2030, while 90% self-sufficiency seems feasible but far from certain. Thus, domestic production shortfalls are more likely than not.

Category:

---

**Answer:** Policy Problem

---

Table 50: **Example for Task 6: Scientific Knowledge Understanding.**

---

**Prompt:**
You are given a multiple-choice question. Read the question carefully and select the best answer from the provided options. Each option is labeled with a capital letter (A, B, C, ...). Only one answer is correct. Provide your response as a single letter (e.g., A, B, C, ...).

Question:
An ileostomy opening should be:

Options:
A. less than 1cm in length.
B. invisible to the naked eye.
C. 3-5cm in length.
D. below skin level.
E. protruding 10cm from the skin.
F. over 10cm in length.
G. over 5cm in length.

Answer with a single letter (e.g., A, B, C, ...). **Strictly follow this format–do not include any explanations or additional text.**

---

**Answer:** C

---

Table 51: **Example for Task 7: Policy Problem Summarization.**

---

**Prompt:**
You are given a passage from a scientific paper that describes part of the policy problem motivating the research.

Summarize the specific issue mentioned in the passage using policy-brief style sentences. Your output should:
- Highlight the problem or challenge described
- Use accessible, non-technical language (technical terms are allowed when necessary)
- Focus only on what is present in the passage

Scientific Text:
Research shows that restrictive immigration policies and practices are associated with poor health, but far less is known about the relationship between inclusive immigration policies and health. Using data from the United States natality files, we estimate associations between state laws granting undocumented immigrants access to driver's licenses and perinatal outcomes among 4,047,067 singleton births to Mexican and Central American immigrant birthing people (2008-2021). Fitting multivariable log binomial and linear models, we find that the implementation of a license law is associated with improvements in low birthweight and mean birthweight. Replicating these analyses among U.S.-born non-Hispanic White birthing people, we find no association between the implementation of a license law and birthweight. These findings support the hypothesis that states' extension of legal rights to immigrants improves the health of the next generation.

Summary:

---

**Answer:** None (Use Gemini-2.5-Pro as the judge).

---

Table 52: **Example for Task 8: Research Findings Summarization.**

**Prompt:**
You are given a passage from a scientific paper that describes part of the research findings.

Summarize the specific findings using policy-brief style sentences. Your output should:
- Clearly state the result(s) presented in the passage
- Use plain and direct language (technical terms are allowed when necessary)
- Focus only on what is present in the passage

Scientific Text:
Table 2 displays the results of the ordered logistic regression analyses of community distrust and subjective social isolation. Model 1 shows that, independent of the controls, respondents in 2020 evidenced a significantly increased risk of community distrust. Being a respondent in the 2020 survey was associated with almost 50% greater odds of reporting a higher level of distrust than being a respondent in the 2019 sample. However, these between-wave differences did not vary by age; the interaction between wave of survey and age in Model 2 is not significant.

Summary:

**Answer:** None (Use Gemini-2.5-Pro as the judge).

Table 53: **Example for Task 9: Study Methods Summarization.**

**Prompt:**
You are given a passage from a scientific paper that describes part of the research study's methodology.

Summarize the method or approach using policy-brief style sentences. Your output should:
- Describe the model, data, or procedure mentioned in the passage
- Use clear and accessible language (technical terms are allowed when necessary)
- Focus only on what is present in the passage

Scientific Text:
The survey data were collected by the Energy Institute at the Johannes Kepler University Linz, following high European Union standards of data protection and voluntary study participation. The methodology used in this paper does not require institutional ethical approval according to the guidelines set out by the Energy Institute at the Johannes Kepler University Linz. Confidentiality and anonymity of participants were ensured, and informed written consent was obtained from all the interviewees.

Summary:

**Answer:** None (Use Gemini-2.5-Pro as the judge).

Table 54: **Example for Task 10: Policy Implications Summarization.**

**Prompt:**
You are given a passage from a scientific paper that describes part of the policy implications of the research findings.

Summarize the implications using policy-brief style sentences. Your output should:
- Explain how the described result or observation could inform or influence policy
- Use accessible language (technical terms are allowed when necessary)
- Focus only on what is present in the passage

Scientific Text:
This study was conducted in Chicago, Illinois. Although these data were collected before recreational marijuana use was legalized in Illinois, it is possible that attitudes toward marijuana use are generally more permissive in Chicago relative to other parts of the country. Future research should examine regional variations in responses to substance use in perinatal care settings. In addition, this analysis is premised on the uptake of legal tasks by individual providers in settings where testing is based on provider discretion. Studies examining how law shapes provider-patient interactions in systems where universal drug testing is carried out may reveal different patterns and outcomes. Another limitation of this study is that the data reflect providers' interpretations and perceptions of hospital testing protocol. In the future, researchers should examine official, documented hospital protocols to understand how formal organizational policy shapes provider practices. Finally, given sample size limitations and the wide range of state and organizational responses to substance use during pregnancy, these findings are not intended to capture practices in perinatal care by and large. Rather, these findings reflect dilemmas workers may contend with in settings where they are given latitude over the degree to which they adopt legal tasks or not.

Summary:

**Answer:** None (Use Gemini-2.5-Pro as the judge).

Table 55: **Example for Task 11: Policy Problem Generation.**

**Prompt:** You are given the full text of a scientific research paper.

Identify and summarize the **policy problem** that this research addresses. Your response should:
- Clearly state the societal, environmental, or strategic issue motivating the study
- Be written in accessible, policy-brief style sentences
- Focus on the core challenge the research aims to solve
- Avoid excessive detail or technical jargon unless necessary for clarity

Write a short paragraph suitable for inclusion under the "Policy Problem" section of a policy brief.

Scientific Research Paper:
{**Scientific paper text**}

Policy Problem:

**Answer:** The rapid diffusion of battery electric vehicles — in addition to the decarbonization of the energy sector — requires an increasing number of batteries. However, the EU's goal to cover 90% of its battery demand from domestic production by 2030 is at risk, as projected demand will likely exceed 1.0 TWh per year and outpace production capacity despite highly ambitious growth rates. If Europe fails to scale up production, it may face severe economic and geopolitical risks, due to increased dependence on external suppliers, weakened industrial competitiveness and potential for delayed decarbonization. An urgent question is therefore whether Europe can realistically meet its future battery demand through domestic production, and what policy actions are needed to ensure success.

Table 56: **Example for Task 12: Research Findings Generation.**

**Prompt:** You are given the full text of a scientific research paper.

Summarize the paper's key **research findings** in the style of a policy brief. Your response should:
- Focus on the major empirical or model-based results
- Present findings that are policy-relevant and supported by the study
- Use concise, non-technical language with essential technical terms as needed
- Avoid quoting directly or summarizing minor details

Write a short, clear paragraph appropriate for the "Scientific Research Findings" section of a policy brief.

Scientific Research Paper:
{**Scientific paper text**}

Scientific Research Findings:

**Answer:** We find that European battery cell demand will likely surpass 1.0 TWh per year by 2030, whereas domestic production capacity is expected to fall short, creating a risk of supply constraints. Although Europe can be expected to meet at least 50-60% of its demand through domestic production by 2030, achieving the EU's 90% self-sufficiency target is feasible but uncertain, as nearly half of our modelled scenarios fail to meet this target. If Europe wants more independence from battery cell imports, our findings highlight the urgency of accelerating production capacity expansion, scaling up a battery supply chain, and implementing strong industrial policies to support competitiveness and supply sovereignty. Our approach is broadly applicable to regions aiming for battery self-sufficiency and should be examined with interacting factors such as policy support and supply chain resilience. However, our analysis does not account for disruptive market shifts, policy reversals, or unexpected technological breakthroughs, which could substantially alter production and demand trajectories.

Table 57: **Example for Task 13: Study Methods Generation.**

**Prompt:** You are given the full text of a scientific research paper.

Summarize the **research methods** used in the study in a way that is informative for policy audiences. Your response should:
- Describe the overall study design, data sources, tools, and models used
- Avoid unnecessary technical details or equations
- Use accessible but precise language

Write a short paragraph suitable for the "Scientific Research Study Methods" section of a policy brief.

Scientific Research Paper:
{**Scientific paper text**}

Scientific Research Study Methods:

**Answer:** Our study uses probabilistic modelling to project future battery demand and domestic production in Europe and evaluates Europe's pathway towards battery self-sufficiency via probabilistic statements. We independently model demand and supply using S-shaped diffusion curves based on historical data, actual announced production capacities, and practice-oriented findings about how these announced capacities materialize over time. Our approach accounts for uncertainties such as construction delays, utilization rates and evolving market conditions, and assesses corresponding raw material needs. This study is particularly relevant given Europe's policy push toward climate neutrality and striving for resilient, sustainable battery value chains with domestic production and global competitiveness. By applying an established technology diffusion framework, we provide a robust, scenario-based outlook rather than relying on overly optimistic industry projections. Our method is suitable for evaluating long-term industrial transformation and supply chain resilience, making it applicable to other regions with similar ambitions.

Table 58: **Example for Task 14: Policy Implications Generation.**

**Prompt:** You are given the full text of a scientific research paper.
Summarize the study's **policy implications**—how the research findings can inform policy or decision-making. Your response should:
- Interpret what the results suggest for governments, regulators, or institutions
- Highlight potential actions, strategies, or decisions informed by the study
- Use policy-appropriate language without speculation beyond the study's conclusions
- Be clear, practical, and informative
Write a short paragraph for the "Policy Implications" section of a policy brief.

Scientific Research Paper:
{**Scientific paper text**}

Policy Implications:

**Answer:**
- Focus on net materialized production capacities rather than mere corporate announcements to ensure realistic policy planning and avoid overestimations, while ensuring a minimal level of production by local companies utilizing domestic intellectual property.
- Create predictable and reliable framework conditions for industry and end users to stimulate market demand and allow capacity announcements to materialize.
- Strengthen public-private partnerships to de-risk investments and streamline European regulations to accelerate the scale-up of battery production and regional supply chains.
- Create a competitive differentiation and level playing field via de-risking industrial policies, sustainability criteria, and local content requirements at the European level, and carefully balance trade policies to foster competitiveness with options for global collaboration and learning.
- Ensure continuous policy support and cascade research and development policies with industrial policies in terms of timing and scope in accordance with the scale-up of battery production and an evolving battery value chain.

Table 59: **Example for Task 15: Policy Brief Generation.**

**Prompt:** You are given the full text of a scientific research paper.

Your task is to generate a structured **policy brief** that includes the following components:
- A concise, descriptive **title** that captures the central theme or policy issue addressed in the paper (do not label it with "Title:").
- **Policy Problem** - What societal or strategic challenge is the paper addressing?
- **Scientific Research Findings** - What are the key results relevant to this issue?
- **Scientific Research Study Methods** - What methodology or data supports these findings?
- **Policy Implications** - How might the findings guide or influence policy?

Write each section in a clear, accessible style suitable for policymakers. Avoid overly technical language, but use essential terms when necessary. Structure the output as a labeled five-part policy brief document.

Scientific Research Paper:
{**Scientific paper text**}

Policy Brief:

**Answer:** {**Full Policy brief text**}

Table 60: **Example for Task 16: Scientific Claims Verification.**

**Prompt:**
You are given a passage from a scientific paper and a research finding derived from it.

Your task is to determine whether the finding is **fully supported** by the information provided in the passage.

Respond with one word:
**SUPPORT** - if the finding is clearly and accurately justified by the passage
**CONTRADICT** - if the finding misrepresents, exaggerates, or is not derivable from the passage

Do not make assumptions beyond the provided text. Use only the given evidence.

Scientific Text:
Fuelled by substantial BEV diffusion up to 2035, European battery demand is likely to surpass 1.0 TWhyr-1 by 2030 (in 69% of all scenarios). The interquartile range (IQR) in 2030 is 0.97-1.2 TWh yr-1. Some high-demand scenarios may exceed the 1 TWh threshold as early as 2026 and even approach 1.6 TWh yr-1 by 2030, with the top 10% exceeding 1.30 TWh yr-1.

Claimed Finding:
In around 69% of model runs, European battery demand exceeds 1 TWh per year by 2030.

Answer with SUPPORT or CONTRADICT.

**Strictly follow this format–do not include any explanations or additional text.**

Answer (SUPPORT / CONTRADICT):

**Answer:** SUPPORT

Table 61: **Example for Task 17: Scientific Claims Verification 2.**

**Prompt:**

You will be shown a claim related to the COVID-19 pandemic and the abstract of a biomedical research paper. Each sentence from the abstract will be on a separate line. Your task is to return a JSON object with two fields:

- "verdict": The fact-checking verdict. If the information in the abstract supports the claim, write "SUPPORT". If the abstract contradicts the claim, write "CONTRADICT".
- "evidence": An array of sentences providing evidence for the verdict. Please copy all relevant sentences verbatim from the abstract.

For instance, if the model were given the claim "the COVID vaccine is safe for healthy adults", the output might be
{
  "verdict": "SUPPORT",
  "evidence": ["Our findings indicate that the risks of severe COVID vaccine side effects are low for healthy adults."]
}

Claim: The d614g mutation in the sars-cov2 spike protein reduces infectivity in an ace2 receptor-dependent manner

Abstract:
The SARS-CoV2 coronavirus responsible for the current COVID19 pandemic has been reported to have a relatively low mutation rate.
Nevertheless, a few prevalent variants have arisen that give the appearance of undergoing positive selection as they are becoming increasingly widespread over time.
Most prominent among these is the D614G amino acid substitution in the SARS-CoV2 Spike protein, which mediates viral entry.
The D614G substitution, however, is in linkage disequilibrium with the ORF1b P314L mutation, where both mutations almost invariably co-occur, making functional inferences problematic.
In addition, the possibility of repeated new introductions of the mutant strain does not allow one to distinguish between a founder effect and an intrinsic genetic property of the virus.
Here, we synthesized and expressed the WT and D614G variant SARS-Cov2 Spike protein, and report that using a SARS-CoV2 Spike protein pseudotyped lentiviral vector we observe that the D614G variant Spike has >1/2 log(10) increased infectivity in human cells expressing the human ACE2 protein as the viral receptor.
The increased binding/fusion activity of the D614G Spike protein was corroborated in a cell fusion assay using Spike and ACE2 proteins expressed in different cells.
These results are consistent with the possibility that the Spike D614G mutant increases the infectivity of SARS-CoV2.

**Answer:**
{
  "verdict": "CONTRADICT",
  "evidence": [
    "Most prominent among these is the D614G amino acid substitution in the SARS-CoV2 Spike protein, which mediates viral entry.",
    "The increased binding/fusion activity of the D614G Spike protein was corroborated in a cell fusion assay using Spike and ACE2 proteins expressed in different cells.",
    "These results are consistent with the possibility that the Spike D614G mutant increases the infectivity of SARS-CoV2."
  ]
}

Table 62: **Example for Task 18: Policy Implications Verification.**

**Prompt:** You are given the full text of a scientific research paper and a proposed policy implication.

Your task is to determine whether the policy implication is **fully supported** by the scientific research paper.

Respond with one word:
**SUPPORT** - if the implication clearly follows from the paper's results and interpretations
**CONTRADICT** - if the implication misrepresents, overstates, or is not justified by the evidence or conclusions

Base your judgment only on what the paper states. Do not speculate beyond the provided content.

Full Scientific Paper:
{**Scientific paper text**}

Policy Implication:
If Europe wants more independence from battery cell imports, our findings highlight the urgency of accelerating production capacity expansion, scaling up a battery supply chain, and implementing strong industrial policies to support competitiveness and supply sovereignty.

Answer with SUPPORT or CONTRADICT.

**Strictly follow this format—do not include any explanations or additional text.**

Answer (SUPPORT / CONTRADICT):

**Answer:** SUPPORT

## L    PROMPTS FOR SCI2POL-BENCH EVALUATION AND DATASET CURATION

We provide detailed prompts used for LLM evaluation and dataset curation for Sci2Pol-Bench.

### L.1    TASKS 7-10 PROMPT FOR REFERENCE-FREE SCORE

We present the detailed prompt used for Gemini-2.5-Pro when serving as the evaluation judge for Tasks 7-10 in Table 63. We average the scores from the JSON output and multiply the result by 20 to scale it to a 0–100 range.

Table 63: **Prompt for LLM-based Judge for Summarization Tasks.**

---

**Prompt:**
You are a strict and critical evaluator of summaries. Evaluate the summary on the following dimensions using a 1-5 scale (1 = very poor, 5 = excellent). Be conservative in your judgments: do not give high scores unless the summary is genuinely outstanding.

(1) **Clarity**: whether the summary is reader-friendly and expresses ideas clearly.
(2) **Accuracy**: whether the summary contains the same information as the source document.
(3) **Coverage**: how well the summary covers the important information from the source document.
(4) **Overall quality**: how good the summary is overall at representing the source document; a good summary is a shorter piece of text that has the essence of the original and tries to convey the same information as the source document.

Return only a JSON object in this format:
```
{
 "clarity": <1-5>,
 "accuracy": <1-5>,
 "coverage": <1-5>,
 "overall_quality": <1-5>
}
```

_

**Source Passage:**
{**source passage text**}

**Summary:**
{**summary text**}

---

## L.2 TASK 11 PROMPT FOR REFERENCE-BASED SCORE

We present the detailed prompt used for Gemini-2.5-Pro when serving as the evaluation judge for Tasks 11 (*Policy Problem Generation*) in Table 64 and Table 65.

Let `prob_imp` and `prob_qual` be their respective JSON outputs, and define the component set

$$C = \{\text{background, existing\_problem, consequence, attention\_problem, supporting\_detail}\}.$$

We compute the raw score of Task 11 as

$$S_{\text{raw}} = \sum_{c \in C} \text{prob\_imp}[c] \cdot \text{prob\_qual}[c]$$

We then multiply the raw score by 20 to scale it to a maximum value of 100.

Table 64: **Prompt for LLM-based Judge for Task 11: Policy Problem Generation (Importance).**

---

**Prompt:**
You are a strict policy-brief evaluator. Given the full scientific PAPER, assign an **importance score** to each structural component for effectively communicating the policy problem, based only on the **PAPER**.

**Components:**
(1) **background** — what drives the problem (e.g., scientific, environmental, or economic context).
(2) **existing_problem** — the current obstacle, mismatch, or challenge.
(3) **consequence** — potential risks if the problem is not addressed.
(4) **attention_problem** — the key policy issue or question requiring urgent attention.
(5) **supporting_detail** — clarification or elaboration of any of the above.

**Scoring Instructions:**
• Assign an importance score between 0.0 and 1.0 for each component.
• A higher score means the component is essential for understanding the policy problem described in the PAPER.
• A lower score means the component is optional, minor, or not clearly relevant.
• If a component is not justified by the PAPER, assign 0.0.

**Strict-grading Instructions:**
• Score conservatively: if unsure, choose the lower score.
• Base each score **only** on the PAPER—no external references or assumptions.
• Return exactly the JSON object below (no explanations, no extra keys).

Return only a JSON object in this format:
```
{
 "background": <0.0-1.0>,
 "existing_problem": <0.0-1.0>,
 "consequence": <0.0-1.0>,
 "attention_problem": <0.0-1.0>,
 "supporting_detail": <0.0-1.0>
}
```

—

**PAPER:**
{**full PAPER text**}

---

Table 65: **Prompt for LLM-based Judge for Task 11: Policy Problem Generation (Quality).**

**Prompt:**
You are a strict policy-brief evaluator. Given the full scientific PAPER and the CANDIDATE's policy problem paragraph, assign **quality scores** to five aspects of how well the problems are conveyed in **CANDIDATE_POLICY_PROBLEM**.

**Components:**
(1) **background** — what drives the problem (e.g., scientific, environmental, or economic context).
(2) **existing_problem** — the current obstacle, mismatch, or challenge.
(3) **consequence** — potential risks if the problem is not addressed.
(4) **attention_problem** — the key policy issue or question requiring urgent attention.
(5) **supporting_detail** — clarification or elaboration of any of the above.

**Scoring Instructions:**
• Assign a quality score between 0.0 and 1.0 for each component.
• A higher score means the content is clear, logical, and strongly aligned with the PAPER.
• A lower score means the content is vague, incorrect, poorly structured, or missing.
• If a component is not addressed, assign 0.0.

**Strict-grading Instructions:**
• Score conservatively: if unsure, choose the lower score.
• Base each score **only** on comparisons between PAPER and CANDIDATE.
• Return exactly the JSON object below (no explanations, no extra keys).
• Only evaluate content in CANDIDATE_POLICY_PROBLEM.

**Return only a JSON object in this format:**
```
{
 "background": <0.0-1.0>,
 "existing_problem": <0.0-1.0>,
 "consequence": <0.0-1.0>,
 "attention_problem": <0.0-1.0>,
 "supporting_detail": <0.0-1.0>
}
```

_

**PAPER:**
{**full PAPER text**}

**CANDIDATE_POLICY_PROBLEM:**
{**candidate policy problem paragraph**}

### L.3 TASK 12 PROMPT FOR REFERENCE-BASED SCORE

We present the detailed prompt used for Gemini-2.5-Pro when serving as the evaluation judge for Task 12 (*Research Findings Generation*) in Table 66. We average the scores from the JSON output and multiply the result by 100 to scale it to a 0–100 range.

Table 66: **Prompt for LLM-based Judge for Task 12: Research Findings Generation.**

---

**Prompt:**
You are a strict policy-brief evaluator. Given the full scientific PAPER and the CANDIDATE's findings section, assign **quality scores** to five aspects of how well the findings are conveyed in **CANDIDATE_FINDINGS**.

**Criteria:**
(1) **completeness** — does the section include *all important findings* from the PAPER?
(2) **importance** — are the findings mentioned actually *important* according to the PAPER?
(3) **accuracy** — are the described findings *factually correct* and consistent with the PAPER?
(4) **summarizing_findings** — does the section effectively *emphasize and summarize* the key messages or implications from the data, rather than just listing facts?
(5) **specification_to_findings** — does the section *clarify the scope, context, or limitations* of the findings, including conditions under which they apply?

**Scoring Instructions:**
• Assign a score between 0.0 and 1.0 for each criterion.
• A higher score means the section performs well on that criterion.
• A lower score means the section is vague, misleading, incomplete, or missing that dimension.

**Strict-grading Instructions:**
• Score conservatively: if unsure, choose the lower score.
• Base each score **only** on comparisons between PAPER and CANDIDATE.
• Return exactly the JSON object below (no explanations, no extra keys).

**Return only a JSON object in this format:**
```
{
 "completeness": <0.0-1.0>,
 "importance": <0.0-1.0>,
 "accuracy": <0.0-1.0>,
 "summarizing_findings": <0.0-1.0>,
 "specification_to_findings": <0.0-1.0>
}
```

—

**PAPER:**
{**full PAPER text**}

**CANDIDATE_FINDINGS:**
{**candidate findings section**}

---

### L.4 TASK 13 PROMPT FOR REFERENCE-BASED SCORE

We present the detailed prompt used for Gemini-2.5-Pro when serving as the evaluation judge for Task 13 (*Study Methods Generation*) in Table 67. We use a weighted rubric with the scores from the JSON output. We give greater weight to the first two criteria, because they carry more information, while the third serves only as an auxiliary signal.

$$S_{\text{raw}} = 2 \times \texttt{methods\_clarity\_and\_purpose}$$
$$+ 2 \times \texttt{methods\_technicality\_appropriateness}$$
$$+ \texttt{methods\_explanation\_of\_terms}$$

We average the above score by 5 and multiply the result by 100 to scale it to a 0–100 range.

Table 67: **Prompt for LLM-based Judge for Task 13: Study Methods Generation.**

---

**Prompt:**
You are a strict policy-brief evaluator. Given the full scientific PAPER and the CANDIDATE's methods section, assign **quality scores** to three core aspects of how the methodology is described. Each score should be a float between 0.0 and 1.0.

**Criteria:**
(1) **clarity_and_purpose** — Is the method described in a clear, structured way that highlights *what was done and why*, rather than simply listing tools or data sources?
(2) **technicality_appropriateness** — Is the level of technical detail appropriate for a policy audience without excessive jargon, complexity, or irrelevant detail?
(3) **explanation_of_terms** — Are technical terms, models, or data sources *explained* in accessible language and context without unexplained acronyms or unclear references?

**Scoring Instructions:**
• Assign a score between 0.0 and 1.0 for each criterion.
• A higher score means the section performs well on that criterion.
• A lower score means the section is vague, overly technical, unexplained, or missing that dimension.

**Strict-grading Instructions:**
• Score conservatively: if unsure, choose the lower score.
• Base each score **only** on comparisons between PAPER and CANDIDATE.
• Return exactly the JSON object below (no explanations, no extra keys).

**Return only a JSON object in this format:**
{
 "clarity_and_purpose": <0.0-1.0>,
 "technicality_appropriateness": <0.0-1.0>,
 "explanation_of_terms": <0.0-1.0>
}

—

**PAPER:**
{**full PAPER text**}

**CANDIDATE_METHOD:**
{**candidate methods section**}

---

## L.5 TASK 14 PROMPT FOR REFERENCE-BASED SCORE

We present the detailed prompt used for Gemini-2.5-Pro when serving as the evaluation judge for Task 14 (*Policy Implications Generation*) in Table 68. We average the scores from the JSON output and multiply the result by 100 to scale it to a 0–100 range.

Table 68: **Prompt for LLM-based Judge for Task 14: Policy Implications Generation.**

---

**Prompt:**
You are a **strict** policy-brief evaluator. Given the full scientific PAPER and the CANDIDATE's policy implications section, assign **quality scores** to the following four criteria.

**Dimensions:**
(1) **accuracy** — Are the implications *explicitly supported* by the PAPER without speculative or hallucinated claims?
(2) **coverage** — Does the section capture *all major implications* stated in the PAPER?
(3) **conciseness_and_distinctness** — Are the implications *concise and non-redundant*? Each point should make a *distinct* contribution.
(4) **alignment_with_paper_intent** — Does the implication reflect the PAPER's *main message or takeaway* (e.g., recommendation, warning, scientific insight, call to awareness)?

**Scoring Instructions:**
• Assign a score between 0.0 and 1.0 for each dimension.
• A higher score means the section performs well on that dimension.
• A lower score means the section is vague, incorrect, redundant, or misaligned.

**Strict-grading Instructions:**
• Score conservatively: if unsure, choose the lower score.
• Base each score **only** on comparisons between PAPER and CANDIDATE.
• Return exactly the JSON object below (no explanations, no extra keys).

**Return only a JSON object in this format:**
```
{
 "accuracy": <0.0-1.0>,
 "coverage": <0.0-1.0>,
 "conciseness_and_distinctness": <0.0-1.0>,
 "alignment_with_paper_intent": <0.0-1.0>
}
```

—

**PAPER:**
{**full PAPER text**}

**CANDIDATE_IMPLICATION:**
{**candidate policy implications section**}

---

## L.6 Task 15 Prompt for Reference-based Score

We present the detailed prompt used for Gemini-2.5-Pro when serving as the evaluation judge for Task 15 (*Policy Brief Generation*) in Table 69. We average the scores from the JSON output and multiply the result by 100 to scale it to a 0–100 range.

Table 69: **Prompt for LLM-based Judge for Task 15: Policy Brief Generation.**

---

**Prompt:**
You are a strict policy-brief evaluator. Given the full scientific PAPER, an EXPERT-written reference brief, and a CANDIDATE brief, grade the CANDIDATE on four dimensions and produce a compact JSON report.

**Evaluation dimensions & conservative 0-5 rubric**
Start each score at 0 and add points only when the brief clearly meets the criterion. Reserve 4 or 5 for near-flawless performance; 3 means "solid but with notable gaps"; 2 or below signals clear problems.
0 = disastrous    1 = poor    2 = fair    3 = good    4 = very good    5 = excellent
(1) **ContextualDepth**: Does the CANDIDATE capture the study's essential quantitative findings, methods, and broader context (e.g., raw-material outlook, scenario count) *without missing key facts or adding fluff*?
(2) **HallucinationRisk**: Are *all* claims traceable to the PAPER (or universally known)? Deduct heavily for any unsupported number or causal claim.
(3) **ReadabilityTone**: Is the brief concise, logically ordered, written in active voice, and appropriate for policymakers? Penalize lengthy sentences or jargon.
(4) **Actionability**: Are policy implications concrete, tied directly to evidence, and immediately useful? Vague or speculative advice $\leq 2$.

**Output format (MUST be valid JSON; numeric scores only, no prose):**
```
{
 "contextual_depth": <0-5>,
 "hallucination_risk": <0-5>,
 "readability_tone": <0-5>,
 "actionability": <0-5>
}
```

**Strict-grading instructions:**
• Score conservatively: if unsure, choose the lower score.
• Base each score only on comparisons between PAPER and CANDIDATE; EXPERT_BRIEF is reference context.
• Return exactly the JSON object above (no explanations, no extra keys).

–

**PAPER:**
{**full PAPER text**}

**EXPERT_BRIEF:**
{**expert-written brief**}

**CANDIDATE_BRIEF:**
{**candidate brief**}

---

## L.7 TASK 5 PROMPT FOR DATA CURATION

We present the detailed prompt used with GPT-o3 when curating the dataset for Task 5 in Table 70.

Table 70: **Prompt for the Data Curation in Task 5.**

---

**Prompt:**
You are given a scientific paper and a corresponding policy brief. The policy brief includes four components:
1. Policy Problem
2. Scientific Research Findings
3. Scientific Research Study Methods
4. Policy Implications

Your task is to extract valuable, content-rich passages from the scientific paper that correspond to each of these components, **as reflected in the policy brief**. Each sample should preferably contain **three or more coherent and consecutive sentences**, copied **verbatim** from the scientific paper. However, shorter excerpts are acceptable if they are highly informative. Avoid random, trivial, or disjointed selections.

Assign one of the following five labels to each extracted sample:
(1) Policy Problem
(2) Scientific Research Findings
(3) Scientific Research Study Methods (e.g., experimental design, data sources, modeling, and implementation details)
(4) Policy Implications
(5) None: for content unrelated to the policy translation task (including acknowledgments, author contributions, and institutional affiliations)

Return exactly:
- 1 sample for Policy Problem
- 5 samples for Scientific Research Findings
- 5 samples for Scientific Research Study Methods
- 2 samples for Policy Implications
- 2 samples for None

Output the result as a JSON array of objects, each with the following fields:
- `"label"`: one of ["Policy Problem", "Scientific Research Findings", "Scientific Research Study Methods", "Policy Implications", "None"]
- `"text"`: the extracted passage copied verbatim from the scientific paper

**Do not** paraphrase. **Do not** include commentary. Only output a valid JSON array of labeled, **verbatim** text segments.

---

## L.8 TASK 11 PROMPT FOR DATA CURATION

We present the detailed prompt used with GPT-o3 when curating the dataset for Task 11 in Table 71.

Table 71: **Prompt for the Data Curation in Task 11.**

---

**Prompt:**
In the following, you will see three examples. Each example includes a scientific research paper and a paragraph describing the **policy problem** that the research addresses, as written for a policy brief.

Your task is to write a new **policy problem** paragraph for a different scientific paper that I will provide.

**Note:** I will also give you an additional paragraph related to the policy problem for the new paper. You may refer to it for context, but it is **not** the desired output.

**Example 1:**
Scientific paper: {**scientific paper text**}
Policy problem paragraph: {**policy problem text**}

**Example 2:**
Scientific paper: {**scientific paper text**}
Policy problem paragraph: {**policy problem text**}

**Example 3:**
Scientific paper: {**scientific paper text**}
Policy problem paragraph: {**policy problem text**}

**New Paper:** {**scientific paper text**}
Related paragraph for policy problem: {**related policy problem text**}
Policy problem paragraph:

---

## L.9  TASK 13 PROMPT FOR DATA CURATION

We present the detailed prompt used with GPT-o3 when curating the dataset for Task 13 in Table 72.

Table 72: **Prompt for the Data Curation in Task 13.**

---

**Prompt:**
In the following, you will see three examples. Each example includes a scientific research paper and a paragraph describing the **scientific research study method** that the research addresses, as written for a policy brief.

Your task is to write a new **scientific research study method** paragraph for a different scientific paper that I will provide.

**Note:** I will also give you an additional paragraph related to the scientific research study method for the new paper. You may refer to it for context, but it is **not** the desired output.

**Example 1:**
Scientific paper: {**scientific paper text**}
Scientific research study method paragraph: {**scientific research study method text**}

**Example 2:**
Scientific paper: {**scientific paper text**}
Scientific research study method paragraph: {**scientific research study method text**}

**Example 3:**
Scientific paper: {**scientific paper text**}
Scientific research study method paragraph: {**scientific research study method text**}

**New Paper:** {**scientific paper text**}
Related paragraph for scientific research study method: {**scientific research study method text**}
Scientific research study method paragraph:

---

## L.10 TASK 16 PROMPT FOR DATA CURATION

We present the detailed prompt used with GPT-o3 when curating the dataset for Task 16 in Table 73.

Table 73: **Prompt for the Data Curation in Task 16.**

---

**Prompt:**
You are given a scientific research paper.

Your task is to generate **ten** query-answer pairs for the following binary classification task:

> Determine whether a stated research finding is *fully supported* by the research results reported in the scientific paper.

Each query-answer pair must include:
1. **research_results** - Copy **one or two consecutive paragraphs verbatim** from the paper that present empirical findings, statistics, or core observations.
2. **research_finding** - Write a concise sentence that either:
   - *Accurately follows* from the results (**SUPPORT**)
   - *Sounds plausible*, but is **not actually supported**, misstates causal direction, overgeneralizes, or infers something beyond the evidence (**CONTRADICT**)
3. **answer** - Either `"SUPPORT"` or `"CONTRADICT"`

**Requirements:**
- Return **exactly 10 entries** in total.
- Include **5** SUPPORT and **5** CONTRADICT examples–no more, no fewer.
- Use a *different* results passage for each entry–**do not reuse**.
- Make the distinction between SUPPORT and CONTRADICT **subtle and challenging** (e.g., include plausible misinterpretations, causal reversals, or logical overextensions).
- Reproduce **all paper text exactly as written**–no paraphrasing, truncation, or ellipses.
- Output only a valid **JSON file** containing a list of 10 dictionaries.
- Each dictionary must contain exactly the following keys: `"research_results"`, `"research_finding"`, and `"answer"`.

**Output JSON format:**
```
{[
  {
   "research_results": "<verbatim paragraph(s)>",
   "research_finding": "<concise sentence>",
   "answer": "SUPPORT" | "CONTRADICT"
  },
  ...
]}
```

---

### L.11   TASK 18 PROMPT FOR DATA CURATION

We present the detailed prompt used with GPT-o3 when curating the dataset for Task 18 in Table 74.

Table 74: **Prompt for the Data Curation in Task 18.**

---

**Prompt:**
Please rewrite each of the following policy recommendations to express the opposite meaning as clearly and thoroughly as possible.

**Policy Implications:**
{**policy implication text**}

---

## M    PROMPTS FOR SCI2POL-CORPUS CURATION

In this section, we present the detailed prompts used for Sci2Pol-Corpus curation: (i) the coarse-grained filtering prompt in Section 3.2 (Table 75); (ii) the fine-grained filtering prompt in Section 3.2 (Table 76); and (iii) the in-context polishing prompt in Section 3.3 (Table 77).

Table 75: **Prompt for the Coarse-grained Filtering Step in Section 3.2**

---

**Prompt:**
I will give you a policy document and a scientific article abstract. Your task is to evaluate whether the policy document is primarily about the scientific article it cites. Consider the following criteria:
1. Discussion of the Article's Content: The policy document must explicitly discuss the findings, methodology, or conclusions of the scientific article in detail.
2. Policy Implications: The document must connect the scientific article to policy decisions, recommendations, or implications for policymakers.
3. Central Focus: The scientific article should be a key focus of the policy document, rather than being just one of many references or a minor supporting citation.

Evaluation Steps:
- Read the scientific article abstract to understand its key points.
- Analyze the policy document to determine whether it engages with the article's content, its implications, and whether the article is a central focus.
- Score the policy document on the following dimensions:
  • Relevance (0-5): How central is the scientific article to the policy document? (0 = only briefly mentioned, 5 = core focus).

  • Depth of Discussion (0-5): To what extent does the policy document engage with the scientific article's content (e.g., findings, methodology, conclusions)? (0 = minimal detail, 5 = in-depth discussion).

  • Policy Connection (0-5): How well does the policy document translate the scientific article into policy implications or recommendations? (0 = no connection, 5 = strong, explicit connection).

  • Citation Frequency & Emphasis (0-5): How frequently and prominently is the article referenced in the policy document? (0 = one minor mention, 5 = referenced throughout as a key source).
- Provide a final verdict on whether the policy document is primarily about the scientific article.
- Return the output in a valid JSON format.

**Output JSON format:**
{[
  {
    "verdict": "Yes" | "No",
    "scores": {
      "relevance": 0-5,
      "depth_of_discussion": 0-5,
      "policy_connection": 0-5,
      "citation_frequency_emphasis": 0-5
    },
    "justification": "<3-5 sentence explanation>"
  }
]}

---

Table 76: **Prompt for the Fine-grained Filtering Step in Section 3.2**

**Prompt:**

I will give you a policy document and a scientific article. Your task is to evaluate whether the policy document is primarily about the scientific article. Consider the following criteria:

1. Discussion of the Article's Content: The policy document must explicitly discuss the findings, methodology, or conclusions of the scientific article in detail.

2. Policy Implications: The document must connect the scientific article to policy decisions, recommendations, or implications for policymakers.

3. Central Focus: The scientific article should be a key focus of the policy document, rather than being just one of many references or a minor supporting citation.

Evaluation Steps:

- Read the scientific article to understand its key points.
- Analyze the policy document to determine whether it engages with the article's content, its implications, and whether the article is a central focus.
- Score the policy document on the following dimensions:
  - Relevance (0-5): How central is the scientific article to the policy document? (0 = only briefly mentioned, 5 = core focus).
  - Depth of Discussion (0-5): To what extent does the policy document engage with the scientific article's content (e.g., findings, methodology, conclusions)? (0 = minimal detail, 5 = in-depth discussion).
  - Policy Connection (0-5): How well does the policy document translate the scientific article into policy implications or recommendations? (0 = no connection, 5 = strong, explicit connection).
  - Citation Frequency & Emphasis (0-5): How frequently and prominently is the article referenced in the policy document? (0 = one minor mention, 5 = referenced throughout as a key source)
  - Document Similarity (0-5): Are the policy document and the scientific article almost exactly the same with only minor formatting differences? (0 = the text of policy document is very different from the scientific article, 5 = the two documents are nearly identical).
- Provide a final verdict on whether the policy document is primarily about the scientific article.
- Return the output in a valid JSON format.

**Output Format:**
```
{[
  {
    "verdict": "Yes" | "No",
    "scores": {
      "relevance": 0-5,
      "depth_of_discussion": 0-5,
      "policy_connection": 0-5,
      "citation_frequency_emphasis": 0-5,
      "doc_similarity": 0-5
    },
    "justification": "<3-5 sentence explanation>"
  }
]}
```

Table 77: **Prompt for the In-context Polishing Step in Section 3.3.**

---

**Prompt:**
You are a professional editor specializing in policy briefs based on scientific research. Use the sample scientific papers and their corresponding policy briefs as the standard for tone, structure, and formatting. Based on this reference, revise the draft policy brief for the target scientific paper. Ensure the revised brief is clear, accurate, concise, and policy-relevant, matching the quality of the samples.

Sample Scientific Paper 1: {**scientific paper text**}
Sample Policy Brief 1: {**policy brief text**}

Sample Scientific Paper 2: {**scientific paper text**}
Sample Policy Brief 2: {**policy brief text**}

Sample Scientific Paper 3: {**scientific paper text**}
Sample Policy Brief 3: {**policy brief text**}

Target Scientific Paper: {**scientific paper text**}
Draft Policy Brief: {**policy brief text**}

Respond with the **revised policy brief only**, using the following format:
- **Policy Problem**: Concise and precise, aligned with sample quality.
- **Scientific Research Findings**: Comprehensive and coherent (no bullet points), matching the structure of the original paper.
- **Scientific Research Study Methods**: Narrative format (no point form), at the same level of generality and technicality as the samples.
- **Policy Implications**: Bullet points only; grounded strictly in the paper's findings without speculation or external examples.

**Requirements:**
- Maintain the same **functional length** as the samples: each section should be long enough to reflect the depth and structure of the specific paper, not artificially extended or shortened to match sample length. Do not pad with filler, overexplain to match longer samples, or oversimplify to match shorter ones.
- Use a professional, policy-oriented voice for a scientifically literate audience.
- Ensure strict factual alignment with the target scientific paper.

---

