# OpenReview forum: "Sci2Pol: Evaluating and Fine-tuning LLMs on Scientific-to-Policy Brief Generation"
_ICLR.cc/2026/Conference — ICLR 2026 Poster_

### Official Review · Reviewer_6pgv · 2025-10-30

[review text omitted: it was posted to a different submission]

---

> ### Author Response · Authors · 2025-11-19
> **Cross-domain generalization testing and human evaluation**
>
> Thank you for your detailed review. We have uploaded the revised manuscript, with **all changes marked in blue**.
>
> We note that the content of this review does not align with our submission. It appears intended for the paper “Automatically Advancing LLM Expertise in Technology Judgment” (ID: 22563), reviewed by Reviewer CuVx.  To proceed, the following response directly addresses the questions we have extracted from the Reviewer CuVx's comments (under [review link](https://openreview.net/forum?id=KMZ7QHguGz)).
>
> ---
>
> > `W1`: Revision of introduction.
>
> **Response**
>
> Thanks for your valuable comments. We have added the preliminary references and modified the statement in the revised manuscript. We quote here for clarity:
>
> "Although LLMs demonstrate strong general capabilities, prior work shows they hallucinate scientific content, misverify claims, and produce unstable or biased policy reasoning [1,2,3]."
>
> [1] Coz. et.al. What Would an LLM Do? Evaluating Policymaking Capabilities of Large Language Models.
>
> [2] Bai. et.al. LLM-generated messages can persuade humans on policy issues.
>
> [3] Wadden. et.al. SciRIFF: A Resource to Enhance Language Model Instruction-Following over Scientific Literature
>
> > `W2`: Cross-domain generalization analysis.
>
> **Response**
>
> Thanks for your insightful comment. We have added the cross-domain generalization analysis in **Appendix I**.
>
> Due to the scarcity of readily available peer-reviewed scientific paper-policy brief pairs in other domains, we conduct an exploratory evaluation using one representative paper from each of the following important fields: AI, Robotics, Mathematics, Economics, and Humanities.
> The generated policy briefs are then subjected to human evaluation.
> We test three models: Gemma-27B-SFT, DeepSeek-V3, and GPT-4o.
>
> We provide the details of 5 papers as follows:
>
> (i) AI: "Generative AI needs adaptive governance". We show the results in Tables 36 and 37.
>
> (ii) Robotics: "Anticipatory policy design for robots in public spaces". We show the results in Tables 38 and 39.
>
> (iii) Mathematics: "Educational policy to improve mathematics instruction at scale: Conceptualizing contextual factors". We show the results in Tables 40 and 41.
>
> (iv) Economics: "Financial inclusion, the shadow economy and economic growth in developing economies". We show the results in Tables 42 and 43.
>
> (v) Humanities: "Science cited in policy documents: Evidence from the Overton database". We show the results in Tables 44 and 45.
>
> The resulting evaluation indicates that our supervised fine-tuning successfully generalizes to other domains.
>
> Please refer to Appendix I of the revised manuscript for details.
>
> > `W3`: Problem with LLM judge.
>
> **Response (Part 1)**
>
> Thanks for the insightful comment. We add the “Human Evaluation” experiments in Appendix H.7.1. and “Independent Judge” experiments in Appendix H.6.
>
> 1. **Human Evaluation** (Appendix H.7.1 in the revised version). We ask two human evaluators to directly score model outputs and compare their averaged scores with those produced by the Gemini judge. We randomly sample 10 cases from Tasks 9 and 11-15 for human evaluation (using the same 10 cases across Tasks 11--15). Due to the limited number of evaluation questions, we report only the final results and do not conduct bootstrap significance testing. We show the results in the following Table 1. The results show that Gemma-27B-SFT consistently outperforms DeepSeek-V3 and GPT-4o, aligning with our Gemini-based evaluation. For completeness, we present one representative example for Tasks 9 and 15 along with brief analyses in Appendix H.7.1 (Tables 22 - 24)
>
>
> ### **Table 1 (Part 1): Direct Human Scoring on Tasks 9, 11, 12**
>
> | Model | T9 Gemini | T9 Human | T11 Gemini | T11 Human | T12 Gemini | T12 Human |
> |-------|-----------|----------|------------|-----------|------------|-----------|
> | **Gemma-3-27B-SFT** | 83.40±1.03 | 82.00 | 91.76±0.90 | 89.20 | 78.26±1.35 | 75.10 |
> | **DeepSeek-V3** | 76.90±0.05 | 74.00 | 91.50±0.93 | 88.40 | 78.24±1.46 | 75.60 |
> | **GPT-4o** | 71.38±0.06 | 70.00 | 89.17±0.98 | 86.30 | 72.69±1.05 | 71.30 |
>
> ---
>
> ### **Table 1 (Part 2): Direct Human Scoring on Tasks 13, 14, 15**
>
> | Model | T13 Gemini | T13 Human | T14 Gemini | T14 Human | T15 Gemini | T15 Human |
> |-------|-------------|------------|-------------|------------|-------------|------------|
> | **Gemma-3-27B-SFT** | 67.34±2.45 | 65.20 | 86.65±1.53 | 83.25 | 83.65±1.75 | 80.00 |
> | **DeepSeek-V3** | 90.29±0.99 | 84.50 | 87.62±1.28 | 84.50 | 83.51±1.65 | 79.10 |
> | **GPT-4o** | 72.81±1.50 | 68.20 | 78.36±1.54 | 78.25 | 68.93±1.31 | 65.90 |

---

> ### Author Response · Authors · 2025-11-19
> **Independent judge**
>
> > `W3`: Problem with LLM judge.
>
> **Response (Part 2)**
>
> 2. **Independent Judge** (Appendix H.6 in the revised version). To reduce potential bias from using Gemini as a single automated judge in Tasks 7-15, we include additional evaluation results using MiniMax-M2 as an independent judge. We first report the evaluation results obtained using MiniMax-M2. The averaged scores are presented in the following Table 2, and the detailed task-level results for Tasks 7–15 are shown in Table 3. Across all models and tasks, the scores produced by Gemini and MiniMax-M2 exhibit strong agreement in both relative model ranking and within-task performance trends. These analyses demonstrate that the conclusion from using Gemini as the judge is reliable.
>
> ### **Table 2: Averaged Performance Using MiniMax-M2 as the Judge**
>
> | Model | Auto. (1–4) | Under. (5–6) | Sum. (7–10) | Gene. (11–15) | Ver. (16–18) | Avg. | Gain |
> |------|-------------|---------------|--------------|----------------|---------------|------|------|
> | **LLaMA-3.1-8B-IT** | 27.12±2.53 | 47.74±1.54 | 68.61±1.20 | 66.83±1.75 | 56.59±1.27 | 54.59±1.66 | – |
> | **LLaMA-3.1-8B-SFT** | 31.27±2.90 | 44.34±1.48 | 73.81±1.06 | 76.81±1.32 | 63.50±1.08 | 60.20±1.57 | **+5.61** |
> | **Gemma-3-12B** | 42.96±2.79 | 69.61±1.28 | 84.39±0.98 | 73.42±1.45 | 63.05±1.06 | 66.49±1.51 | – |
> | **Gemma-3-12B-SFT** | 43.14±2.86 | 69.53±1.25 | 85.91±0.96 | 82.89±1.53 | 64.59±1.05 | 70.19±1.53 | **+3.70** |
> | **Gemma-3-27B** | 43.60±2.83 | 67.82±1.42 | 82.72±0.91 | 79.65±1.56 | 65.79±0.98 | 68.70±1.54 | – |
> | **Gemma-3-27B-SFT** | 45.39±2.90 | 67.44±1.36 | 85.91±0.99 | 85.41±1.43 | 67.17±0.96 | **71.59±1.57** | **+2.89** |
> | **DeepSeek-V3** | 39.54±3.06 | 79.35±1.28 | 85.70±0.98 | 83.95±1.60 | 67.74±0.85 | 71.25±1.55 | – |
> | **GPT-4o** | 52.17±3.00 | 77.17±1.32 | 80.01±0.97 | 77.80±1.48 | 62.98±0.82 | 70.05±1.52 | – |
>
> ---
>
> ### **Table 3 (Part 1): Agreement Between Gemini and MiniMax-M2 on Tasks 7–10**
>
> | Model | T7 Gemini | T7 MiniMax | T8 Gemini | T8 MiniMax | T9 Gemini | T9 MiniMax | T10 Gemini | T10 MiniMax |
> |-------|-----------|------------|-----------|------------|-----------|------------|-------------|--------------|
> | **LLaMA-3.1-8B-IT** | 67.38±0.05 | 64.11±1.14 | 67.38±0.05 | 70.40±1.36 | 59.03±0.05 | 68.93±1.23 | 63.88±0.05 | 71.00±1.07 |
> | **LLaMA-3.1-8B-SFT** | 71.17±1.45 | 71.89±1.05 | 76.00±1.14 | 76.92±1.18 | 84.72±1.10 | 81.80±1.14 | 86.66±1.11 | 76.64±0.89 |
> | **Gemma-3-12B** | 71.18±0.05 | 76.50±0.98 | 65.45±0.05 | 67.64±0.99 | 76.71±0.05 | 74.06±0.95 | 85.60±1.02 | 75.48±0.98 |
> | **Gemma-3-12B-SFT** | 79.17±1.39 | 81.63±0.99 | 81.75±1.22 | 87.34±1.09 | 88.38±1.03 | 82.06±0.95 | 89.76±1.00 | 80.53±0.81 |
> | **Gemma-3-27B** | 75.98±0.05 | 79.96±0.88 | 68.57±0.05 | 71.29±0.95 | 76.03±0.05 | 74.63±0.96 | 91.33±0.87 | 83.99±0.84 |
> | **Gemma-3-27B-SFT** | 85.42±1.15 | 83.76±1.02 | 83.40±1.03 | 89.35±1.09 | 87.30±1.00 | 82.95±0.98 | 91.76±0.90 | 85.57±0.89 |
> | **DeepSeek-V3** | 80.95±0.05 | 82.90±0.95 | 81.90±0.05 | 87.65±0.93 | 77.72±0.05 | 80.78±1.09 | 91.50±0.93 | 84.46±0.93 |
> | **GPT-4o** | 75.20±0.06 | 77.30±0.91 | 71.38±0.06 | 75.99±1.00 | 72.70±0.06 | 78.13±0.97 | 89.17±0.98 | 79.76±0.99 |
>
> ---
>
> ### **Table 3 (Part 2): Agreement Between Gemini and MiniMax-M2 on Tasks 11–15**
>
> | Model | T11 Gemini | T11 MiniMax | T12 Gemini | T12 MiniMax | T13 Gemini | T13 MiniMax | T14 Gemini | T14 MiniMax | T15 Gemini | T15 MiniMax |
> |--------|-------------|--------------|--------------|--------------|--------------|--------------|--------------|--------------|--------------|--------------|
> | **LLaMA-3.1-8B-IT** | 82.35±1.55 | 44.95±2.61 | 64.62±1.65 | 55.48±1.51 | 55.12±1.90 | 51.66±1.16 | 67.74±1.89 | 70.27±1.83 | 59.05±1.55 | 60.61±1.64 |
> | **LLaMA-3.1-8B-SFT** | 86.66±1.11 | 51.28±1.56 | 71.10±1.47 | 62.27±1.22 | 74.19±1.83 | 62.56±1.43 | 83.41±1.61 | 74.05±1.03 | 72.76±1.75 | 67.35±1.36 |
> | **Gemma-3-12B** | 85.60±1.02 | 50.71±1.89 | 70.22±1.15 | 60.23±1.26 | 70.69±2.29 | 60.89±1.25 | 84.46±1.29 | 74.89±1.41 | 75.71±1.43 | 68.55±1.44 |
> | **Gemma-3-12B-SFT** | 89.76±1.00 | 53.58±2.50 | 71.93±1.39 | 63.09±1.30 | 70.92±2.08 | 61.99±1.38 | 84.18±1.52 | 75.06±1.24 | 76.06±1.66 | 69.21±1.24 |
> | **Gemma-3-27B** | 91.33±0.87 | 53.50±2.27 | 75.23±1.12 | 63.22±1.51 | 88.12±1.32 | 63.63±1.33 | 88.54±1.24 | 77.76±1.21 | 80.89±1.26 | 70.82±1.50 |
> | **Gemma-3-27B-SFT** | 91.76±0.90 | 55.20±2.20 | 78.26±1.35 | 66.53±1.11 | 67.34±2.45 | 60.53±1.45 | 86.65±1.53 | 77.78±1.02 | 83.65±1.75 | 75.81±1.37 |
> | **DeepSeek-V3** | 91.50±0.93 | 54.32±2.44 | 78.24±1.46 | 66.38±1.71 | 90.29±0.99 | 65.56±1.08 | 87.62±1.28 | 77.98±1.47 | 83.51±1.65 | 74.45±1.30 |
> | **GPT-4o** | 89.17±0.98 | 52.01±2.40 | 72.69±1.05 | 63.19±1.11 | 72.81±1.50 | 62.54±1.24 | 78.36±1.54 | 71.56±1.42 | 68.93±1.31 | 65.61±1.25 |

---

> ### Author Response · Authors · 2025-11-19
>
> > `W4`: Lack of experimental analysis.
>
> **Response**
>
> Thanks for your insightful comments. We are sorry for the oversight. We have added (i) common failure modes analysis in Appendix H.3; (ii) what supervised learning learns in Appendix H.4; (iii) saturation testing in Appendix H.12.
>
> Please refer to the revised manuscript for details, and we present the core idea here for clarity:
>
> (i) Common failure modes analysis (Appendix H.3): We summarize the most salient failure modes of strong general-purpose LLMs by task group and connect them to our design supervised fine-tuning targets.
>
> (ii) What supervised learning learns (Appendix H.4): We identify four aspects of learning: factual grounding, style, pattern alignment, and implicit skill acquisition. A detailed comparison of base and fine-tuned outputs is provided in Table 15, with additional analyses of leading-model shortcomings in Tables 16 and 17.
>
> (iii) Saturation testing (Appendix H.12): We report the relationship between Sci2Pol-Bench performance and three commonly used model characteristics: (a) model size, (b) pre-training FLOPs, and (c) release date. The results show that Sci2Pol-Bench performance does not follow a simple scaling law. Model size, compute, and release date each matter, but do not fully explain cross-model differences. Instead, high performance is often associated with improved alignment strategies, targeted supervised finetuning, and richer training mixtures. These are factors that help models reason about scientific evidence and policy trade-offs beyond what raw scale alone can provide.
>
> > `W5`: Figure and organization.
>
> **Response**
>
> Thanks for your valuable feedback. We are sorry for the confusion.
>
> We have clearly annotated Figure 1 (we also define the JHSB in the figure), as shown here: [figure link](https://www.dropbox.com/scl/fi/gshtp3gdzet2jnc6k95dc/figure1.png?rlkey=0x7byfimh9tz4vzyp0fxj8nkl&st=230jqrbf&dl=0)
>
> We also adjust the organization and put Table 2 (Performance of LLMs on Sci2Pol-Bench) and Figure 3 (Visualization of Performance of 13 LLMs on Sci2Pol Bench) together for better readability.
>
> ---
>
> We hope that we address your concerns. Please feel free to ask if you have further concerns.

---

### Official Review · Reviewer_F3Nb · 2025-10-31

**Soundness:** 3
**Presentation:** 3
**Contribution:** 3
**Rating:** 6
**Confidence:** 4

**Summary:**

This paper introduces Sci2Pol-Bench and Sci2Pol-Corpus for evaluating and fine-tuning LLMs on policy brief generation from scientific papers. Sci2Pol-Bench contains 18 tasks across a five-stage taxonomy (Autocompletion, Understanding, Summarization, Generation, Verification) built on 85 expert-written paper-brief pairs. The authors evaluate 13 LLMs and reveal significant limitations. Sci2Pol-Corpus contains 639 paper-brief pairs curated from 5.6M policy documents. Fine-tuning three models on this corpus yields improvements, with Gemma-27B surpassing GPT-4o and DeepSeek-V3.

**Strengths:**

1. Addresses the gap between scientific research and policy communication through a well-designed five-stage taxonomy that mirrors real brief-writing workflows.

2. Uses 85 expert-written paper-brief pairs from prestigious venues, ensuring reliable ground truth. The 18 tasks enable fine-grained evaluation across multiple capabilities.

3. Three-stage pipeline (retrieval from 5.6M documents, LLM filtering, in-context polishing) with validation studies addressing information leakage concerns.

**Weaknesses:**

1. Only 85 test pairs and 639 training pairs, concentrated in climate/energy and health sociology. No evidence of generalization to other scientific domains (CS, math, humanities, economics).

2. Tasks 7-15 use Gemini-2.5-Pro as judge, while GPT-o3 generates data for Tasks 5,11,13,16,18 and polishes all corpus documents. Models in GPT/Gemini families may benefit from evaluation alignment with their own generation patterns.

3. Table 3 shows Gemma-27B-SFT outperforms GPT-4o, but lacks analysis of what was learned (factual grounding vs. style vs. pattern memorization). No qualitative comparison of base vs. fine-tuned outputs.

**Questions:**

1. Can you validate LLM judges with human experts? Provide inter-rater agreement between Gemini-2.5-Pro and multiple political science experts for Tasks 11-15.

2. How does performance transfer across domains? Test fine-tuned models on policy briefs from economics, technology policy, or other areas beyond Nature/JHSB domains.

3. What are common failure modes? Categorize errors (factual/structural/stylistic/logical) for top models on each task.

---

> ### Author Response · Authors · 2025-11-19
> **Cross-domain generalization analysis & MiniMax-M2 for dataset construction**
>
> We appreciate the detailed and valuable review. The revised manuscript has now been uploaded, with all **changes highlighted in blue**.
>
> ---
>
> > `W1 & Q2`: Cross-domain generalization analysis.
>
> **Response**
>
> Thanks for your insightful comment. We have added the cross-domain generalization analysis in **Appendix I**.
>
> Due to the scarcity of readily available peer-reviewed scientific paper-policy brief pairs in other domains, we conduct an exploratory evaluation using one representative paper from each of the following important fields: AI, Robotics, Mathematics, Economics, and Humanities.
> The generated policy briefs are then subjected to human evaluation.
> We test three models: Gemma-27B-SFT, DeepSeek-V3, and GPT-4o.
>
> We provide the details of 5 papers as follows:
>
> (i) AI: "Generative AI needs adaptive governance". We show the results in Tables 36 and 37.
>
> (ii) Robotics: "Anticipatory policy design for robots in public spaces". We show the results in Tables 38 and 39.
>
> (iii) Mathematics: "Educational policy to improve mathematics instruction at scale: Conceptualizing contextual factors". We show the results in Tables 40 and 41.
>
> (iv) Economics: "Financial inclusion, the shadow economy and economic growth in developing economies". We show the results in Tables 42 and 43.
>
> (v) Humanities: "Science cited in policy documents: Evidence from the Overton database". We show the results in Tables 44 and 45.
>
> The resulting evaluation indicates that our supervised fine-tuning successfully generalizes to other domains.
>
> Please refer to Appendix I of the revised manuscript for details.
>
> ---
>
> > `W2`: Problems when using Gemini as a judge and GPT-o3 to construct the dataset.
>
> **Response (Part 1)**
>
> Thanks for your valuable comment. For the problem of using GPT for dataset construction,  we replace GPT-o3 by MiniMax-M2 to do the experiments again in Appendix H.11.2. For the problem of using Gemini as a judge, we add the “Independent Judge” experiments in Appendix H.6.
>
> 1. **Problem of Using GPT-o3 for Dataset Construction**
>
> We design an ablation study using Tasks 11 and 13 as illustrative examples. The experiment proceeds as follows:
>
> (i) We replicate the original data construction method, but replace the data generation model (GPT-o3) with MiniMax-M2.
>
> (ii) We then evaluate the performance of GPT-4o, DeepSeek-V3, and Mistral-Large on the data constructed by MiniMax-M2 for Tasks 11 and 13.
>
> (iii) The evaluation utilizes a dual LLM Judge setup, employing both Gemini and MiniMax-M2. The full results are presented in the following Table 1. These results demonstrate consistency when employing different large language models in the dataset construction process.
>
> ### **Table 1 (Part 1): Model Performance on Tasks 11 and 13 (Gemini as Judge)**
>
> | Model        | T11 (GPT-o3) | T13 (GPT-o3) | T11 (MiniMax-M2) | T13 (MiniMax-M2) |
> |------------------|------------------|------------------|-----------------------|-----------------------|
> | **DeepSeek-V3**     | 91.50±0.93 | 90.29±0.99 | 91.10±1.01 | 89.93±0.94 |
> | **GPT-4o**           | 89.17±0.98 | 72.81±1.50 | 89.53±0.94 | 71.97±1.43 |
> | **Mistral-Large**   | 87.64±0.92 | 64.10±1.93 | 87.02±0.97 | 63.98±1.34 |
>
> ### **Table 1 (Part 2): Model Performance on Tasks 11 and 13 (MiniMax-M2 as Judge)**
>
> | Model        | T11 (GPT-o3) | T13 (GPT-o3) | T11 (MiniMax-M2) | T13 (MiniMax-M2) |
> |------------------|------------------|------------------|-----------------------|-----------------------|
> | **DeepSeek-V3**      | 54.32±2.44 | 65.56±1.08 | 54.98±2.34 | 65.78±1.11 |
> | **GPT-4o**           | 52.01±2.40 | 62.54±1.24 | 51.78±2.34 | 62.83±1.29 |
> | **Mistral-Large**    | 51.71±1.56 | 61.64±1.46 | 50.94±1.36 | 61.03±1.44 |

---

> ### Author Response · Authors · 2025-11-19
> **Independent judge testing**
>
> > `W2`: Problems when using Gemini as a judge and GPT-o3 to construct the dataset.
>
> **Response (Part 2)**
>
> 2. **Problem of Using Gemini as Judge**
>
> Independent Judge: To reduce potential bias from using Gemini as a single automated judge in Tasks 7-15, we include additional evaluation results using MiniMax-M2 as an independent judge. We first report the evaluation results obtained using MiniMax-M2. The averaged scores are presented in the following Table 2, and the detailed task-level results for Tasks 7–15 are shown in Table 3. Across all models and tasks, the scores produced by Gemini and MiniMax-M2 exhibit strong agreement in both relative model ranking and within-task performance trends. These analyses demonstrate that the conclusion from using Gemini as the judge is reliable.
>
> ### **Table 2: Averaged Performance Using MiniMax-M2 as the Judge**
>
> | Model | Auto. (1–4) | Under. (5–6) | Sum. (7–10) | Gene. (11–15) | Ver. (16–18) | Avg. | Gain |
> |------|-------------|---------------|--------------|----------------|---------------|------|------|
> | **LLaMA-3.1-8B-IT** | 27.12±2.53 | 47.74±1.54 | 68.61±1.20 | 66.83±1.75 | 56.59±1.27 | 54.59±1.66 | – |
> | **LLaMA-3.1-8B-SFT** | 31.27±2.90 | 44.34±1.48 | 73.81±1.06 | 76.81±1.32 | 63.50±1.08 | 60.20±1.57 | **+5.61** |
> | **Gemma-3-12B** | 42.96±2.79 | 69.61±1.28 | 84.39±0.98 | 73.42±1.45 | 63.05±1.06 | 66.49±1.51 | – |
> | **Gemma-3-12B-SFT** | 43.14±2.86 | 69.53±1.25 | 85.91±0.96 | 82.89±1.53 | 64.59±1.05 | 70.19±1.53 | **+3.70** |
> | **Gemma-3-27B** | 43.60±2.83 | 67.82±1.42 | 82.72±0.91 | 79.65±1.56 | 65.79±0.98 | 68.70±1.54 | – |
> | **Gemma-3-27B-SFT** | 45.39±2.90 | 67.44±1.36 | 85.91±0.99 | 85.41±1.43 | 67.17±0.96 | **71.59±1.57** | **+2.89** |
> | **DeepSeek-V3** | 39.54±3.06 | 79.35±1.28 | 85.70±0.98 | 83.95±1.60 | 67.74±0.85 | 71.25±1.55 | – |
> | **GPT-4o** | 52.17±3.00 | 77.17±1.32 | 80.01±0.97 | 77.80±1.48 | 62.98±0.82 | 70.05±1.52 | – |
>
> ---
>
> ### **Table 3 (Part 1): Agreement Between Gemini and MiniMax-M2 on Tasks 7–10**
>
> | Model | T7 Gemini | T7 MiniMax | T8 Gemini | T8 MiniMax | T9 Gemini | T9 MiniMax | T10 Gemini | T10 MiniMax |
> |-------|-----------|------------|-----------|------------|-----------|------------|-------------|--------------|
> | **LLaMA-3.1-8B-IT** | 67.38±0.05 | 64.11±1.14 | 67.38±0.05 | 70.40±1.36 | 59.03±0.05 | 68.93±1.23 | 63.88±0.05 | 71.00±1.07 |
> | **LLaMA-3.1-8B-SFT** | 71.17±1.45 | 71.89±1.05 | 76.00±1.14 | 76.92±1.18 | 84.72±1.10 | 81.80±1.14 | 86.66±1.11 | 76.64±0.89 |
> | **Gemma-3-12B** | 71.18±0.05 | 76.50±0.98 | 65.45±0.05 | 67.64±0.99 | 76.71±0.05 | 74.06±0.95 | 85.60±1.02 | 75.48±0.98 |
> | **Gemma-3-12B-SFT** | 79.17±1.39 | 81.63±0.99 | 81.75±1.22 | 87.34±1.09 | 88.38±1.03 | 82.06±0.95 | 89.76±1.00 | 80.53±0.81 |
> | **Gemma-3-27B** | 75.98±0.05 | 79.96±0.88 | 68.57±0.05 | 71.29±0.95 | 76.03±0.05 | 74.63±0.96 | 91.33±0.87 | 83.99±0.84 |
> | **Gemma-3-27B-SFT** | 85.42±1.15 | 83.76±1.02 | 83.40±1.03 | 89.35±1.09 | 87.30±1.00 | 82.95±0.98 | 91.76±0.90 | 85.57±0.89 |
> | **DeepSeek-V3** | 80.95±0.05 | 82.90±0.95 | 81.90±0.05 | 87.65±0.93 | 77.72±0.05 | 80.78±1.09 | 91.50±0.93 | 84.46±0.93 |
> | **GPT-4o** | 75.20±0.06 | 77.30±0.91 | 71.38±0.06 | 75.99±1.00 | 72.70±0.06 | 78.13±0.97 | 89.17±0.98 | 79.76±0.99 |
>
> ---
>
> ### **Table 3 (Part 2): Agreement Between Gemini and MiniMax-M2 on Tasks 11–15**
>
> | Model | T11 Gemini | T11 MiniMax | T12 Gemini | T12 MiniMax | T13 Gemini | T13 MiniMax | T14 Gemini | T14 MiniMax | T15 Gemini | T15 MiniMax |
> |--------|-------------|--------------|--------------|--------------|--------------|--------------|--------------|--------------|--------------|--------------|
> | **LLaMA-3.1-8B-IT** | 82.35±1.55 | 44.95±2.61 | 64.62±1.65 | 55.48±1.51 | 55.12±1.90 | 51.66±1.16 | 67.74±1.89 | 70.27±1.83 | 59.05±1.55 | 60.61±1.64 |
> | **LLaMA-3.1-8B-SFT** | 86.66±1.11 | 51.28±1.56 | 71.10±1.47 | 62.27±1.22 | 74.19±1.83 | 62.56±1.43 | 83.41±1.61 | 74.05±1.03 | 72.76±1.75 | 67.35±1.36 |
> | **Gemma-3-12B** | 85.60±1.02 | 50.71±1.89 | 70.22±1.15 | 60.23±1.26 | 70.69±2.29 | 60.89±1.25 | 84.46±1.29 | 74.89±1.41 | 75.71±1.43 | 68.55±1.44 |
> | **Gemma-3-12B-SFT** | 89.76±1.00 | 53.58±2.50 | 71.93±1.39 | 63.09±1.30 | 70.92±2.08 | 61.99±1.38 | 84.18±1.52 | 75.06±1.24 | 76.06±1.66 | 69.21±1.24 |
> | **Gemma-3-27B** | 91.33±0.87 | 53.50±2.27 | 75.23±1.12 | 63.22±1.51 | 88.12±1.32 | 63.63±1.33 | 88.54±1.24 | 77.76±1.21 | 80.89±1.26 | 70.82±1.50 |
> | **Gemma-3-27B-SFT** | 91.76±0.90 | 55.20±2.20 | 78.26±1.35 | 66.53±1.11 | 67.34±2.45 | 60.53±1.45 | 86.65±1.53 | 77.78±1.02 | 83.65±1.75 | 75.81±1.37 |
> | **DeepSeek-V3** | 91.50±0.93 | 54.32±2.44 | 78.24±1.46 | 66.38±1.71 | 90.29±0.99 | 65.56±1.08 | 87.62±1.28 | 77.98±1.47 | 83.51±1.65 | 74.45±1.30 |
> | **GPT-4o** | 89.17±0.98 | 52.01±2.40 | 72.69±1.05 | 63.19±1.11 | 72.81±1.50 | 62.54±1.24 | 78.36±1.54 | 71.56±1.42 | 68.93±1.31 | 65.61±1.25 |

---

> ### Author Response · Authors · 2025-11-19
> **SFT analysis and human agreement testing**
>
> > `W3`: What SFT learns.
>
> **Response**
>
> Thanks for your comment. We are sorry for the oversight. We have added the detailed analysis in Appendix H.4.
>
>  We identify four aspects of learning: factual grounding, style, pattern alignment, and implicit skill acquisition. A detailed comparison of base and fine-tuned outputs is provided in Table 15, with additional analyses of leading-model shortcomings in Tables 16 and 17. We present the core ideas here for clarity. For complete details, please refer to Appendix H.4.
>
> 1. Factual Grounding. Enforces "strict factual alignment" by: (i) Relying Exclusively on Paper: Prohibiting external information and speculation. (ii) Capturing Major Findings: Ensuring summaries match the study's full scope. (iii) Avoiding Hallucinations: Reducing invented facts via exposure to accurate exemplars.
>
> 2. Style and Structure. Exposure to policy conventions and the fixed four-part layout (Policy Problem, Findings, Methods, Implications) enables: (i) Consistent Structure: Outputs reliably follow the required four-section organization. (ii) Policy-Brief Style: Outputs use concise, declarative phrasing and policy-relevant language. (iii) Improved Organization: Coherent transitions and clear logical links between sections. (iv) Reduced Verbosity: Maintaining informational density without unnecessary padding.
>
> 3. Pattern Alignment. When constructing the supervised training dataset, we use three (paper, brief) exemplars and a fixed schema for in-context polishing. This step teaches generalizable structural patterns: (i) Abstract Patterns: Learning how to frame a policy problem and derive implications strictly from the study's results. (ii) Cross-Paper Generalization: Transferring the shared functional structure to unseen domains.
>
> 4. Implicit Skill Acquisition. SFT promotes crucial latent skills: (i) Functional-Length Adaptation: Adjusting discussion depth based on paper complexity, not sample length. (ii) Disciplined Reasoning: Consistently avoiding speculation and providing accurate interpretation of empirical results. (iii) Strong Alignment: Producing policy implications that logically and directly follow the paper’s findings.
>
> > `Q1`: Human agreement testing.
>
> **Response**
>
> Thanks for your question. We add the human evaluation agreement in Appendix H.7.1.
>
> We ask two human evaluators to directly score model outputs and compare their averaged scores with those produced by the Gemini judge. We randomly sample 10 cases from Tasks 9 and 11-15 for human evaluation (using the same 10 cases across Tasks 11--15). Due to the limited number of evaluation questions, we report only the final results and do not conduct bootstrap significance testing. We show the results in the following Table 3. The results show that Gemma-27B-SFT consistently outperforms DeepSeek-V3 and GPT-4o, aligning with our Gemini-based evaluation. For completeness, we present one representative example for Tasks 9 and 15 along with brief analyses in Appendix H.7.1 (Tables 22 - 24)
>
>
> ### **Table 4 (Part 1): Direct Human Scoring on Tasks 9, 11, 12**
>
> | Model | T9 Gemini | T9 Human | T11 Gemini | T11 Human | T12 Gemini | T12 Human |
> |-------|-----------|----------|------------|-----------|------------|-----------|
> | **Gemma-3-27B-SFT** | 83.40±1.03 | 82.00 | 91.76±0.90 | 89.20 | 78.26±1.35 | 75.10 |
> | **DeepSeek-V3** | 76.90±0.05 | 74.00 | 91.50±0.93 | 88.40 | 78.24±1.46 | 75.60 |
> | **GPT-4o** | 71.38±0.06 | 70.00 | 89.17±0.98 | 86.30 | 72.69±1.05 | 71.30 |
>
> ---
>
> ### **Table 4 (Part 2): Direct Human Scoring on Tasks 13, 14, 15**
>
> | Model | T13 Gemini | T13 Human | T14 Gemini | T14 Human | T15 Gemini | T15 Human |
> |-------|-------------|------------|-------------|------------|-------------|------------|
> | **Gemma-3-27B-SFT** | 67.34±2.45 | 65.20 | 86.65±1.53 | 83.25 | 83.65±1.75 | 80.00 |
> | **DeepSeek-V3** | 90.29±0.99 | 84.50 | 87.62±1.28 | 84.50 | 83.51±1.65 | 79.10 |
> | **GPT-4o** | 72.81±1.50 | 68.20 | 78.36±1.54 | 78.25 | 68.93±1.31 | 65.90 |

---

> ### Author Response · Authors · 2025-11-19
> **Common failure modes**
>
> > `Q3`: Common failure modes.
>
> **Reponse**
>
> Thanks for your question. We have added the common failure modes in Appendix H.3. We quote here for clarity.
>
> Autocompletion (Tasks 1--4). On scientific and policy autocompletion and sentence reordering, leading models often rely on shallow lexical cues rather than discourse structure. For Tasks 1--2, models frequently choose continuations that share surface overlap (entities, topical words) but violate local rhetorical flow (e.g., jumping to limitations or policy implications prematurely). For Tasks 3--4, they can correctly identify the ``introductory'' sentence but struggle to order the remaining sentences when all share similar vocabulary. These failures reflect limited pattern alignment to domain-specific discourse patterns and weak implicit modeling of how scientific and policy arguments unfold.
>
> Understanding (Tasks 5--6). For sentence-level classification (Task 5), frontier models often conflate Policy Implications with Recommendations, or mislabel methodological descriptions as findings when they contain numerical results. This indicates insufficient specialization to Sci2Pol rhetorical roles and weak style awareness of policy-brief section functions. In contrast, performance on general scientific knowledge (Task 6) is comparatively strong and less affected by our domain-specific SFT; we treat Task 6 as a sanity check that specialization to Sci2Pol does not destroy broad knowledge.
>
> Summarization (Tasks 7--10). For policy-oriented summarization, leading models systematically underperform along two axes. First, failures in factual grounding: models often hallucinate policy framings, quantitative effects, or stakeholder groups that are not present in the selected paragraphs, especially in Tasks 7 (Policy Problem) and 10 (Policy Implications). Second, failures in style and pattern alignment: generated outputs drift toward generic "science communication" prose, ignoring the requested section type (e.g., mixing methods into findings, or inserting recommendations in the policy problem). Tasks 8--9 reveal additional issues with functional-length adaptation: models either compress complex methods and findings into vague topical summaries, or over-elaborate with boilerplate explanations not grounded in the input text.
>
> Generation (Tasks 11--15). When asked to generate sections or full briefs from the full paper, strong models exhibit compounding errors. For Tasks 11--14, they often produce sections that (i) are not cleanly separated, (ii) rely on generic policy tropes rather than paper-specific evidence, or (iii) introduce external context and speculative recommendations beyond the study’s scope. These reflect weaknesses in factual grounding, style discipline, and implicit skill at mapping paper structure into the four-section brief template. For full-brief generation (Task 15), models sometimes maintain local coherence within paragraphs but fail at global pattern alignment: section ordering is irregular, headings are omitted or merged, and later sections contradict earlier claims.
>
> Verification (Tasks 16--18). In verification tasks, frontier models tend to over-accept plausible-sounding claims, leading to high recall but poor precision. For Task 16, models often label partially supported or overstated claims as fully supported, revealing limited sensitivity to scope qualifiers and caveats in the source paper. For Tasks 17--18, they struggle to distinguish subtly contradicted policy implications from valid ones, especially when the implication is globally reasonable but locally misaligned with the provided evidence. These failure modes are rooted in weak factual grounding and insufficient disciplined reasoning about what is actually entailed by the text.
>
> ---
>
> We hope that we address your concerns. Please feel free to ask if you have further concerns.

---

### Official Review · Reviewer_RJb4 · 2025-10-31

**Soundness:** 3
**Presentation:** 3
**Contribution:** 4
**Rating:** 6
**Confidence:** 4

**Summary:**

The authors contribute an evaluation benchmark and corpus for the translation of scientific research papers to policy briefs: Sci2Pol-Bench and Sci2Pol-Corpus. For Sci2Pol-Bench, they detail a rigorous construction process of 18 tasks drawn from 85 expert-written paper-brief pairs, e.g., multiple-choice probes and open-ended generation tasks, along with associated metrics (e.g., F1, LLM-as-judge). For Sci2Pol-Corpus, they mine high-quality paper-brief pairs using a large-scale web dataset and LLM-as-judge filtering, obtaining 639 pairs. Lastly, they evaluate several frontier models and find finetuning on Sci2Pol-Corpus enables smaller models (e.g., Gemma-3-27B) to beat near-frontier models (DeepSeek-V3).

**Strengths:**

- Important setting and strong motivation: the conversion of scientific knowledge into content helpful to policymakers is increasingly important as AI matures and its real-world impact increases. Many simpler benchmarks are saturated by frontier models, and therefore contributing challenging expert-driven, real-world tasks is an important direction for the field.
- The paper is clear and well-written.
- The pipeline for evaluation and data curation is very meticulous. These details are clearly presented and will be very useful, such as to those tackling settings such as constructing RL environments for other domains.

**Weaknesses:**

The major challenge with new benchmarks is whether they are already saturated or likely to be saturated soon.
* Could you provide a human baseline on Sci2Pol-Bench (e.g., with a policy expert)? Without it, it is hard to judge whether the benchmark is really unsaturated, or if there is just very high label noise or ambiguity.
* Could you provide an analysis of the saturation of Sci2Pol-Bench with respect to some measure of model capability, such as pretraining FLOPs or release date (e.g., similar to the analysis of Epoch AI [1])? This would provide some idea of how soon this benchmark might be saturated.
* [Less critical] If possible, could you report the agreement between expert human graders and the LLM grader function, e.g., for the Reference-Free Score in Tasks 7-10 or Reference-based Score in Tasks 11-15?

[1] Kwa et al., Measuring AI Ability to Complete Long Tasks, 2025. https://arxiv.org/abs/2503.14499

**Questions:**

Please see Weaknesses above for my concrete suggestions to improve the paper. These are more minor clarifications:
- How did you source your experts, e.g., the political experts mentioned at line 202?
- Do you use experts in all of your task constructions involving selection, e.g., Tasks 7-10 where you find summarizing paragraphs?

---

> ### Author Response · Authors · 2025-11-19
> **Human baseline and saturation analysis**
>
> Thanks for your detailed and valuable review. We have uploaded the revised manuscript. All revisions are **marked in blue**.
>
> ---
>
> > `W1`: Human baseline.
>
> **Response**
>
> Thanks for the insightful comments. We have added the human baseline in Appendix H.2.
>
> We recruit two university-affiliated evaluators and compensate each with \$100. Because full evaluation is time-intensive, we do not assess all questions. Instead, we randomly sample 25 questions from each task in Tasks 1–10 and 16–18. For Tasks 11–15, we randomly sample three questions each, using the same three underlying scientific papers across all five tasks to maintain consistency. Due to the limited number of evaluation questions, we report only the final results and do not conduct bootstrap significance testing. We show the results in the following Table 1, together with the state-of-the-art Grok-3-beta results.
> These results indicate that current LLMs still have meaningful room for improvement.
>
> ### **Table 1 (Part 1): Autocompletion (T1–T4), Understanding (T5–T6), Summarization (T7)**
>
> | Model | T1 (F1) | T2 (F1) | T3 (F1) | T4 (F1) | T5 (F1) | T6 (F1) | T7 (Gem.) |
> |--------|---------|---------|---------|---------|---------|---------|-----------|
> | **Evaluator 1** | 92.00 | 84.00 | 80.00 | 84.00 | 100.00 | 92.00 | 94.20 |
> | **Evaluator 2** | 96.00 | 92.00 | 76.00 | 80.00 | 96.00 | 92.00 | 91.60 |
> | **Grok-3-beta** | 71.05±2.77 | 63.76±2.79 | 33.12±2.92 | 35.15±3.08 | 90.42±0.80 | 69.83±1.64 | 83.45±0.05 |
>
> ---
>
> ### **Table 1 (Part 2): Summarization (T8–T10), Generation (T11–T14)**
>
> | Model | T8 (Gem.) | T9 (Gem.) | T10 (Gem.) | T11 (Gem.) | T12 (Gem.) | T13 (Gem.) | T14 (Gem.) |
> |--------|------------|------------|-------------|-------------|-------------|-------------|-------------|
> | **Evaluator 1** | 96.20 | 91.00 | 89.00 | 95.67 | 87.24 | 88.74 | 96.84 |
> | **Evaluator 2** | 94.60 | 92.20 | 87.20 | 94.19 | 89.12 | 87.32 | 91.23 |
> | **Grok-3-beta** | 86.88±0.05 | 83.55±0.05 | 79.15±0.05 | 92.35±0.63 | 79.50±0.86 | 82.67±1.40 | 89.38±1.06 |
>
> ---
>
> ### **Table 1 (Part 3): Generation (T15), Verification (T16–T18), Overall Average**
>
> | Model | T15 (Gem.) | T16 (F1) | T17 (F1) | T18 (F1) | Average |
> |--------|-------------|-----------|-----------|-----------|----------|
> | **Evaluator 1** | 93.45 | 100.00 | 84.00 | 100.00 | 91.57 |
> | **Evaluator 2** | 95.12 | 100.00 | 92.00 | 100.00 | 91.48 |
> | **Grok-3-beta** | 89.58±0.94 | 98.60±0.42 | 59.26±1.67 | 98.48±0.50 | 77.01±1.20 |
>
> ---
>
> > `W2`: Saturation analysis.
>
> **Response**
>
> Thanks for your important comments. We have added the detailed analysis in Appendix H.12.
>
> We show the visualization of saturation analysis as here: [figure for saturation analysis](https://www.dropbox.com/scl/fi/q72b1387a897qmbel0u5b/saturation_analysis.png?rlkey=rorstlnp1ahr5dwa2x95y8c3u&st=6tgvgbwp&dl=0).
>
> We report the relationship between Sci2Pol-Bench performance and three commonly used model characteristics: (a) model size, (b) pre-training FLOPs, and (c) release date.
>
> (i) Score vs. Model Size. Very large models (e.g., GPT-4o) perform strongly, but several mid-sized models, such as DeepSeek-V3, match or exceed them, indicating that parameter count alone does not determine policy-reasoning quality. Smaller models ($<10B$) consistently underperform, suggesting a minimum capacity threshold.
>
> (ii) Score vs. Pre-training FLOPs. Although the highest-FLOPs models generally achieve the strongest results, many models in the $10^{24}$--$10^{25}$ FLOP range overlap substantially, showing diminishing returns once compute passes a sufficient scale.
>
> (iii) Score vs. Release Date. The third panel visualizes performance relative to the earliest publicly documented release month. While newer models (2024–2025) generally occupy the upper performance range, the significant variation observed within yearly cohorts indicates that recency is not the dominant factor. Instead, our supervised fine-tuning results indicate that alignment strategies and the specific data mixture used in training show a stronger influence on Sci2Pol-Bench performance.
>
> Overall, the three panels show that Sci2Pol-Bench performance does not follow a simple scaling law. Model size, compute, and release date each matter, but do not fully explain cross-model differences. Instead, high performance is often associated with improved alignment strategies, targeted supervised finetuning, and richer training mixtures. These are factors that help models reason about scientific evidence and policy trade-offs beyond what raw scale alone can provide.

---

> ### Author Response · Authors · 2025-11-19
> **Human evaluation agreement & political experts clarification**
>
> > `W3`: Human evaluation agreement.
>
> **Response**
>
> Thanks for your valuable feedback. We have added the human agreement testing in Appendix H.7.1.
>
> We ask two human evaluators to directly score model outputs and compare their averaged scores with those produced by the Gemini judge. We randomly sample 10 cases from Tasks 9 and 11-15 for human evaluation (using the same 10 cases across Tasks 11--15). Due to the limited number of evaluation questions, we report only the final results and do not conduct bootstrap significance testing. We show the results in the following Table 2. The results show that Gemma-27B-SFT consistently outperforms DeepSeek-V3 and GPT-4o, aligning with our Gemini-based evaluation. For completeness, we present one representative example for Tasks 9 and 15 along with brief analyses in Appendix H.7.1 (Tables 22 - 24)
>
>
> ### **Table 2 (Part 1): Direct Human Scoring on Tasks 9, 11, 12**
>
> | Model | T9 Gemini | T9 Human | T11 Gemini | T11 Human | T12 Gemini | T12 Human |
> |-------|-----------|----------|------------|-----------|------------|-----------|
> | **Gemma-3-27B-SFT** | 83.40±1.03 | 82.00 | 91.76±0.90 | 89.20 | 78.26±1.35 | 75.10 |
> | **DeepSeek-V3** | 76.90±0.05 | 74.00 | 91.50±0.93 | 88.40 | 78.24±1.46 | 75.60 |
> | **GPT-4o** | 71.38±0.06 | 70.00 | 89.17±0.98 | 86.30 | 72.69±1.05 | 71.30 |
>
> ---
>
> ### **Table 2 (Part 2): Direct Human Scoring on Tasks 13, 14, 15**
>
> | Model | T13 Gemini | T13 Human | T14 Gemini | T14 Human | T15 Gemini | T15 Human |
> |-------|-------------|------------|-------------|------------|-------------|------------|
> | **Gemma-3-27B-SFT** | 67.34±2.45 | 65.20 | 86.65±1.53 | 83.25 | 83.65±1.75 | 80.00 |
> | **DeepSeek-V3** | 90.29±0.99 | 84.50 | 87.62±1.28 | 84.50 | 83.51±1.65 | 79.10 |
> | **GPT-4o** | 72.81±1.50 | 68.20 | 78.36±1.54 | 78.25 | 68.93±1.31 | 65.90 |
>
> ---
>
> > `Q1`: How did you source your experts?
>
> **Response**
>
> Our experts were sourced from internal collaborators within the same research affiliation.
>
> (i) Specifically, one of our corresponding authors is a domain expert in policy analysis, and several postdoctoral researchers on the team possess formal expertise in public policy.
>
> (ii) This project was initiated by the corresponding author with expertise in policy analysis, who proposed the core idea of a scientific paper to policy brief generation.
>
> (iii) The interdisciplinary nature of this task necessitated collaboration between policy experts and AI researchers, bringing our team together. These domain experts directly participated in the evaluation, and they possessed the relevant subject-matter knowledge required for Sci2Pol-Bench.
>
> > `Q2`: Do you use experts in all of your task constructions involving selection?
>
> **Response**
>
> Yes, experts were involved in every task that required selection, interpretation, or validation of scientific or policy-relevant content. For Tasks 7–10 (reference-free summarization), the experts provided guidance on how to perform paragraph selection, and all extracted or selected text was subsequently reviewed by our experts to ensure that the chosen segments accurately reflected the scientific claims in the source paper.
>
> ---
>
> We hope that we address your concerns. Please feel free to ask if you have further concerns.

---

> ### Comment · Reviewer_RJb4 · 2025-11-22
>
> I appreciate the authors' thorough response. The added human baseline and saturation analysis increases my confidence that the Sci2Pol benchmark is unsaturated by current frontier LLMs and also has a high ceiling. Along with the clear importance of the problem setting, this convinces me that this benchmark will be broadly useful to the community, so I am increasing my score.

---

> > ### Author Response · Authors · 2025-11-22
> >
> > Thank you very much for your feedback. We are glad to hear that we were able to address your concerns, and we appreciate your effort throughout the review process and your willingness to raise the score. If you have any further questions or suggestions, please feel free to let us know.

---

### Official Review · Reviewer_UZs4 · 2025-11-01

**Soundness:** 2
**Presentation:** 3
**Contribution:** 3
**Rating:** 4
**Confidence:** 3

**Summary:**

The paper introduces Sci2Pol-Bench, a five-stage benchmark (Autocompletion, Understanding, Summarization, Generation, Verification) for turning scientific papers into policy briefs, and Sci2Pol-Corpus, a curated training set to fine-tune LLMs for this task. The benchmark draws on 85 expert-written paper–brief pairs from venues such as Nature Energy and Nature Climate Change and spans 18 tasks (mixed MCQ and open-ended). For generation tasks, the authors argue ROUGE/BERTScore are inadequate and instead use an LLM-as-judge rubric aligned to paper-grounded criteria. They evaluate 13 models, then fine-tune three open models on a 639-pair corpus created via citation linking over 5.6M policy documents, LLM-based filtering, and in-context polishing. They report consistent gains and claim a fine-tuned Gemma-27B can outperform GPT-4o/DeepSeek-V3 on their benchmark.

**Strengths:**

* Clear problem framing & timely task: Scientific-to-policy translation is impactful and under-served in current evaluation suites; the five-stage taxonomy mirrors a realistic writing workflow and is easy to reason about.

* Well-structured benchmark: 18 tasks with transparent construction and metrics per stage; the dataset composition and task table are clearly presented and grounded in real expert-authored briefs.

* Thoughtful metric discussion: The paper diagnoses why ROUGE/BERTScore miss quality signals in policy writing and proposes rubric-driven LLM judging tailored to sections (problem, findings, methods, implications) and full briefs.

* Empirical breadth: the authors evaluate13 diverse models, and results are summarized across taxonomy stages.

**Weaknesses:**

* Evaluation circularity & judge dependence: Many key results rely on proprietary LLM judges (Gemini-2.5-Pro for reference-free and reference-based scoring; o3 for filtering/polishing). This creates potential vendor/prompt-specific bias, and the headline claim “Gemma-27B-SFT surpasses GPT-4o/DeepSeek-V3” is only as strong as the judge’s alignment to human experts. Human evaluation (even on a subset) or judge triangulation (≥2 independent judges + agreement analysis) would strengthen credibility.

* Risk of construction-time leakage/style imprinting: The in-context polishing step conditions on three expert exemplars from the 85-pair set; while Appendix H.6 reportedly checks leakage, the paper still asks the reader to trust that polishing didn’t imprint benchmark-specific style cues that advantage similar models/decoding. A stronger defense would include ablations that: (i) polish with held-out exemplars from outside the 85 pairs; (ii) evaluate on a disjoint external set of expert briefs.

* Heuristic retrieval bias: Keeping only policy documents citing ≤3 papers biases toward single-paper briefs and may not reflect real policy synthesis, which often aggregates many sources. An analysis of how this heuristic shapes style/length and whether models trained on such pairs generalize to multi-source briefs is missing.

* Limited human oversight in label generation: Several tasks use o3 to generate items/labels with “manual review” mentioned briefly (e.g., “correct three issues” in Lines 267-268). More details on reviewer counts, agreement, and correction rates would calibrate trust in classification ground truth.

* Ethics & release clarity: The Ethics section is brief; given the policy domain, more detail is warranted on licensing, redistribution boundaries, bias auditing of the judge, and safeguards to prevent models from emitting over-confident policy prescriptions not supported by evidence. The paper mentions anonymous links and fair-use reasoning but lacks concrete release protocols and impact/risk assessments specific to policy advice.

**Questions:**

Please address all issues raised in the Weaknesses section.

---

> ### Author Response · Authors · 2025-11-19
> **Independent Judge Results for Weakness 1**
>
> Thank you for the detailed and valuable review. We have uploaded the revised manuscript, with **all revisions marked in blue**.
>
> ---
>
> > `W1`: Human evaluation or independent judge.
>
> **Response (Part 1 for W1)**
>
> Thanks for the insightful comment. We add the “Independent Judge” experiments in Appendix H.6 and “Human Evaluation” experiments in Appendix H.7.1.
>
> 1. **Independent Judge** (Appendix H.6). To reduce potential bias from using Gemini as a single automated judge in Tasks 7-15, we include additional evaluation results using MiniMax-M2 as an independent judge. We first report the evaluation results obtained using MiniMax-M2. The averaged scores are presented in the following Table 1, and the detailed task-level results for Tasks 7–15 are shown in Table 2. Across all models and tasks, the scores produced by Gemini and MiniMax-M2 exhibit strong agreement in both relative model ranking and within-task performance trends.
>
> ### **Table 1: Averaged Performance Using MiniMax-M2 as the Judge**
>
> | Model | Auto. (1–4) | Under. (5–6) | Sum. (7–10) | Gene. (11–15) | Ver. (16–18) | Avg. | Gain |
> |------|-------------|---------------|--------------|----------------|---------------|------|------|
> | **LLaMA-3.1-8B-IT** | 27.12±2.53 | 47.74±1.54 | 68.61±1.20 | 66.83±1.75 | 56.59±1.27 | 54.59±1.66 | – |
> | **LLaMA-3.1-8B-SFT** | 31.27±2.90 | 44.34±1.48 | 73.81±1.06 | 76.81±1.32 | 63.50±1.08 | 60.20±1.57 | **+5.61** |
> | **Gemma-3-12B** | 42.96±2.79 | 69.61±1.28 | 84.39±0.98 | 73.42±1.45 | 63.05±1.06 | 66.49±1.51 | – |
> | **Gemma-3-12B-SFT** | 43.14±2.86 | 69.53±1.25 | 85.91±0.96 | 82.89±1.53 | 64.59±1.05 | 70.19±1.53 | **+3.70** |
> | **Gemma-3-27B** | 43.60±2.83 | 67.82±1.42 | 82.72±0.91 | 79.65±1.56 | 65.79±0.98 | 68.70±1.54 | – |
> | **Gemma-3-27B-SFT** | 45.39±2.90 | 67.44±1.36 | 85.91±0.99 | 85.41±1.43 | 67.17±0.96 | **71.59±1.57** | **+2.89** |
> | **DeepSeek-V3** | 39.54±3.06 | 79.35±1.28 | 85.70±0.98 | 83.95±1.60 | 67.74±0.85 | 71.25±1.55 | – |
> | **GPT-4o** | 52.17±3.00 | 77.17±1.32 | 80.01±0.97 | 77.80±1.48 | 62.98±0.82 | 70.05±1.52 | – |
>
> ---
>
> ### **Table 2 (Part 1): Agreement Between Gemini and MiniMax-M2 on Tasks 7–10**
>
> | Model | T7 Gemini | T7 MiniMax | T8 Gemini | T8 MiniMax | T9 Gemini | T9 MiniMax | T10 Gemini | T10 MiniMax |
> |-------|-----------|------------|-----------|------------|-----------|------------|-------------|--------------|
> | **LLaMA-3.1-8B-IT** | 67.38±0.05 | 64.11±1.14 | 67.38±0.05 | 70.40±1.36 | 59.03±0.05 | 68.93±1.23 | 63.88±0.05 | 71.00±1.07 |
> | **LLaMA-3.1-8B-SFT** | 71.17±1.45 | 71.89±1.05 | 76.00±1.14 | 76.92±1.18 | 84.72±1.10 | 81.80±1.14 | 86.66±1.11 | 76.64±0.89 |
> | **Gemma-3-12B** | 71.18±0.05 | 76.50±0.98 | 65.45±0.05 | 67.64±0.99 | 76.71±0.05 | 74.06±0.95 | 85.60±1.02 | 75.48±0.98 |
> | **Gemma-3-12B-SFT** | 79.17±1.39 | 81.63±0.99 | 81.75±1.22 | 87.34±1.09 | 88.38±1.03 | 82.06±0.95 | 89.76±1.00 | 80.53±0.81 |
> | **Gemma-3-27B** | 75.98±0.05 | 79.96±0.88 | 68.57±0.05 | 71.29±0.95 | 76.03±0.05 | 74.63±0.96 | 91.33±0.87 | 83.99±0.84 |
> | **Gemma-3-27B-SFT** | 85.42±1.15 | 83.76±1.02 | 83.40±1.03 | 89.35±1.09 | 87.30±1.00 | 82.95±0.98 | 91.76±0.90 | 85.57±0.89 |
> | **DeepSeek-V3** | 80.95±0.05 | 82.90±0.95 | 81.90±0.05 | 87.65±0.93 | 77.72±0.05 | 80.78±1.09 | 91.50±0.93 | 84.46±0.93 |
> | **GPT-4o** | 75.20±0.06 | 77.30±0.91 | 71.38±0.06 | 75.99±1.00 | 72.70±0.06 | 78.13±0.97 | 89.17±0.98 | 79.76±0.99 |
>
> ---
>
> ### **Table 2 (Part 2): Agreement Between Gemini and MiniMax-M2 on Tasks 11–15**
>
> | Model | T11 Gemini | T11 MiniMax | T12 Gemini | T12 MiniMax | T13 Gemini | T13 MiniMax | T14 Gemini | T14 MiniMax | T15 Gemini | T15 MiniMax |
> |--------|-------------|--------------|--------------|--------------|--------------|--------------|--------------|--------------|--------------|--------------|
> | **LLaMA-3.1-8B-IT** | 82.35±1.55 | 44.95±2.61 | 64.62±1.65 | 55.48±1.51 | 55.12±1.90 | 51.66±1.16 | 67.74±1.89 | 70.27±1.83 | 59.05±1.55 | 60.61±1.64 |
> | **LLaMA-3.1-8B-SFT** | 86.66±1.11 | 51.28±1.56 | 71.10±1.47 | 62.27±1.22 | 74.19±1.83 | 62.56±1.43 | 83.41±1.61 | 74.05±1.03 | 72.76±1.75 | 67.35±1.36 |
> | **Gemma-3-12B** | 85.60±1.02 | 50.71±1.89 | 70.22±1.15 | 60.23±1.26 | 70.69±2.29 | 60.89±1.25 | 84.46±1.29 | 74.89±1.41 | 75.71±1.43 | 68.55±1.44 |
> | **Gemma-3-12B-SFT** | 89.76±1.00 | 53.58±2.50 | 71.93±1.39 | 63.09±1.30 | 70.92±2.08 | 61.99±1.38 | 84.18±1.52 | 75.06±1.24 | 76.06±1.66 | 69.21±1.24 |
> | **Gemma-3-27B** | 91.33±0.87 | 53.50±2.27 | 75.23±1.12 | 63.22±1.51 | 88.12±1.32 | 63.63±1.33 | 88.54±1.24 | 77.76±1.21 | 80.89±1.26 | 70.82±1.50 |
> | **Gemma-3-27B-SFT** | 91.76±0.90 | 55.20±2.20 | 78.26±1.35 | 66.53±1.11 | 67.34±2.45 | 60.53±1.45 | 86.65±1.53 | 77.78±1.02 | 83.65±1.75 | 75.81±1.37 |
> | **DeepSeek-V3** | 91.50±0.93 | 54.32±2.44 | 78.24±1.46 | 66.38±1.71 | 90.29±0.99 | 65.56±1.08 | 87.62±1.28 | 77.98±1.47 | 83.51±1.65 | 74.45±1.30 |
> | **GPT-4o** | 89.17±0.98 | 52.01±2.40 | 72.69±1.05 | 63.19±1.11 | 72.81±1.50 | 62.54±1.24 | 78.36±1.54 | 71.56±1.42 | 68.93±1.31 | 65.61±1.25 |
>
> ---

---

> ### Author Response · Authors · 2025-11-19
> **Human evaluation and information leakage testing of in-context polishing**
>
> > `W1`: Human evaluation or independent judge.
>
> **Response (Part 2 for W1)**
>
> 2. **Human Evaluation** (Appendix H.7.1 in the revised version). We ask two human evaluators to directly score model outputs and compare their averaged scores with those produced by the Gemini judge. We randomly sample 10 cases from Tasks 9 and 11-15 for human evaluation (using the same 10 cases across Tasks 11--15). Due to the limited number of evaluation questions, we report only the final results and do not conduct bootstrap significance testing. We show the results in the following Table 3. The results show that Gemma-27B-SFT consistently outperforms DeepSeek-V3 and GPT-4o, aligning with our Gemini-based evaluation. For completeness, we present one representative example for Tasks 9 and 15 along with brief analyses in Appendix H.7.1 (Tables 22 - 24).
>
>
> ### **Table 3 (Part 1): Direct Human Scoring on Tasks 9, 11, 12**
>
> | Model | T9 Gemini | T9 Human | T11 Gemini | T11 Human | T12 Gemini | T12 Human |
> |-------|-----------|----------|------------|-----------|------------|-----------|
> | **Gemma-3-27B-SFT** | 83.40±1.03 | 82.00 | 91.76±0.90 | 89.20 | 78.26±1.35 | 75.10 |
> | **DeepSeek-V3** | 76.90±0.05 | 74.00 | 91.50±0.93 | 88.40 | 78.24±1.46 | 75.60 |
> | **GPT-4o** | 71.38±0.06 | 70.00 | 89.17±0.98 | 86.30 | 72.69±1.05 | 71.30 |
>
> ---
>
> ### **Table 3 (Part 2): Direct Human Scoring on Tasks 13, 14, 15**
>
> | Model | T13 Gemini | T13 Human | T14 Gemini | T14 Human | T15 Gemini | T15 Human |
> |-------|-------------|------------|-------------|------------|-------------|------------|
> | **Gemma-3-27B-SFT** | 67.34±2.45 | 65.20 | 86.65±1.53 | 83.25 | 83.65±1.75 | 80.00 |
> | **DeepSeek-V3** | 90.29±0.99 | 84.50 | 87.62±1.28 | 84.50 | 83.51±1.65 | 79.10 |
> | **GPT-4o** | 72.81±1.50 | 68.20 | 78.36±1.54 | 78.25 | 68.93±1.31 | 65.90 |
>
> ---
>
> > `W2`:  Information leakage of in-context polishing.
>
> **Response**
>
> Thank you for your comment. We have addressed this by including a detailed discussion on “Polishing with held-out exemplars” in Appendix H.10.2. Furthermore, the evaluation on a “disjoint external set” is now described in Appendix H.10.3.
>
> 1. **In-Context Polishing with Three New Published Pairs** (Appendix H.10.2). We use the three latest published paper-brief pairs as the three in-context examples to polish the dataset, and use this newly polished dataset to do supervised tuning. The three examples include: “Nutritional outcomes of irrigation expansion”, “Managing development choices is essential to reduce coastal flood risk in China”, and “Heat pumps can help alleviate residential energy insecurity in the USA”. We show the results in the following Table 4. Gains remain stable, and this shows that in-context polishing does not induce leakage.
>
> ### **Table 4: Performance on Task 15 using three new non-overlapping in-context polishing examples.**
>
> | Model | T15 (Original 3 ICL Samples) | Gain | T15 (New 3 ICL Samples) | Gain |
> |-------|----------------------------|------|---------------------------|------|
> | **LLaMA-3.1-8B-IT** | 59.05±1.55 | – | 59.05±1.55 | – |
> | **LLaMA-3.1-8B-SFT** | 72.76±1.75 | **+13.71** | 72.91±1.96 | **+13.86** |
> | **Gemma-3-12B** | 75.71±1.43 | – | 75.71±1.43 | – |
> | **Gemma-3-12B-SFT** | 76.06±1.66 | **+0.35** | 76.31±1.65 | **+0.60** |
> | **Gemma-3-27B** | 80.89±1.26 | – | 80.89±1.26 | – |
> | **Gemma-3-27B-SFT** | 83.65±1.75 | **+2.76** | 83.43±1.80 | **+2.54** |
>
> 2. **Evaluating on New Eight Test Samples** (Appendix H.10.3).
> We use another eight newly published paper-brief pairs to generate Task 15. These include all the new peer-reviewed pairs from May 10th, 2025, to Nov. 12th, 2025. We list the details of these eight new pairs in Appendix J.3.
> We show the results in the following Table 5. Gains remain stable, and this shows that in-context polishing does not induce leakage.
>
> ### **Table 5: Performance on Task 15 with eight new expert-written samples.**
>
> | Model | T15 (Original 85 Expert Pairs) | Gain | T15 (New 8 Expert Pairs) | Gain |
> |-------|--------------------------------|------|----------------------------|------|
> | **LLaMA-3.1-8B-IT** | 59.05±1.55 | – | 58.43±1.12 | – |
> | **LLaMA-3.1-8B-SFT** | 72.76±1.75 | **+13.71** | 72.02±1.36 | **+13.59** |
> | **Gemma-3-12B** | 75.71±1.43 | – | 74.81±1.12 | – |
> | **Gemma-3-12B-SFT** | 76.06±1.66 | **+0.35** | 75.58±1.43 | **+0.77** |
> | **Gemma-3-27B** | 80.89±1.26 | – | 81.02±1.62 | – |
> | **Gemma-3-27B-SFT** | 83.65±1.75 | **+2.76** | 83.21±1.46 | **+2.19** |
> | **DeepSeek-V3** | 83.51±1.65 | – | 82.52±1.25 | – |
> | **GPT-4o** | 68.93±1.31 | – | 67.53±1.59 | – |

---

> ### Author Response · Authors · 2025-11-19
>
> > `W3`: Heuristic retrieval bias when using policy documents citing fewer than 3 papers.
>
> **Response**
>
> Thank you for raising this point. We are sorry for the confusion. We believe there is a misunderstanding about the goal of our benchmark.
>
> Our aim is not to model or evaluate broad policy synthesis across many sources. Instead, Sci2Pol-Bench intentionally focuses on a specific, widely used genre of policy writing: policy briefs written about a single scientific paper.  Because our target task is **generate a policy brief for a given paper**, we curated examples where the policy document is explicitly centered on one scientific study. The $\leq$3-citation heuristic is therefore not a biasing constraint but a filtering step to ensure that the retrieved policy outputs are indeed single-paper briefs, rather than multi-source synthesis documents.
>
> Given this design choice, models trained on Sci2Pol-Bench are specifically intended to generalize within the single-document policy brief genre. Our focus is on summarizing one scientific article into a policy-style brief, not on producing multi-document syntheses. We acknowledge that multi-source policy synthesis is an important, but distinct, task that falls outside the scope of this benchmark.
>
> Furthermore, we have **added this precise scope limitation to the paper in Appendix D.1**, which we quote here for clarity:
>
> “While Sci2Pol-Bench is intentionally designed around single-paper policy briefs, this focus also imposes an inherent limitation. Our benchmark does not capture the broader genre of multi-source policy synthesis, where policymakers integrate evidence from many studies. As a result, our benchmark does not assess, nor does our dataset enhance, a model’s capacity to aggregate heterogeneous sources or integrate multiple lines of evidence.”
>
> > `W4`: Limited human oversight in label generation.
>
> **Response**
>
> Thank you for this helpful and important comment. We sincerely apologize for this oversight. We have now incorporated the necessary details in the Ethic Statement (Page 10) under the section “Details of Human Involvement in Label Generation”, which we quote here for clarity:
>
> "Three tasks (5, 16, and 18) use GPT-o3 to propose initial labeled items, and all of them undergo full human verification. Each task is reviewed independently by two reviewers, with 100\% reviewer agreement across all accepted items. For all 1,200 samples in Task 5, no corrections are required. For Task 16, reviewers examine all 850 samples and correct three mislabeled cases (correction rate $\approx 0.35\%$). For Task 18, reviewers validate all 706 generated contradictory implications, and no corrections are needed."
>
> > `W5`: Ethics & release clarity.
>
> **Response**
>
> Thank you for this important point. We are sorry for this oversight. We have now incorporated the necessary details in the Ethic Statement (Page 10) under the section “Copyright and Use of Scientific Papers”, which we quote here for clarity:
>
> "The benchmark and dataset are built from publicly accessible content such as article metadata. We release only article DOIs, metadata, access dates, policy brief metadata, and our annotations. The benchmark and dataset are released under CC-BY-NC 4.0, with a two-tier protocol (open metadata + controlled transformed excerpts), and no publisher-owned text is redistributed.
> Derived tasks (e.g., summarization in Sci2Pol-Bench) rely solely on reformulated excerpts for non-commercial academic purposes.
>
> We also document potential impacts and implement explicit safeguards: all tasks require evidence-based summarization (not prescriptive advice), prompts prohibit unsupported claims, and evaluation judges penalize fabricated, speculative, or over-confident policy recommendations. To reduce evaluation variance and bias, we employ rubric-anchored scoring, multi-model judging, and expert-labeled calibration samples. These measures establish clear licensing boundaries, release protocols, bias-mitigation procedures, and risk-assessment practices appropriate for policy-oriented datasets.
>
> For transparency, we provide a detailed list of all used paper–brief pairs in Appendix J, including DOIs, sources, and access dates."
>
> ---
>
> We hope that we address your concerns. Please feel free to ask if you have further concerns.

---

### Author Response · Authors · 2025-11-26
**Global Rebuttal Response**

**Dear Reviewers and Area Chairs**:

We thank the reviewers for the helpful comments and detailed reviews. We have answered all the questions and addressed all the concerns in detail in rebuttal and revisions. Any changes or modifications made to the submitted version are **highlighted in blue** in this updated draft.

In response to the reviewers’ suggestions, we have substantially improved the clarity and completeness of the paper. Revisions include updated figures and reorganized tables, refined statements, clearer ethics and release documentation, addition of a human baseline, independent-judge and human evaluations, evaluation on newly discovered pairs and cross-domain datasets, potential circularity analysis using GPT, saturation testing, and more detailed analyses of experimental results.

---

**Revision Details**

**Major revisions** include:
1. Added a human baseline for Sci2Pol-Bench (Appendix H.2).

2. Used MiniMax-M2 as an independent judge to replace Gemini (Appendix H.6).

3. Replaced the Gemini judge with human evaluators (Appendix H.7.1).

4. Incorporated newly identified paper-brief pairs for in-context polishing and reconstructed the “Generation” task (Appendix H.10.2 and H.10.3).

5. Added cross-domain generalization analysis (Appendix I).

6. Tested potential circularity when using GPT to generate datasets (Appendix H.11.2).

7. Performed saturation testing (Appendix H.12).

8. Added a detailed analysis of common failure modes of LLMs (Appendix H.3).

9. Added an analysis of what supervised fine-tuning learns (Appendix H.4).

**Minor revisions** include:

1. Page 1: Added preliminary references and refined statements.

2. Page 2: Improved figure clarity and wrote out the full name of JSHB.

3. Page 8: Reorganized Table 2 and Figure 3 for better readability.

---

We hope these revisions address the reviewers' concerns and improve the overall quality of our paper.

---

### Meta-Review · Area_Chair_RVt4 · 2026-01-06

**Summary:**

This paper proposes Sci2Pol-Bench and Sci2Pol-Corpus, a benchmark and fine-tuning dataset for scientific-to-policy brief generation. Reviewers broadly agree that the problem setting is important, the benchmark taxonomy is well motivated, and the construction pipeline is carefully engineered. The key decision hinges on whether the authors sufficiently mitigate concerns around evaluation circularity, judge dependence, limited human validation, domain scope, and generalization.

The rebuttal meaningfully strengthens the paper by adding human baselines, incorporating an additional independent language model judge, conducting leakage checks and saturation analysis. These additions substantially reduce several of the previously raised concerns. While some validation remains relatively small in scale, the overall evidence now supports the robustness and relevance of the benchmark. Taken together, the paper meets the acceptance bar and would represent a valuable contribution to the community.

**Reviewer Concerns:**

**Concerns Addressed:**
* Added human baselines across multiple stages, showing that current models remain below expert performance.
* Introduction of an additional independent LLM judge (MiniMax-M2), along with limited human agreement checks, partially mitigates judge dependence..
* Additional experiments with held-out exemplars and newly published paper–brief pairs reduce the risk of construction-time leakage.
* Saturation analysis suggests the benchmark is not trivially solved by scale alone.

**Concerns Still Outstanding:**
* Human evaluations remain very limited in scale, without statistical testing, making them supportive but not decisive.
* Cross-domain evaluation is exploratory, relying on single examples per domain, and is insufficient to substantiate broad generalization claims.
* Despite triangulation, the benchmark still fundamentally relies on LLM-based evaluation for its core conclusions.
* The focus on single-paper briefs is well-justified but limits real-world applicability to broader policy synthesis.

**Reviewer Scores:**

Reviewer UZs4: 4  -> 5 (although 5 points are not selectable)
* The rebuttal addresses many technical concerns, but remaining issues around evaluation robustness likely keep the paper borderline.

Reviewer RJb4: 6 -> 8
* Explicitly reports increased confidence following the addition of human baselines and saturation analysis.

Reviewer F3Nb: 6 -> 6
* Acknowledges improvements, but concerns about domain coverage and judge dependence remain.

**Overall:**
This is a well executed and timely benchmark paper that has been substantially strengthened through the rebuttal. While some limitations remain, particularly the limited scale of human validation and the continued reliance on LLM judges, these issues do not outweigh the overall contributions.  I lean slightly toward acceptance.

---

### Decision · Program_Chairs · 2026-01-26

Accept (Poster)